# Is Cross-Validation the Gold Standard to Estimate Out-of-sample Model Performance?

**Garud Iyengar, Henry Lam, Tianyu Wang**[*]
Department of Industrial Engineering and Operations Research
Columbia University
New York, NY 10027
`{gi10,khl2114,tw2837}@columbia.edu`

## Abstract

Cross-Validation (CV) is the default choice for estimate the out-of-sample performance of machine learning models. Despite its wide usage, their statistical benefits have remained half-understood, especially in challenging nonparametric regimes. In this paper we fill in this gap and show that, in terms of estimating the out-of-sample performances, for a wide spectrum of models, CV does not statistically outperform the simple "plug-in" approach where one reuses training data for testing evaluation. Specifically, in terms of both the asymptotic bias and coverage accuracy of the associated interval for out-of-sample evaluation, $K$-fold CV provably cannot outperform plug-in regardless of the rate at which the parametric or nonparametric models converge. Leave-one-out CV can have a smaller bias as compared to plug-in; however, this bias improvement is negligible compared to the variability of the evaluation, and in some important cases leave-one-out again does not outperform plug-in once this variability is taken into account. We obtain our theoretical comparisons via a novel higher-order Taylor analysis that dissects the limit theorems of testing evaluations, which applies to model classes that are not amenable to previously known sufficient conditions. Our numerical results demonstrate that plug-in performs indeed no worse than CV in estimating model performance across a wide range of examples.

## 1 Introduction

Cross-validation (CV) is considered the default choice for estimating the out-of-sample performance of machine learning models [54, 31, 4] and more general data-driven optimization models [13]. Its main rationale is to evaluate models using a testing set that is different from training, so as to provide a reliable estimate of the model generalization ability. Leave-one-out CV (LOOCV) [5, 7], which repeatedly evaluates models trained using all but one observation on the left-out observation, is a prime approach; however, it is computationally demanding as it requires model re-training for the same number of times as the sample size. Because of this, $K$-fold CV, which reduces the number of model re-training down to $K$ times (where $K$ is typically 5–10), becomes a popular substitute [34, 42].

Despite their wide usage, the statistical benefits of CV have remained understood mostly for parametric models. In nonparametric regimes, especially those involving slow model convergence rates, their general performances as well as comparisons with the naive "plug-in" approach, i.e., simply reuses all the same training data for model evaluation, stay essentially open. Part of the challenge comes from the subtle inter-dependence of model convergence rates and other characteristics with

---

[*]Authors ordered alphabetically. More information on the data and code are available at `https://github.com/wangtianyu61/CV_GoldStandard`.

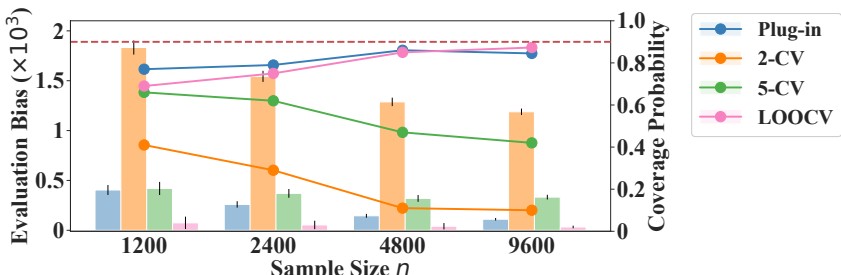

Figure 1: Evaluation biases and coverage probabilities of interval estimates (with nominal level 90%) for the mean-squared error evaluation of a fitted random forest regressor (default setup in `scikit-learn` in [49] with $n^{0.4}$ subsamples in each tree), across 500 experimental replications. The bar chart shows the evaluation bias, defined as the absolute mean difference between the estimated and true performance (the vertical line at the top of each bar shows the corresponding standard error). The lines show the coverage probabilities.

the correlation between training and validation sets across folds. Consequently, existing results are either based on limit theorems designed to "center" at the average-of-folds instead of full-size model [60], or restricted to specific models (e.g., linear) [7, 9] or specific (fast) rates [57]. Our goal in this paper, on a high level, is to fill in the challenging regimes beyond these established results, and as such answer the question: *Are LOOCV and $K$-fold CV a "must-use" in estimating out-of-sample model performance in general and, if not, then under what situations are they worthwhile?*

More precisely, in this paper we conduct a systematic analysis to compare the accuracies in estimating model performances using LOOCV, $K$-fold CV and plug-in. We focus on the asymptotic bias and coverage accuracy of the associated interval estimate for the out-of-sample evaluation. Our main messages are: First, in terms of these asymptotic criteria, *$K$-fold CV never outperforms plug-in, regardless of the rate at which a parametric or nonparametric model converges*. Second, while LOOCV can have a smaller bias than plug-in, this bias improvement can be negligible compared to the evaluation variability and therefore, *in a range of important cases, LOOCV again does not outperform plug-in*. In particular, we show that all parametric models, as well as some nonparametric models including random forests and kNN with sufficient smoothness, fall into this range. Since LOOCV requires significantly more computation resources, this raises the caution that its use is not always necessary, despite its robust performance for model evaluation.

As a simple illustration, Figure 1 shows the evaluation quality of the squared error of a random forest regressor using 2-fold CV, 5-fold CV, LOOCV and plug-in. We see that 2- and 5-fold CVs suffer from larger biases than plug-in (shown in the bar chart) especially for large sample sizes, and correspondingly also significantly poorer coverages of the associated interval estimates (shown by the lines). On the other hand, LOOCV exhibits smaller biases than plug-in, but these do not transform into better coverages since the bias improvement is negligible compared to the statistical variability in the evaluation. Both intervals provide valid coverage guarantees for large sample sizes (shown by the lines). We highlight that this example is not a "cherry pick": Section 5 and Appendix F show similar conclusions for a wide array of numerical examples.

We close this introduction by briefly discussing our technical novelty. Our analysis framework to conclude all our comparisons is propelled by a novel higher-order Taylor analysis on the out-of-sample evaluation that account for the dependence between the trained model and the testing data. In contrast to merely sufficient conditions in the literature, under which plug-in and CV variants exhibit low biases and valid coverages, this analysis helps us provide a complete breakdown of how bias and coverage depend on the convergence rate of the model at hand. This in turn fills in the gap in understanding which methods outperform which others, in regimes that have appeared challenging for previous works.

## 2 Problem Framework

We consider the supervised learning setting with observations $\mathcal{D}_n := \{(X_i, Y_i)\}_{i \in [n]}$ drawn i.i.d. from the joint distribution $\mathbb{P}_{(X,Y)} := \mathbb{P}_X \times \mathbb{P}_{Y|X}$. We obtain a predictor $\hat{z}(x) = \mathcal{A}(\mathcal{D}_n; x)$ as a

function of $x$ with the output domain $\mathcal{Z}$, through a training procedure $\mathcal{A}$ on $\mathcal{D}_n$. We are interested in evaluating the out-of-sample performance $\mathbb{E}_{\mathbb{P}_{(X,Y)}}[\ell(\hat{z}(X);Y)]$, where $\ell(z;Y): \mathcal{Z} \times \mathcal{Y} \mapsto \mathbb{R}$ is the cost function. This evaluation can be a point estimate, or more generally an interval estimate $I(\alpha)$ that covers $\mathbb{E}_{\mathbb{P}_{(X,Y)}}[\ell(\hat{z}(X);Y)]$ with $1-\alpha$ probability, i.e., we aim to satisfy $\mathbb{P}_{\mathcal{D}_n}(\mathbb{E}_{\mathbb{P}_{(X,Y)}}[\ell(\hat{z}(X);Y)] \in I(\alpha)) \approx 1-\alpha$, where the outer probability $\mathbb{P}_{\mathcal{D}_n}$ is with respect to the data $\mathcal{D}_n$ used to construct $\hat{z}(\cdot)$. We focus on the low-dimensional asymptotic setting where $n \to \infty$ and $\mathcal{X}, \mathcal{Y}, \mathcal{Z}$ are of fixed dimensions and defer discussions to other regimes in Section 6.

Regarding the scope of our setup, $\ell$ can be the loss function for supervised learning (e.g., squared loss, cross-entropy loss), in which case $\hat{z}$ naturally denotes the predicted label and $(X, Y)$ denotes the feature-label pair. More generally, $\ell$ can denote a downstream optimization objective in a decision-making problem, in which case $Y$ denotes a random outcome that affects the objective given the contextual information $X$. This latter setup, which is called *contextual stochastic optimization* [13, 51], can be viewed as a generalization of supervised learning from building prediction models to prescriptive decision policies. For example, in the so-called newsvendor problem in operations management, the cost refers to monetary loss of a retailer determined by the order quantity $z$, and covariate $X$ refers to the market condition that drives stochastic demand $Y$ [8, 13]. Our framework in this paper applies to both the traditional supervised learning and prescriptive data-driven decision-making settings.

We consider three main methods: plug-in, LOOCV, and $K$-fold CV. We denote $\hat{A}_m$ and $I_m(\alpha)$ as the point estimate and $(1-\alpha)$-level interval estimate, using method $m \in \mathcal{M} = \{p, loocv, kcv\}$ referring to plug-in, LOOCV and $K$-fold CV respectively. For convenience, we denote $\mathbb{P}^*$ as the true joint distribution $\mathbb{P}_{(X,Y)}$ and $\hat{\mathbb{P}}_n = (1/n)\sum_{i=1}^n \delta_{(X_i,Y_i)}$ as the empirical distribution, and we denote $c(z)$ and $c_n(z)$ as the out-of-sample performance $\mathbb{E}_{\mathbb{P}^*}[\ell(z(X);Y)]$ and the plug-in evaluation of the out-of-sample performance $\mathbb{E}_{\hat{\mathbb{P}}_n}[\ell(z(X);Y)]$ for any decision mapping $z(x)$ respectively. We present our considered point and interval estimates, where the latter are all written in the form $I_m(\alpha) = [\hat{A}_m - z_{1-\alpha/2}\hat{\sigma}_m/\sqrt{n}, \hat{A}_m + z_{1-\alpha/2}\hat{\sigma}_m/\sqrt{n}], \forall m \in \mathcal{M}$ with:

$$\hat{A}_p = c_n(\hat{z}), \quad \hat{\sigma}_p = \sqrt{\frac{1}{n}\sum_{i\in[n]}(\ell(\hat{z}(X_i);Y_i) - \hat{A}_p)^2}, \tag{1}$$

$$\hat{A}_{kcv} = \frac{1}{n}\sum_{k\in[K]}\sum_{i\in N_k}\ell(\hat{z}^{(-N_k)}(X_i);Y_i), \quad \hat{\sigma}_{kcv} = \sqrt{\frac{1}{n}\sum_{k\in[K]}\sum_{i\in N_k}(\ell(\hat{z}^{(-N_k)}(X_i);Y_i) - \hat{A}_{kcv})^2}, \tag{2}$$

where $z_{1-\alpha/2}$ is the $(1-\alpha/2)$-quantile of the standard normal distribution, $\{N_k, k \in [K]\}$ is the collection of $K$ equal-length partitions of $[n]$ (for simplicity we assume $n$ is divisible by $K$) and $\hat{z}^{(-N_k)}(\cdot) := \mathcal{A}(\mathcal{D}_{(-N_k)}; \cdot)$, with $\mathcal{D}_{(-N_k)}$ denoting the data set that leaves out $\{(X_i, Y_i)\}_{i\in N_k}$. For the $K$-fold CV estimates, $\hat{A}_{kcv}$ and $I_{kcv}$, we always assume $K$ is fixed with respect to $n$ (e.g., $K = 2, 5, 10$). On the other hand, $\hat{A}_{loocv}$ and $I_{loocv}(\alpha)$ are defined by setting $K = n$ in (2). Note that there are alternative approaches to construct the interval estimates (e.g., nested cross validation in Appendix G.3), but the above are the most natural and have been shown to have statistical consistency properties as well as superior empirical performance over other intervals [10].

We impose the following regularity and optimality conditions on the cost function:

**Assumption 1 (Smoothness of Expected Cost)** *For any $x \in \mathcal{X}$, $v(z; x) := \mathbb{E}_{\mathbb{P}_{Y|x}}[\ell(z; Y)]$ is twice differentiable with respect to $z$ everywhere, where $\mathbb{P}_{Y|x}$ is the conditional distribution of $Y$ given $x$.*

**Assumption 2 (Regularity of Cost Function)** *For any $y \in \mathcal{Y}$, $\ell(z; y)$ is twice differentiable with respect to $z$ for every $y$. Moreover, $|\ell(z; y)|, \|\nabla_z \ell(z; y)\|_2$ are uniformly bounded in $z \in \mathcal{Z}$ and almost surely in $y$.*

**Assumption 3 (Optimality Conditions)** *$\mathcal{Z}$ is a bounded open set. The best mapping $z_o^*(x)$ that minimizes $v(z; x), \forall x \in \mathcal{X}$ satisfies the first and second-order optimality conditions. More precisely, $\forall x \in \mathcal{X}, \nabla_z v(z_o^*(x); x) = 0$, and $\nabla_{zz} v(z_o^*(x); x)$ is positive definite.*

While Assumption 2 is standard, we can relax it further to some non-smooth objectives including piecewise linear functions (e.g., $\ell(z; Y) = |z - Y|$); see Assumption 5 in Appendix B.2. The other assumptions above are commonly used in stochastic optimization [25, 29, 38]. We further allow constrained problems in Assumption 6 in Appendix B.2.

**Example 1** *The function $\ell(z; Y) = (z - Y)^2$ ($\ell_2$-regression), compact set $\mathcal{Y}$, and $z_o^*(x) = \mathbb{E}[Y | X = x]$ with $\mathcal{Z} = \{z : |z| < B\}$ for any $B > \max_{x \in \mathcal{X}} z_o^*(x)$ satisfy Assumptions 1, 2 and 3.*

Next, we distinguish between parametric and nonparametric models in Definitions 1 and 2 below, leaving further details on technical regularity conditions in Appendix B.3.

**Definition 1 (Parametric Model)** *$\hat{z}(x) = G(\hat{\theta}; x)$, where $\hat{\theta} = \operatorname{argmin}_{\theta \in \Theta} \sum_{i=1}^{n} \ell(G(\theta; X_i); Y_i) + \lambda_n R(\theta)$ for some regularization function $R(\theta)$. Regularity conditions of $G(\theta; X)$ are provided in Assumption 7 in Appendix B.3.1, which are satisfied by linear models (Example 2 of Section 3).*

**Definition 2 (Nonparametric Model)** *$\hat{z}(\cdot)$ is obtained through $\hat{z}(x) \in \operatorname{argmin}_{z \in \mathcal{Z}} \sum_{i \in [n]} w_{n,i}(x) \ell(z; Y_i)$ with weights $\{w_{n,i}(x)\}_{i \in [n]}$ depending on $\mathcal{D}_n$ and $x$. Regularity conditions of $w_{n,i}(\cdot)$ are provided in Assumption 8 in Appendix B.3.2, which are satisfied by the classical k-Nearest Neighbor (kNN) and forest learners (Examples 3, 4 of Section 3).*

Define $z^*(\cdot) := \mathcal{A}(\mathcal{D}_\infty; \cdot)$ as the *oracle* best model using the training procedure $\mathcal{A}$ with infinite data $\mathcal{D}_\infty$. We now define the notion of convergence rate order for model $\hat{z}(\cdot)$:

**Definition 3 (Convergence Rate)** *For a model $\hat{z}(\cdot)$, we say it has a convergence rate of order $\gamma \in (0, 1/2]$ if $\mathbb{E}_{\mathcal{D}_n}[\|\hat{z}(x) - z^*(x)\|_2] = \Theta(n^{-\gamma})$ for almost every $x$. Furthermore, we say $\hat{z}(\cdot)$ has a bias and variability convergence rate $\gamma_b, \gamma_v$ respectively if $\mathbb{E}_{\mathcal{D}_n}[\|\mathbb{E}_{\mathcal{D}_n}[\hat{z}(x)] - z^*(x)\|_2] = \Theta(n^{-\gamma_b})$ and $\mathbb{E}_{\mathcal{D}_n}[\|\hat{z}(x) - \mathbb{E}_{\mathcal{D}_n}[\hat{z}(x)]\|_2] = \Theta(n^{-\gamma_v})$ for almost every $x$. Consequently, $\gamma = \min\{\gamma_b, \gamma_v\}$.*

The overall convergence order $\gamma$ of $\hat{z}(\cdot)$ is determined by both its bias $\gamma_b$ and variability $\gamma_v$, whichever dominates. For parametric models in Definition 2, we naturally have $\gamma = \gamma_v = 1/2$ (see Proposition 1 in Appendix B.3.1). However, unless $\{G(\theta; x) : \theta \in \Theta\}$ contains the model $z_o^*(\cdot)$ that optimizes $v(z; \cdot)$, there is a discrepancy between $z_o^*(\cdot)$ and the limiting model $z^*(\cdot)$. For nonparametric models in Assumption 2, both $\gamma_b$ and $\gamma_v$ depend on the hyperparameter configuration and are often smaller than $1/2$. When their hyperparameters are properly chosen (e.g. Theorems 5 - 9 in [13]), we have $z^*(\cdot) = z_o^*(\cdot)$ thanks to the nonparametric power in eliminating model misspecification.

Lastly, we introduce the following stability conditions:

**Definition 4 (Stability)** *Denote two leave-one-out (LOO) stability notions $\alpha_n, \beta_n$ by:*

$$\text{(Expected LOO Stability)} \qquad \alpha_n := \max_{i \in [n]} \left\{ (\mathbb{E}_{\mathbb{P}^*, \mathcal{D}_n}[\|\hat{z}(X) - \hat{z}^{(-i)}(X)\|^2])^{\frac{1}{2}} \right\}, \qquad (3)$$

$$\text{(Pointwise LOO Stability)} \qquad \beta_n := \max_{i \in [n]} \left\{ \mathbb{E}_{\mathcal{D}_n}[|\hat{z}(X_i) - \hat{z}^{(-i)}(X_i)|] \right\}, \qquad (4)$$

*where the expectation in (3) is with respect to both the data $\mathcal{D}_n$, used to construct $\hat{z}$ and $\hat{z}^{(-i)}$, and $\mathbb{P}^*$ that generates $X$, while (4) has expectation taken with respect to only $\mathcal{D}_n$.*

Stability notions are first proposed in [15, 28, 45] and commonly used to provide generalization guarantees for CV [37, 39] as well as refined bounds under more relaxed stability in [16, 1, 2]. We assume the following:

**Assumption 4 (LOO Stability)** *$\hat{z}(\cdot)$ satisfies the expected LOO stability with $\alpha_n = o(n^{-1/2})$.*

This condition holds for many models in Definitions 1 and 2 [15, 28, 17] and is often imposed for the validity of plug-in and CV, e.g. in [10, 18]. For illustration, we provide an example of 1-NN in Appendix B.4.

## 3   Main Results

We present our main results on the evaluation bias and interval coverage for plug-in, $K$-fold CV and LOOCV. Unless specified otherwise, $\mathbb{E}$ and $\mathbb{P}$ in the following are taken with respect to $\mathcal{D}_n$.

**Theorem 1 (Bias)** *Suppose Assumptions 1, 2, 3 and 4 hold. Recall $\gamma, \gamma_v$ in Definition 3. Then, for $\hat{z}(\cdot)$ in Definitions 1 and 2:*

$$\mathbb{E}[c(\hat{z}) - \hat{A}_p] = \Theta(n^{-2\gamma_v}) > 0, \mathbb{E}[c(\hat{z}) - \hat{A}_{kcv}] = \Theta(n^{-2\gamma}) < 0, \mathbb{E}[c(\hat{z}) - \hat{A}_{loocv}] = o(n^{-1}) < 0.$$

**Theorem 2 (Coverage Validity)** *Suppose Assumptions 1, 2, 3 and 4 hold. Recall $\mathcal{M} = \{p, kcv, loocv\}$. Then, for $\hat{z}(\cdot)$ and $z^*(\cdot)$ in Definitions 1 and 2:*

- *If $\gamma > 1/4$, then: $\lim_{n\to\infty} \mathbb{P}(c(\hat{z}) \in I_m) = \lim_{n\to\infty} \mathbb{P}(c(z^*) \in I_m) = 1 - \alpha$, for $m \in \mathcal{M}$.*

- *If $\gamma \leq 1/4$, then $\lim_{n\to\infty} \mathbb{P}(c(z^*) \in I_m) < 1 - \alpha$ for $m \in \mathcal{M}$.*

  - *$\lim_{n\to\infty} \mathbb{P}(c(\hat{z}) \in I_p) \leq 1 - \alpha$, where equality holds if and only if $\beta_n = o\left(n^{-1/2}\right)$ (or $\gamma_v > 1/4$).*
  - *$\lim_{n\to\infty} \mathbb{P}(c(\hat{z}) \in I_{kcv}) < 1 - \alpha$.*
  - *$\lim_{n\to\infty} \mathbb{P}(c(\hat{z}) \in I_{loocv}) = 1 - \alpha$.*

In the following, we use Theorems 1 and 2 to compare plug-in, $K$-fold CV and LOOCV, and highlight our novelty relative to what is known in the literature. In a nutshell, the regime $\gamma \leq 1/4$ has been wide open and comprises our major contribution and necessitates our new theory described in Section 4.

**Comparing plug-in and $K$-fold CV.** Theorems 1 and 2 together stipulate that *plug-in is always no worse than $K$-fold CV* in terms of estimating out-of-sample performance. In terms of evaluation bias, plug-in is optimistic while $K$-fold CV is pessimistic. For such a bias direction, optimistic bias refers to underestimating the expected cost. Plug-in suffers such bias since it is estimated on the same training data. In contrast, pessimistic bias refers to overestimating the expected cost. CV suffers such bias since CV is unbiased for the evaluation using fewer training samples than the whole dataset, and appears more erroneous than it should be compared to the true evaluation using the whole dataset. For parametric models, since $\gamma = \gamma_v = 1/2$, their biases in Theorem 1 are both $\Theta(n^{-1})$. This recovers the results in [30, 36] in which case the bias is negligible when constructing intervals. However, this bias size is unknown for general nonparametric models in the literature. For these models, our new results show that the bias of plug-in is $\Theta(n^{-2\gamma_v})$ which is no bigger than that of $K$-fold CV since $\gamma \leq \gamma_v$. This behavior arises because, even though plug-in incurs an underestimation of $\Theta(n^{-2\gamma_v})$ due to the reuse of training and evaluation set, $K$-fold CV loses efficiency due to a loss of training sample from the data splitting, thus leading to an even larger bias of $\Theta(n^{-2\gamma})$.

The above comparisons are inherited to interval coverage. While plug-in and $K$-fold CV both exhibit asymptotically exact coverage for parametric models (included in the case $\gamma > 1/4$), their coverages differ for nonparametric models, with plug-in still always no worse than $K$-fold CV. Specifically, when $\gamma \leq 1/4$, $K$-fold CV incurs invalid coverage, whereas plug-in still yields valid coverage as long as $\gamma_v > 1/4$. This is because, in this regime, the bias of $K$-fold CV is bigger than its variability to affect coverage significantly while the bias of plug-in remains small enough to retain coverage validity. In the literature, [21, 18] show valid coverage using plug-in under some stability conditions, but it is unclear regarding their applicability to general models. On the other hand, for CVs, central limit theorems and hence coverage guarantees have been derived generally, but they are centered at the average performance of trained models across folds [42, 10, 26], and thus bear a gap between such an averaged performance and the true model performance. Recently, [56, 7] further show CV intervals can provide coverage guarantees when $\gamma > 1/4$, but they do not touch on the case $\gamma \leq 1/4$. Moreover, all the literature above do not demonstrate an explicit difference between $K$-fold and LOOCV in their results [7, 57, 10]. From these, our results on the regime $\gamma \leq 1/4$ where we characterize and conclude the difference between $K$-fold and LOOCV appear the first in the literature.

**Comparing plug-in and LOOCV.** When comparing with plug-in, LOOCV has a smaller, and pessimistic, bias $o(1/n)$. However, this bias improvement can be negligible compared to the evaluation variability captured in interval coverage, specifically when $\gamma > 1/4$ (which includes all parametric models) and when $\gamma \leq 1/4, \gamma_v > 1/4$. In the latter case in particular, the bias of plug-in, even though larger than LOOCV, is small enough to ensure valid coverage.

We visually summarize our discussions on biases and interval coverages in Figure 2. In particular, Figure 2(a)(b) display our new contributions on both plug-in and CVs under slow rate $\gamma$. We also see that plug-in intervals provide valid coverages for $(b)(c)$, while $K$-fold CV is only valid for $(c)$ and LOOCV is valid across $(a)(b)(c)$.

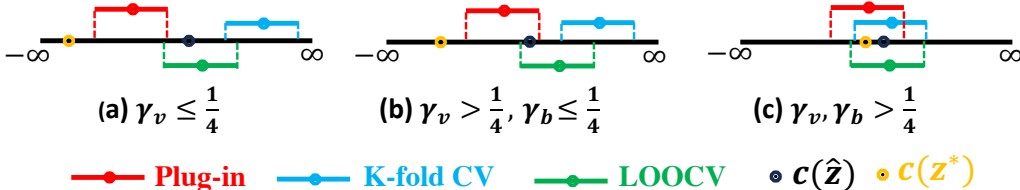

Figure 2: Concept plots of interval coverages of $c(\hat{z})$, and also $c(z^*)$, for our considered approaches across model rates, where the black line represents the value of the expected cost and a point is considered covered if it falls within the corresponding interval.

**Examples.** We exemplify the above insights with several specific models. Denote $d_x, d_y$ as the dimensions of $X$ and $Y$. We consider a regression problem with $\ell(z; Y) = (z - Y)^2$ and $d_x = 4, d_y = 1$, where $\mathbb{P}^* \in \mathcal{P}$ with:

$$\mathcal{P} = \{\mathbb{P}_X = U([0,1]^4), \mathbb{P}_{Y|x} = N(f(x), 1), \forall x \text{ with Lipschitz continuous } f(x)\}. \tag{5}$$

We consider the worst-case instance of $\mathbb{P}^*$, in the sense that $\gamma_b, \gamma_v$ take the smallest attainable values in Chapter 3 of [33].

**Example 2 ((Regularized) Linear-ERM [8])** $G(\theta; x) = \theta^\top x$, $\lambda_n = 1, R(\theta) = \|\theta\|_2^2$ satisfy Assumption 1. Specifically, $\gamma = \gamma_v = 1/2$, and all of plug-in, K-fold CV and LOOCV provide valid coverages for $c(\hat{z})$.

**Example 3 (kNN-Learner)** Denote the nearest index set $\mathcal{N}_{\mathcal{D}_n, x}(k_n) = \{i | X_i \text{ is a kNN of } x\}$. Then $w_{n,i}(x) = \mathbf{1}_{\{i \in \mathcal{N}_{\mathcal{D}_n, x}(k_n)\}}$ satisfies Assumption 2 with hyperparameter $k_n$. Specifically, $\gamma_v = -\log k_n/(2 \log n)$ and $\gamma_b = \log(k_n/n)/(4 \log n)$ (from Chapter 6 in [33]). If $k_n = \omega(\sqrt{n})$, then LOOCV and plug-in provide valid coverages for $c(\hat{z})$; otherwise, only LOOCV provide valid coverages.

**Example 4 (Forest Learner)** Consider a forest $\mathcal{F} = \{\tau_1, \ldots, \tau_T\}$, where each $\tau_i : \mathbb{R}^{d_x} \mapsto \{1, \ldots, L_i\}$ is a (tree) partition of $R^{d_x}$ into $L_i$ regions. Then $w_{n,i}(x) = \sum_{j=1}^T \mathbf{1}_{\{\tau_j(X_i) = \tau_j(x)\}}/T$ satisfies Assumption 2, where the hyperparameter is the subsampling ratio $\beta < 2/3$ in each tree. Specifically, $\gamma_v = (1 - \beta)/2$ from [57] and $\gamma_b < 1/6$ (from Lemma 5 in Appendix B.1). Then LOOCV and plug-in provide valid coverages for $c(\hat{z})$.

We summarize our theoretical comparisons in this section in Table 1, the first half of which shows our general comparisons in terms of both evaluation bias and interval coverage, while the second half illustrates our considered examples.

Table 1: Asymptotic bias and coverage for each approach, where ✓ and ✗ denote valid and invalid coverages. $o(\cdot), \Omega(\cdot)$ and $\omega(\cdot)$ follow the standard big O notation.

| Model | Specifications | $c(\hat{z}) - \hat{A}_p$ | $c(\hat{z}) - \hat{A}_{kcv}$ | $c(\hat{z}) - \hat{A}_{loocv}$ | $I_p$ | $I_{kcv}$ | $I_{loocv}$ |
|---|---|---|---|---|---|---|---|
| | | | Bias | | | Coverage Validity | |
| General | $\gamma > 1/4$ | $o(n^{-1/2})$ | $o(n^{-1/2})$ | | ✓ | ✓ | ✓ |
| | $\gamma_b \leq 1/4, \gamma_v > 1/4$ | $o(n^{-1/2})$ | $\omega(n^{-1/2})$ | $o(n^{-1})$ | ✓ | ✗ | ✓ |
| | $\gamma_v \leq 1/4$ | $\Omega(n^{-1/2})$ | $\Omega(n^{-1/2})$ | | ✗ | ✗ | ✓ |
| Specific $d_x = 4$ | Linear-ERM | $\Theta(n^{-1})$ | $\Theta(n^{-1})$ | | ✓ | ✓ | ✓ |
| | kNN with $k_n = \Theta(n^{1/4})$ | $\Theta(n^{-1/8})$ | $\Theta(n^{-1/8})$ | $o(n^{-1})$ | ✗ | ✗ | ✓ |
| | kNN with $k_n = \Theta(n^{2/3})$ | $\Theta(n^{-2/3})$ | $\Theta(n^{-1/12})$ | | ✓ | ✗ | ✓ |
| | Forest with $\beta = 0.4$ | $\Theta(n^{-0.3})$ | $\omega(n^{-1/6})$ | | ✓ | ✗ | ✓ |

Finally, we point out that Theorem 2 also shows, in addition to our evaluation target $c(\hat{z})$, the coverage on the *oracle* best performance $c(z^*)$. The latter is generally a different quantity than $c(\hat{z})$, but it plays an important role in our analysis. When $\gamma > 1/4$, $c(\hat{z})$ and $c(z^*)$ are very close and an interval for $c(\hat{z})$ is also valid to cover $c(z^*)$. On the other hand, when $\gamma \leq 1/4$, the statistical discrepancy

between $c(\hat{z})$ and $c(z^*)$ is too large for any interval estimates of $c(\hat{z})$ to be valid for $c(z^*)$, but nonetheless LOOCV and plug-in for $\gamma_v > 1/4$ can still validly cover $c(\hat{z})$ thanks to their small evaluation biases.

# 4 Roadmap of Theoretical Developments

We present the main theoretical ideas to show Theorems 1 and 2. Before going into details, we highlight the main novelties of our analyses: First, the biases for general nonparametric models in Theorem 1, which are unknown in the literature, require a different Taylor analysis compared with the parametric case available in [30]. Second, parts of Theorem 2 come from verifying the central limit theorems (CLTs) in [18, 10]. However, [18] only shows that CLTs hold for $c(\hat{z})$ when $\alpha_n, \beta_n = o(n^{-1/2})$ and does not show exactly when CLT fails; [10] only gives CLTs for CVs with a different center than $c(\hat{z}), c(z^*)$. In this regard, our main technical contribution is to fill in these theoretical gaps and characterize necessary conditions, instead of merely sufficient conditions, to conclude interval (in)validity across the entire spectrum.

## 4.1 Evaluation Bias

One key component of Theorem 1 hinges on the characterization of optimistic bias for plug-in, namely $\mathbb{E}_{\mathcal{D}_n}[c_n(\hat{z}) - c(\hat{z})]$, which captures the underestimation amount of the objective value relative to the truth when using an empirical estimator:

**Lemma 1 (Plug-in Bias)** *Suppose Assumptions 1, 2, 3 and 4 hold. For $\hat{z}(\cdot)$ in Assumption 2,*
$\mathbb{E}_{\mathcal{D}_n}[c_n(\hat{z})] - \mathbb{E}_{\mathcal{D}_n}[c(\hat{z})] = \Theta(n^{-2\gamma_v}) < 0.$

The proof of Lemma 1 relies on a novel second-order Taylor expansion centered at the *deterministic* decision $z_n(x) := \mathbb{E}[\hat{z}(x)]$ on both the empirical gap $c_n(\hat{z}) - c_n(z_n)$ and the true gap $c(z_n) - c(\hat{z})$ in nonparametric models $\hat{z}(\cdot)$. Here, we do not center them at the limiting decision $z^*(x)$ compared with the parametric setup from [4, 36] since in nonparametric models, $\hat{z}(\cdot) - z_n(\cdot)$ already captures the variability term that leads to the plug-in evaluation bias. Besides, in these nonparametric models, we need to further analyze the second-order difference $\mathbb{E}[\|\hat{z}(x) - z_n(x)\|_2^2]$ and $\mathbb{E}[\|\hat{z}(X_i) - z_n(X_i)\|_2^2]$, which requires a more involved analysis through a comparison on the asymptotic expansion terms of $\hat{z}(\cdot) - z_n(\cdot)$. To understand the optimistic bias further, we provide a constructive proof for kNN with an optimistic bias $\Theta(1/k_n)$ in Proposition 4 in Appendix D.1.

The results for CVs follow from the observation that $K$-fold CV (where here $K$ can be any number up to $n$) gives an unbiased evaluation for the model trained with $n(1 - 1/K)$ samples:

**Lemma 2 (CV Bias)** *Suppose Assumptions 1, 2, 3 and 4 hold. For $\hat{z}(\cdot)$ in Definitions 1 and 2,*
$\mathbb{E}_{\mathcal{D}_n}[\hat{A}_{kcv}] - \mathbb{E}_{\mathcal{D}_n}[c(\hat{z})] = \Theta(K^{-1}n^{-2\gamma}) > 0, \mathbb{E}_{\mathcal{D}_n}[\hat{A}_{loocv}] - \mathbb{E}_{\mathcal{D}_n}[c(\hat{z})] = o(n^{-1}) > 0.$

## 4.2 Interval Coverage

The proof of Theorem 2 hinges on the following equivalences of conditions among stability, convergence rate, and coverage validity. For plug-in, we have:

**Theorem 3 (Equivalence among Stability, Convergence Rate and Coverage Validity for Plug-in)**
*Suppose Assumptions 1, 2, 3 and 4 hold. Then the following three conditions are equivalent:*

$$\texttt{S1}: \beta_n = o(n^{-1/2}), \quad \texttt{S2}: \lim_{n \to \infty} \mathbb{P}(c(\hat{z}) \in I_p) = 1 - \alpha, \quad \texttt{S3}: \gamma_v > 1/4.$$

Theorem 3 is shown through three components of arguments, where the first two are our new technical contributions:

(1) `S3` is sufficient for `S1`. Intuitively, a faster rate of $\gamma_v$ means that the effect of one data point is usually small, implying a fast pointwise stability. This result follows from a refined decomposition of the variability term $\hat{z}(x) - \hat{z}^{(-i)}(x)$ through the influence of each point based on the asymptotic expansions of $\hat{z}(\cdot)$ in Assumption 2. We examine its bias and variability respectively, with formal results provided in Proposition 5 in Appendix E.1.1.

(2) S3 is necessary for S2. Since the variability of the interval width does not differ significantly (Lemma 7 in Appendix C), only the bias would lead to interval invalidity. From Lemma 1, if S3 does not hold, i.e., $\gamma_v \leq 1/4$, then $\mathbb{E}_{\mathcal{D}_n}[c_n(\hat{z}) - c(\hat{z})] = \Theta(n^{-1/2})$ and this implies that S2 does not hold from Proposition 6 in Appendix E.1.2.

(3) S1 is sufficient for S2. This follows from a direct verification of Lemma 2 in [18] to ensure asymptotic normality for plug-in and the small variability of the interval width (Lemma 7 in Appendix C). Furthermore, due to a small difference between $c(\hat{z})$ and $c(z^*)$ (Lemma 6 in Appendix C), the plug-in interval can also cover the quantity $\mathbb{E}_{\mathcal{D}_n}[c(\hat{z})]$ and $c(z^*)$ if $\gamma > 1/4$ (i.e. Corollary 1 in Appendix E.1.4).

In the above, we show that the condition $\beta_n = o(n^{-1/2})$ or $\gamma_v < 1/4$ is a necessary and sufficient condition to ensure a valid plug-in interval if Assumption 4 holds. Note that [18] demonstrate that in a specific nonparametric model, Assumption 4 is a necessary condition for the coverage invalidity of plug-in by providing a counterexample (Lemma 3 there). Our results are not directly comparable to theirs. First, we assume Assumption 4 throughout the entire paper and show that plug-in does not provide valid coverage guarantees when another stability notion, $\beta_n$, is large. Second, our results apply to a more general class of nonparametric models in Assumption 2 instead of the particular models and cost functions $\ell$ in [18].

**Theorem 4 (Equivalence between Convergence Rate and Coverage Validity for CV)** *Suppose Assumptions 1, 2, 3 and 4 hold. Then for $K$-fold CV, the following two conditions are equivalent:*

$$S4 : \gamma > 1/4, \quad S5 : \lim_{n \to \infty} \mathbb{P}(c(\hat{z}) \in I_{kcv}) = 1 - \alpha.$$

*For LOOCV, we always have $\lim_{n \to \infty} \mathbb{P}(c(\hat{z}) \in I_{loocv}) = 1 - \alpha$. However, $\lim_{n \to \infty} \mathbb{P}(c(z^*) \in I_m) < 1 - \alpha, \forall m \in \{kcv, loocv\}$ if $\gamma \leq 1/4$.*

To understand the necessary condition for $K$-fold CV, since $\gamma$ is small, the difference between the expected performance between the decision $\hat{z}(x)$ and $\hat{z}^{(-N_k)}(x)$ is not negligible, leading to a larger overestimate of performance from $c(\hat{z})$ compared with the interval width. However, following the stability condition from Assumption 4, LOOCV can still provide a valid coverage guarantee for $c(\hat{z})$, since the difference between $c_n(\hat{z})$ and $\hat{A}_{loocv}$ is always $o_p(n^{-1/2})$. In contrast, the sufficient condition for S5 follows by the asymptotic normality of CV through verifying Theorem 1 in [10].

## 5 Numerical Experiments

**Setups.** We consider two synthetic experiments to validate our theoretical results: (1) Regression problem: $\ell(z; Y) = (z - Y)^2$; (2) Conditional value-at-risk (CVaR) portfolio optimization: $\ell(z; Y) = z_v + \frac{1}{\eta}(-z_p^\top Y - z_v)^+$ with $\mathcal{Z} = \{z = (z_p, z_v) | \mathbf{1}^\top z_p = 1, z_p \geq \mathbf{0}\}$ for some $\eta \in (0, 1)$. In the regression example, we consider ridge regression, kNN with $k_n = \Theta(n^{2/3})$, and Forest with $\beta = 0.4$ (recall Example 4). In portfolio optimization, we consider sample average approximation (SAA, which belongs to Assumption 1) and kNN with $k_n = \Theta(n^{1/4})$. We run plug-in, 5-fold CV and LOOCV with nominal level $1 - \alpha = 0.9$. For each setting, we evaluate the following metrics with 500 experimental repetitions: (1) Coverage Probability (cov90): coverage probability of $c(\hat{z})$ with parentheses denoting that of $c(z^*)$; (2) Interval Width (IW); and (3) bias: Difference between $c(\hat{z})$ and the midpoint of the interval $\mathbb{E}[c(\hat{z})] - \mathbb{E}[\hat{A}_.]$.

Full experimental setup details are deferred to Appendices F.1 and F.2 with results of more $K$-fold CV results with $K = 2, 10, 20$. Moreover, we present a real-world regression dataset in Appendix F.3.

**Results.** Table 2 shows that, in terms of coverage, parametric models including ridge regression and SAA cover both $c(\hat{z})$ and $c(z^*)$ when $n$ is large, across all methods. On the other hand, nonparametric models (kNN and Forest) incur invalid coverages, due to their slow rates $\gamma < 1/4$. When using kNN with $k_n = \Theta(n^{1/4})$, only LOOCV provides valid coverage for $c(\hat{z})$ (nearly 90%) while plug-in and 5-CV fail when $n$ becomes larger. When using Forest and kNN with $k_n = \Theta(n^{2/3})$, plug-in is valid while 5-CV does not work. This matches the theoretical coverage guarantees in Table 1.

Table 2 also reports interval widths and biases to understand how some of the intervals fail. Lengths are comparable across each method, with plug-in usually shorter than LOOCV and 5-CV. This can be

attributed to that plug-in approximates $\hat{z}(\cdot)$ better while extra variability arises from the data splitting in CVs, an observation in line with those in [18]. Biases are relatively large for nonparametric models for all methods. However, when $n$ is large, both kNN with $k_n = \Theta(n^{2/3})$ and the Forest provide valid coverage for plug-in, attributed to the small bias relative to interval width, but not the case for other approaches (e.g., 5-CV for Forest).

Table 2: Evaluation performance of different methods, where boldfaced values mean **valid coverage** for $c(\hat{z})$ (i.e., within [0.85, 0.95]) and boldfaced values in parantheses mean **valid coverage** for $c(z^*)$. IW and biases for kNN and Forest in the regression problem are presented in unit $\times 10^3$. Results on other sample sizes and numerical reports on standard errors can be found in Tables 3 and 4 in Appendix F.

| method | $n$ | Plug-in | | | 5-CV | | | LOOCV | | |
|---|---|---|---|---|---|---|---|---|---|---|
| - | - | cov90 | IW | bias | cov90 | IW | bias | cov90 | IW | bias |
| **Regression Problem** ($d_x = 10, d_y = 1$) | | | | | | | | | | |
| Ridge | 1200 | 0.77 **(0.95)** | 0.16 | 0.02 | 0.55 (0.31) | 0.18 | -0.08 | 0.78 (0.90) | 0.17 | 0.00 |
| | 2400 | **0.85** (0.97) | 0.11 | 0.01 | 0.84 **(0.92)** | 0.12 | -0.02 | **0.86 (0.95)** | 0.12 | -0.00 |
| | 4800 | **0.88 (0.93)** | 0.08 | 0.00 | **0.89 (0.92)** | 0.08 | -0.01 | **0.89 (0.92)** | 0.08 | 0.00 |
| kNN $n^{2/3}$ | 2400 | 0.84 (0.00) | 1.63 | 0.08 | 0.78 (0.00) | 1.68 | -0.38 | **0.85** (0.00) | 1.64 | -0.01 |
| | 4800 | **0.87** (0.00) | 1.11 | 0.03 | 0.66 (0.00) | 1.14 | -0.37 | **0.86** (0.00) | 1.11 | -0.03 |
| | 9600 | **0.88** (0.00) | 0.75 | 0.02 | 0.61 (0.00) | 0.77 | -0.28 | **0.88** (0.00) | 0.76 | -0.01 |
| Forest | 2400 | 0.77 (0.00) | 1.77 | 0.26 | 0.66 (0.00) | 1.83 | -0.37 | 0.72 (0.00) | 1.80 | 0.01 |
| | 4800 | **0.86** (0.00) | 1.19 | 0.47 | 0.47 (0.00) | 1.24 | -0.32 | **0.85** (0.00) | 1.20 | -0.03 |
| | 9600 | **0.85** (0.00) | 0.73 | 0.11 | 0.42 (0.00) | 0.76 | -0.28 | **0.90** (0.00) | 0.75 | -0.02 |
| **CVaR-Portfolio Optimization** ($d_x = 5, d_y = 5$) | | | | | | | | | | |
| SAA | 1200 | 0.82 **(0.88)** | 0.04 | 0.00 | 0.82 **(0.87)** | 0.04 | 0.00 | **0.89 (0.88)** | 0.04 | -0.01 |
| | 2400 | **0.90 (0.89)** | 0.02 | -0.00 | **0.91 (0.89)** | 0.02 | -0.00 | **0.92 (0.89)** | 0.02 | -0.01 |
| kNN $n^{1/4}$ | 2400 | 0.00 (0.00) | 0.17 | 1.72 | 0.76 (0.00) | 0.34 | -0.08 | **0.92** (0.00) | 0.33 | -0.00 |
| | 4800 | 0.00 (0.00) | 0.12 | 1.43 | 0.72 (0.00) | 0.23 | -0.04 | **0.88** (0.00) | 0.22 | -0.00 |
| | 9600 | 0.00 (0.00) | 0.09 | 1.11 | 0.66 (0.00) | 0.15 | -0.03 | **0.89** (0.00) | 0.14 | 0.000 |

## 6  Discussions, Limitations and Future Directions

We close this paper with some guidance to practitioners in light of the results we have obtained, as well as cautionary notes on the limitations of our study and future directions.

**Guidance to Practitioners.**  In estimating out-of-sample model performances, our results suggest the following practical guidance in choosing different methods. First, in terms of the magnitude of bias, LOOCV is always smaller than plug-in, while plug-in is no larger than $K$-fold CV. Despite this bias ordering, the adoption of a method over another should also take into account the variability and computational demand, specifically:

- For parametric and nonparametric models with a fast rate ($\gamma > 1/4$, including the so-called sieve estimators in [19] when the true function $f(x)$ is $2d_x$-th continuously differentiable under $\mathcal{P}$ in (5)), the biases in all three considered procedures, plug-in, LOOCV and $K$-fold CV, are negligible compared to the variability captured in interval coverage. Correspondingly, all three intervals provide valid statistical coverages. Among them, plug-in is the most computationally efficient and should be preferred.

- For nonparametric models with a slow rate but small variability ($\gamma_v > 1/4, \gamma \le 1/4$), which include kNN with $k_n = \omega(\sqrt{n})$ in Example 3 and the forest learner in Example 4, the biases in plug-in and LOOCV are negligible but $K$-fold is not. Correspondingly, both plug-in and LOOCV provide valid coverages but $K$-fold CV does not. Since plug-in is computationally much lighter than LOOCV, it is again preferred.

- For nonparametric models with slow rate ($\gamma_v \le 1/4$), which include kNN with $k_n = \Theta(\sqrt{n})$ in Example 3, only LOOCV has a negligible bias and provides valid coverages, and hence should be adopted.

Our comparisons in model evaluation show that plug-in is preferable to $K$-fold CV, both statistically and computationally since plug-in works across a wider range of models and does not require additional model training. The above being said, we caution that, in terms of the direction of the bias, plug-in is optimistic while $K$-fold CV is pessimistic, and so the latter can be preferred if a conservative evaluation is needed to address high-stake scenarios. On the other hand, LOOCV provides valid coverages for the widest range of models, but it is computationally demanding.

Due to such computational complexity, some alternatives, including approximate leave-one-out (ALO) [12, 32], bias-corrected $K$-fold CV [30, 3] and bias-corrected plug-in [27, 35], aim to control computation load while retaining the statistical benefit of LOOCV through analytical model knowledge. For example, ALO approximates each leave-one-out solution using the so-called influence function in parametric models. However, these ALO approaches are difficult to generalize in our problem setup due to difficulties in approximating the analytical form of influence function in nonparametric models (e.g., random forest). Other variants of cross-validation, e.g., fold number increasing with $n$ [5], can potentially help improve the statistical-computational tradeoff and we leave the investigations of these procedures as future work.

**Model Evaluation versus Model Selection.**   We caution the distinction between model evaluation and selection, namely the selection of hyperparameter among a class of models. Depending on what this class is, our model evaluation comparisons may or may not translate into the performances in model selection. On a high level, this is because the evaluation bias may be correlated among different hyperparameter values and ultimately leading to a low error in the selection task. A general investigation of this issue in relation to model rates appears open, even though specific cases have been studied [5]. For example, it has been pointed out that $K$-Fold CV can perform better for ridge or lasso linear models than plug-in for model selection [43, 56, 22]. We provide further discussions on the failure cases of the plug-in procedure for model selection in Appendix G.1.

**Asymptotic versus Nonasymptotic Behaviors.**   Our results are asymptotic and there is an obvious open question on extending to finite-sample results. Nonetheless, our results still shed light on the finite-sample performances of different approaches. For example, our numerical results, which show finite-sample coverage behaviors in Figure 1 and Table 2, conform to our asymptotic theories. Note that there are some non-asymptotic intervals based on concentration inequalities with exact coverage guarantees [2, 23, 16]. However, they may be too loose as they are derived from a worst-case analysis.

**High-Dimensional Problems.**   As we mentioned in Section 2, our paper focuses on the asymptotic setting where $n \to \infty$ and $\mathcal{X}, \mathcal{Y}, \mathcal{Z}$ are of fixed dimensions. Future work includes investigating different model evaluation approaches under other regimes, including high-dimensional settings where both the dimension and sample size go to infinity. We provide some preliminary discussions of our three considered procedures in high-dimensional settings in Appendix G.2. In particular, in these situations, [9] find that standard cross-validation procedures (2) give low coverages and design a nested cross-validation procedure to remedy. However, this latter procedure is designed for high-dimensional parametric models and does not help in the slow rate regimes of nonparametric models in our setting. We provide further comparisons in Appendix G.3.

**Smoothness and Stability.**   Despite the relative generality of our scope, we have assumed sufficient model smoothness and stability. Future work includes relaxations of these conditions to broader model classes and cost functions. This is also related to the extension of our analyses to ALO approaches, as these approaches require explicit smoothness, namely gradient-type estimates on the models, as well as other advanced CV approaches in, e.g., [7, 9].

## Acknowledgments and Disclosure of Funding

The authors thank the area chair and four anonymous referees for offering many useful comments and valuable feedback. We gratefully acknowledge support from the InnoHK initiative, the Government of the HKSAR, Laboratory for AI-Powered Financial Technologies, and the Columbia Innovation Hub Award.

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

# Appendix

## A  Other Related Work and Discussions

**Plug-in Approaches in Standard Stochastic Optimization.**  In the classical stochastic optimization without covariates, the interval of optimal model performance is constructed centered at the plug-in approach set as the empirical objective solved by the sample average approximation [53]. Furthermore, to address the low coverage probability of the naive interval [44, 52, 11, 41], later literature improves the interval construction when the cost objective is nonsmooth and the variance estimate is unstable. In general, constructing the interval in the non-contextual case is generally easy compared with estimating the currentperformance of a function $\hat{z}(\cdot)$ or $z^*(\cdot)$ in the contextual stochastic optimization due to slow rates and easy violations of the asymptotic normality.

**Generability of models with $\gamma < 1/4$.**  In general, many nonparametric models converge with a rate of $\gamma \leq 1/4$. When $\ell(z; Y)$ denotes the $\ell_2$ loss, models such as sieve estimators [19] satisfy the *fast rate* $\gamma > 1/4$. However, to the best of our knowledge, these models are difficult to implement in the general contextual stochastic optimization problem arising from decision complexity. Standard benchmarks there in Assumption 2 may still suffer from $\gamma < 1/4$ (also imply from Lemma 5 as follows), surging the need for studying the evaluation approaches under these regimes.

## B  Details in Section 2.

### B.1  Technical Lemmas

We list the following technical lemmas as well as discussions on their positions in this paper.

**Lemma 3 (Standard M-estimator Result, from Theorem 5.21 in [55])** *For each $\theta$ in an open subset of Euclidean space. Let $x \mapsto m_\theta(x)$ be a measurable function such that $\theta \mapsto m_\theta(x)$ is differentiable at $\theta_0$ for almost every $x$ with derivative $\nabla_x m_{\theta_0}(x)$ and such that for every $\theta_1, \theta_2$ in a neighborhood of $\theta_0$ and a measurable function $K(x)$ with $\mathbb{E}_{\mathbb{P}^*}[K^2(x)] < \infty$:*

$$|m_{\theta_1}(x) - m_{\theta_2}(x)| \leq K(x)\|\theta_1 - \theta_2\|.$$

*Furthermore, assume that the map $\theta \mapsto \mathbb{E}_{\mathbb{P}^*}[m_\theta(x)]$ admits a second-order Taylor expansion at a point of maximum $\theta_0$. If $\hat{\theta} \xrightarrow{p} \hat{\theta}_0$, then:*

$$\sqrt{n}(\hat{\theta} - \theta_0) = -V_{\theta_0}^{-1} \frac{1}{\sqrt{n}} \sum_{i=1}^{n} \nabla_x m_{\theta_0}(X_i) + o_p(1).$$

*In particular, the sequence $\sqrt{n}(\hat{\theta} - \theta_0)$ is asymptotically normal with mean zero and covariance matrix $V_{\theta_0}^{-1} \mathbb{E}_{\mathbb{P}^*}[\nabla_x m_{\theta_0}(x) \nabla_x m_{\theta_0}(x)^\top] V_{\theta_0}^{-1}$.*

We refer readers to Theorem 5.31 in [55] under constrained cases.

Lemma 3 justifies the convergence rate of standard parametric learners. This result is the convergence in distribution for $\hat{\theta}$ in Lemma 3. Compared to the moment convergence condition we use in Definition 3, we apply the standard theory for the moment convergence of $M$-estimator ([59, 47, 20]), we can transform the convergence in distribution for $\hat{\theta}$ in Lemma 3 to the moment convergence result by: $\mathbb{E}_{\mathcal{D}_n}[n\|\hat{\theta} - \theta^*\|_2^2] < \infty$ since values of the cost function and its gradient are all bounded from Assumption 2. Therefore, in the following, we only list the result of convergence in distribution.

**Lemma 4 (Lyapunov Central Limit Theorem, extracted from Theorem 27.3 in [14])** *Suppose $\{X_1, \ldots, X_n, \ldots\}$ is a sequence of independent random variables, each with finite expected variance $\sigma_i^2$. Define $s_n^2 = \sum_{i=1}^{n} \sigma_i^2$. If for some $\delta > 0$, Lyapunov condition $\lim_{n\to\infty}(1/s_n^{2+\delta}) \sum_{i=1}^{n} \mathbb{E}[\|X_i - \mu_i\|_2^{2+\delta}] = 0$ is satisfied, then we have:*

$$\frac{1}{s_n} \sum_{i=1}^{n} (X_i - \mu_i) \xrightarrow{d} N(0, 1).$$

This Lyapunov CLT gives asymptotic normality guarantees when the variance of $X_i$ is not bounded, which happens frequently when it comes to the asymptotic expansion in nonparametric models.

**Lemma 5 (Minimax Nonparametric Lower bounds, extracted from Theorem 3.2 in [33])** *Consider    the class of distributions of $(X, Y)$ such that:*

1. $X$ is uniformly distributed in $[0,1]^{d_x}$;

2. $Y = f(X) + \epsilon$, where $\epsilon$ is the standard normal; And $X$ and $\epsilon$ are independent.

3. $f(X)$ is globally Lipschitz continuous such that for all $x_1, x_2 \in \mathbb{R}^{d_x}$, we have: $|f(x_1) - f(x_2)| \leq C\|x_1 - x_2\|$.

We call that class of distributions by $\mathcal{P}^C$, then we have:

$$\liminf_{n \to \infty} \inf_{f_n} \sup_{(X,Y) \in \mathcal{P}^C} \frac{\mathbb{E}_{\mathcal{D}_n}[\|f_n - f\|^2]}{C^{\frac{2d_x}{2+d_x}} n^{-\frac{2}{2+d_x}}} \geq C_1 > 0,$$

for some constants $C_1$ independent of $C$.

Consider $\ell(z;Y) = (z - Y)^2$. Then any estimated $\hat{z}(\cdot) = f_n(\cdot)$ from $\mathcal{D}_n$ satisfies such lower bound. Recall the bias-variance decomposition of $\|f_n(\cdot) - f\|^2$, as long as $\hat{z}(\cdot)$ converges to $z_o^*(\cdot)$, then $\gamma \leq \frac{1}{2+d_x}$, that is, either $\gamma_B \leq \frac{1}{2+d_x}$ or $\gamma_V \leq \frac{1}{2+d_x}$. This justifies the rate of $\gamma_B$ in Example 4.

## B.2 Technical Regularity Conditions

We first list our extensions of assumptions to nonsmooth and constrained problems, which are both natural in literature [58, 29]:

**Assumption 5 (Regularity of Cost Function)** *For any $y \in \mathcal{Y}$, $\ell(z;Y)$ is differentiable with respect to $z$ almost everywhere. $|\ell(z;Y)| \leq M_0$ and $\|\nabla_z \ell(z;Y)\|_2 \leq M_1$ uniformly in $z \in \mathcal{Z}$ and almost surely in $y$. Furthermore, $\ell(z;Y)$ can be written as a composite function $f(h(z;y))$, where $h(z;y) : \mathbb{R}^{d_z} \times \mathbb{R}^{d_y} \mapsto \mathbb{R}$ is twice differentiable with respect to $z$ everywhere in $z$ for any $y \in \mathcal{Y}$; $f(\cdot) : \mathbb{R} \mapsto \mathbb{R}$ has finite non-differentiable points and is twice differentiable almost everywhere.*

Note that $\ell(z;Y) = |z - Y|$ satisfy such condition.

**Assumption 6 (Optimality Conditions with Constraints)** *The decision space $\mathcal{Z}$ and the optimal solution satisfy the following:*

1. *$\mathcal{Z} \subset \mathbb{R}^{d_z}$ is open in the form $\{z \in \mathbb{R}^{d_z} : g_j(z) \leq 0, j \in J_1; g_j(z) = 0, j \in J_2\}$ with $J = J_1 \cup J_2$ and $g_j(z), \forall j \in J$ twice differentiable with respect to $z$ for any $z$. For any $x$, $\alpha_j^*(x;z)$ is twice differentiable with respect to $z$ near $z_o^*(x)$.*

2. *The KKT condition holds for the oracle problem. That is, for any given $x$, the optimal decision $z_o^*(x)(:= \operatorname{argmin}_{z \in \mathcal{Z}} v(z,x))$ exists and is unique for almost every $x$, with $z_o^*(x)$ and its Lagrange multiplier $\{\alpha_j^*(x;z)\}_{j \in J}$ satisfying the first-order condition:*

$$\nabla_z v(z_o^*(x);x) + \sum_{j \in J} \alpha_j^*(x;z_o^*(x)) \nabla_z g_j(z_o^*(x)) = 0, \forall x \in \mathcal{X};$$

   *and the complementary slackness condition $\alpha_j^*(x;z_o^*(x))g_j(z_o^*(x)) = 0, \forall j \in J, x \in \mathcal{X}$.*

3. *For any $x \in \mathcal{X}$, $\nabla_{zz}\bar{v}(z_o^*(x);x) := \nabla_{zz} v(z_o^*(x);x) + \sum_{j \in J} \alpha_j^*(x;z_o^*(x)) \nabla_{zz} g_j(z_o^*(x))$ is positive definite.*

## B.3 Examples and Justifications of Parametric and Nonparametric Models

### B.3.1 Parametric Models

We assume $G(\theta;x)$ can be any parametrized decision with respect to $\theta$ and $x$, which includes linear and more complicate models in both supervised learning, and contextual stochastic optimization problems [8, 50]:

**Assumption 7 (Additional Conditions in Assumption 1, adapted from Assumption 7 in [29])** *Suppose Assumption 6 holds. And for models in Assumption 1, for any $x \in \mathcal{X}$, $G(\theta;x)$ is twice differentiable and Lipschitz continuous with respect to $\theta$ in a neighborhood of $\theta^* := \operatorname{argmin}_{\theta \in \Theta} \mathbb{E}_{\mathbb{P}^*}[\ell(G(\theta;X);Y)]$ (exists and unique). We allow either of the following two scenarios:*

1. *When $\mathcal{Z}$ is a unconstrained set (i.e. Assumption 3), $\nabla_\theta \mathbb{E}_{\mathbb{P}_X}[v(G(\theta^*;x);x)] = 0$. And $\nabla_{\theta\theta} \mathbb{E}_{\mathbb{P}_X}[v(G(\theta;x);x)]$ is invertible for any $\theta \in \Theta$ and positive definite at $\theta = \theta^*$;*

2. When $\mathcal{Z}$ is a constrained set (i.e., Assumption 6), suppose for any $x \in \mathcal{X}$, $\alpha_j^*(\theta; x)$ is twice differentiable with respect to $\theta$ near $\theta^*$; At the point $\theta = \theta^*$, $G(\theta^*; x)$ and its Lagrange multiplier $\alpha_j^*(\theta^*; x)$ satisfy the first-order condition:

$$\nabla_\theta \mathbb{E}_{\mathbb{P}_X} v(G(\theta^*; x); x) + \sum_{j \in J} \mathbb{E}_{\mathbb{P}_X} [\alpha_j^*(\theta^*; x) g_j(G(\theta^*; x))] = 0;$$

and the complementary slackness condition $\alpha_j^*(\theta; x) g_j(G(\theta; x)) = 0, \forall j \in J, \forall \theta \in \Theta, \forall x \in \mathcal{X}$. And $\nabla_{\theta\theta} \mathbb{E}_{\mathbb{P}_X} [\bar{v}(G(\theta; x); x) (:= \nabla_{\theta\theta} v(G(\theta; x); x) + \sum_{j \in J} \alpha_j^*(x) \nabla_{\theta\theta} g_j(G(\theta; x))))$ is invertible for any $\theta \in \Theta$ and positive definite at $\theta = \theta^*$;

3. In Assumption 1, the first-order optimality condition holds for the empirical $\hat{\theta}$; And we assume $\lambda_n / n \xrightarrow{d} C$ for some $C$ as $n$ goes to infinity. $R(\theta)$ is twice continuously differentiable for any $\theta \in \Theta$ and $\nabla_{\theta\theta}^2 R(\theta)$ is uniformly bounded for any $\theta \in \Theta$.

These conditions are naturally imposed to investigate the constrained stochastic optimization problem in [25, 29]. We verify that $\gamma = 1/2$ for parametric models.

**Proposition 1 (Convergence Rate in Parametric Models)** *For parametric models in Assumption 1, we have* $\gamma = \gamma_v = 1/2$.

*Proof of Proposition 1.* For simplicity, we only consider the unregularized case. In the setup of Assumption 1, assume the following two minimization problems have unique solutions with:

$$\hat{\theta} = \underset{\theta \in \Theta}{\arg\min} \sum_{i=1}^n c(G(\theta; X_i); Y_i), \quad \theta^* = \underset{\theta \in \Theta}{\arg\min} \mathbb{E}_{\mathbb{P}_{(X,Y)}} [c(G(\theta; X); Y)].$$

Suppose Assumption 7 holds. Then combining Lemma 3 we have:

1. If $\mathcal{Z}$ is an unconstrained set, we have:

$$\sqrt{n}(\hat{\theta} - \theta^*) = -\left[\nabla_{\theta\theta} \mathbb{E}_{\mathbb{P}_{(X,Y)}} [\ell(G(\theta; X); Y)]\right]^{-1} \frac{1}{\sqrt{n}} \sum_{i=1}^n \nabla_\theta \ell(G(\theta^*; X_i), Y_i) + o_p(1);$$

2. If $\mathcal{Z}$ is an constrained set, following Corollary 1 of [25], we have:

$$\sqrt{n}(\hat{\theta} - \theta^*) = -P_T \left[\nabla_{\theta\theta} \mathbb{E}_{\mathbb{P}_{(X,Y)}} [\ell(G(\theta; X); Y)]\right]^{-1} P_T \frac{1}{\sqrt{n}} \sum_{i=1}^n \nabla_\theta \ell(G(\theta^*; X_i), Y_i) + o_p(1),$$

where $P_T = I - A^\top (AA^\top)^\dagger A$ and $A$ denotes the matrix of with rows $\mathbb{E}_{\mathbb{P}_X} [\nabla g_j(G(\theta^*; x))], j \in J$.

Regularization cases can be derived similarly (like Theorem 3 in [40]). No matter in each case, following Assumption 1 due to the bounded cost, gradient condition in Assumption 2, we have $\gamma = \frac{1}{2}$ since $\mathbb{E}[\|\hat{z}(x) - z^*(x)\|_2] = \mathbb{E}[\|G(\hat{\theta}; x) - G(\theta^*; x)\|_2] = \Theta(\|\hat{\theta} - \theta^*\|_2) = \Theta(n^{-1/2})$. And since the dominating term is just the variability term $-\left[\nabla_{\theta\theta} \mathbb{E}_{\mathbb{P}_{(X,Y)}} [\ell(G(\theta; X); Y)]\right]^{-1} \frac{1}{\sqrt{n}} \sum_{i=1}^n \nabla_\theta \ell(G(\theta^*; X_i), Y_i)$, we have $\gamma_v = 1/2$. $\square$

This $G(\theta; x)$ can be any function as long as it satisfies Assumption 7. Furthermore, $\{G\theta; x), \theta \in \Theta\}$ does not necessarily include the underlying best $z_o^*(x)$. Since the constrained expansion is similar than the unconstrained version and the moment convergence can be implied from convergence in distribution, in the following, we mainly focus on the unconstrained case of models satisfying Definitions 1 and 2.

### B.3.2 Nonparametric Models

We consider the unconstrained optimization problem under Assumption 3. Especially, we denote $\mathbb{E}[\hat{z}(x)]$ (a function over $x$) as the root of $\mathbb{E}_{\mathcal{D}_n} [\sum_{i \in [n]} w_{n,i}(x) \nabla_z \ell(z; Y_i)] = 0, \forall x$ and abbreviate it as $z_n(x)$ in the following. Recall from Assumption 2, given $\mathcal{D}_n$, we obtain the solution by:

$$\hat{z}(x) \in \underset{x \in \mathcal{X}}{\arg\min} \sum_{i \in [n]} w_{n,i}(x) \ell(z; Y_i). \tag{6}$$

**Assumption 8 (Expansion Conditions)** *Suppose the following expansion holds for the decision obtained from* (6) *for the corresponding* $w_{n,i}$:

$$\hat{z}(x) - z_n(x) = -\frac{H_{\mathcal{D}_n, x}^{-1}}{n} \sum_{i \in [n]} w_{n,i}(x) \nabla_z \ell(z_n(x); Y_i) + o_p(n^{-2\gamma_v}), \tag{7}$$

where $H_{D_n,x} = \nabla^2_{zz}\mathbb{E}_{\mathcal{D}_n}[w_{n,i}(x)\ell(z_n(x);Y)]$. *Intuitively, this result is obtained using similar routines as M-estimator theory from Lemma 3 since $\hat{z}(x) - z_n(x) = o_p(1)$ and conditions for $\hat{\theta}$ and $\theta^*$ there holds similarly for $\hat{z}(x) - z_n(x)$ here. In the following for each specific learner, we classify it through the following one of the two classes:*

$$\hat{z}(x) - z^*(x) = B_{n,x} + \frac{1}{|N(\mathcal{D}_n,x)|}\sum_{i\in N(\mathcal{D}_n,x)}\phi_x((X_i,Y_i)) + o_p(n^{-\gamma_v}), \tag{8}$$

$$\hat{z}(x) - z^*(x) = B_{n,x} + \frac{1}{n}\sum_{i=1}^{n}\phi_{n,x}((X_i,Y_i)) + o_p(n^{-\gamma_v}), \tag{9}$$

*where $B_{n,x} = \Theta(n^{-\gamma_b})$ denotes the deterministic bias term across these two classes, i.e., $z_n(x) - z^*(x)$.*

*For the variability term:*

- *In the first case (8), for almost every $x$, we have: $\mathbb{E}[\phi_x((X,Y))] = 0$, $\mathbb{E}[\|\phi_x((X,Y))\|^2_2] < \infty$, $|N(\mathcal{D}_n,x)| = \Theta(n^{2\gamma_v})$; Furthermore, for $i \in [n]$, we have: $\mathbb{E}_{X_i}[\phi_{X_i}((X_i,Y_i))] = 0$ and $\mathbb{E}[\|\phi_{X_i}((X_i,Y_i))\|^2_2] < \infty$.*

- *In the second case (9), for almost every $x$, we have: $\mathbb{E}_{\mathbb{P}_{(X,Y)}}[\phi_{n,x}((X,Y))] = 0$, $(\mathbb{E}_{\mathbb{P}_{(X,Y)}}[\|\phi_{n,x}((X,Y))\|^2_2])^{\frac{1}{2}} = \Theta(n^{\frac{1}{2}-\gamma_v}), \forall x$; Furthermore, for $i \in [n]$, we have: $\mathbb{E}_{X_i}[\phi_{n,X_i}((X_i,Y_i))] = 0$ and $(\mathbb{E}_{X_i}[\|\phi_{n,X_i}((X_i,Y_i))\|^2_2])^{\frac{1}{2}} = O(n^{1-2\gamma_v})$.*

*Therefore, $\phi_{n,x}((X_i,Y_i)) = H^{-1}_{\mathcal{D}_n,x}w_{n,i}(x)\nabla_z\ell(z_n(x);Y_i)$, where $w_{n,i}(x)$ depends on $X_i$.*

In the following, we verify a number of nonparametric models satisfying (8) or (9). Specifically, we show that Examples 3, 4 satisfy these conditions and present their convergence rates $\gamma, \gamma_b, \gamma_v$ and corresponding $\phi$ as follows.

Example 3. Recall $\hat{z}(x)$ obtained from Example 3 with a hyperparameter $k_n$, then if we choose $k_n = \min\{Cn^\delta, n-1\}$ for some $C, \delta > 0$ from Theorem 5 of [13], $\hat{z}(x)$ converges to $z^*_o(x)$.

**Proposition 2 (Convergence Rate of kNN Learner)** *In kNN learner, we denote $B_{n,x} \le C\mathbb{E}_{\tilde{x}}[\|z^*(\tilde{x}) - z^*(x)\|] = O\left(\left(\frac{k_n}{n}\right)^{\frac{1}{d_x}}\right)$, where $\tilde{x}$ is the nearest sample among $\frac{n}{k_n}$ random points near the covariate $x$ and and the order of $B_{n,x}$ can be tight with respect to $n$ from Lemma 5.*

*The variability term can be regarded as the empirical optimization over a problem where the underlying random distribution is a mixed distribution over $k_n$ conditional distribution with equal weights near the current covariate $x$. Furthermore:*

$$\hat{z}(x) - z_n(x) = -\frac{H^{-1}_{k_n,x}}{k_n}\sum_{i\in\mathcal{N}_{\mathcal{D}_n,x}(k_n)}\nabla_z\ell(z_n(x);Y_i) + o_p(k_n^{-1}), \forall x. \tag{10}$$

*where $H_{k_n,x} = \nabla^2_{zz}\mathbb{E}_{D_n(k_n,x)}[\ell(z_n(x);Y)]$ is the expectation over all the possible datasets $D_n$ and the $k_n$ neighbors around $x$. Therefore, $V_{\mathcal{D}_n,x} = O_p\left(\frac{1}{\sqrt{k_n}}\right)$ following the M-estimator theory of empirical optimization due to $k_n$ samples around. There, $\phi_x((X_i,Y_i)) = -H^{-1}_{k_n,x}\nabla_z\ell(z_n(x);Y_i)$ from (8).*

One can show that (10) is a special case of (8). In this case, the convergence rate of $\hat{z}(\cdot) - z^*(\cdot) = O_p\left(\left(\frac{k_n}{n}\right)^{1/d_x} \vee \frac{1}{\sqrt{k_n}}\right)$ with $p$ being some parameter. Tuning the best $k_n = \Theta(n^{\frac{2}{d_x+2}})$ yielding the convergence rate $\gamma = \frac{1}{d_x+2}$. This result may attain the lower bound of estimation even in the regression (i.e. Lemma 5 above) such that $\gamma = \frac{p}{d_x+2p}$. Therefore, the learner usually belongs to the slow rate regime ($\gamma < \frac{1}{4}$ when $d_x$ is large).

On the other hand, for a number of other nonparametric models, we verify the condition that (9) holds and check the corresponding rate $\gamma_v, \gamma_b$ with $c(z,Y) = (z-Y)^2$ to illustrate basic properties of the learner for readers to get familiarity, where $z^*_o(x)$ becomes $\mathbb{E}[Y|x]$ there.

Example 4. We recall existing results from [57, 6]:

**Example 5 (Convergence Rate of Random Forest Regression)** *When we have no constraints, it reduces to similar problems as in the mean estimation $z^*(x) = \mathbb{E}[Y|x]$ [57]. Then we can represent $B_{n,x} = \hat{z}(x) - \mathbb{E}[\hat{z}(x)] = O\left(n^{-\frac{\pi\beta\log(1-\omega)}{2d_x\log\omega}}\right)$ and $\phi_{n,x}((X_i,Y_i)) = s_n\left(\mathbb{E}[T|(X_i,Y_i)] - \mathbb{E}[T]\right)$ through Hajek projection*

*for some bounded random variable $T$ such that $\frac{1}{n}\phi_{n,x}((X_i,Y_i)) = O_p(n^{\frac{1-\beta}{2}})$. Here, where $s_n$ is the subsample size with $\lim_{n\to\infty} s_n = \infty$, $\lim_{n\to\infty} s_n \log^d(n)/n = 0$ by Theorem 8 in [57] and Theorem 5 in [6] for $s_n = n^\beta$, where $\omega$ means the training examples are making balanced splits in the sense that each split puts at least a fraction $\omega$ of observations in the parent node and $\pi$ means that the probability that the tree splits on the $j$-th feature is bounded from below at every split in the randomization.*

In this case, this result above gives a convergence rate $\gamma_v = \frac{1-\beta}{2}$ and $\gamma_b \geq -\frac{\pi\beta\log(1-\omega)}{2d_x\log\omega}$. Certainly, this also does not obey the lower bound result around the discussion in Lemma 5.

We demonstrate that some kernel estimators also satisfy our previous conditions:

**Example 6 (Kernel Learner)** $w_{n,i}(x) = K((X_i - x)/h_n)$ *with the hyperparameter $h_n$ and the kernel $K(\cdot)$, where $K : \mathbb{R}^{d_x} \mapsto \mathbb{R}$ with $\int K(x)dx < \infty$. Standard kernels include the naive kernel $K(x) = \mathbf{1}_{\|x\|_2 \leq r}$ and Gaussian kernel $K(x) = \exp(-\|x\|^2)$.*

Specifically, for the regression problem, we have:

**Example 7 (Nadaraya-Watson Regression [13])** *Consider the Nadaraya-Watson kernel regression $\hat{z}(x) = \sum_{j=1}^n \frac{K_h(x-X_j)}{\sum_{i=1}^n K_h(x-X_i)} Y_j$ for some $K_h(x) = K(x/h)/h$. This is the solution when $\ell(z;Y) = (z-Y)^2$. Then $B_{n,x} = z^*(x) - \mathbb{E}[\hat{z}(x)]$ and $\phi_{n,x}((X_i,Y_i)) = \frac{K_{h_n}(x-X_i)}{\mathbb{E}_\xi[K_{h_n}(x-\xi)]}(Y_i - \mathbb{E}[Y|X_i])$. If we choose $h_n = \Theta(n^{-\frac{1}{d_x+2}})$, then (9) holds for many classical kernels, e.g. Gaussian kernel $K(x) = \exp(-\|x\|^2)$ and naive kernel $K(x) = \mathbf{1}_{\{\|x\|_2 \leq r\}}$ for some constant $r$.*

**Proposition 3 (Convergence Rate of Kernel Regression)** *Suppose $X$ follows a uniform distribution $[-1,1]^{d_x}$ and $Y = f(X) + g(X)\epsilon$ for the normal $\epsilon$ with some bounded functions $f(X), g(X)$. When we take $K_h(x) = \mathbf{1}_{\{\|x\|_2 \leq h\}}$ (See more details in Chapter 5 of [33]), then $B_{n,x} = O(h_n)$ and $V_{\mathcal{D}_n,x} = O_p(n^{-1/2}h_n^{-d_x/2})$ as long as we tune $h_n$ such that $h_n \to 0$ and $nh_n^{d_x} \to \infty$ as $n \to \infty$. And $\frac{1}{n}\sum_{i=1}^n \phi_{n,x}((X_i,Y_i)) = O_p(n^{-1/2}h_n^{-d_x/2})$ with:*

$$\phi_{n,x}((X_i,Y_i)) = (Y_i - \mathbb{E}[Y|X_i])\mathbf{1}_{\{\|x-X_i\|_2 \leq h_n\}}h_n^{-d_x} = O_p(h_n^{-d_x/2})$$
$$\phi_{n,X_i}((X_i,Y_i)) = (Y_i - \mathbb{E}[Y|X_i])h_n^{-d_x} = O_p(h_n^{-d_x}).$$

In this case, the convergence rate of $\hat{z}(\cdot) - z^*(\cdot) = \left(n^{-1/2}h_n^{-d_x/2} \vee h_n\right)$. We obtain $\gamma \leq \frac{1}{d_x+2}$ when setting $h_n = \Theta(n^{-\frac{1}{d_x+2}})$. This usually belongs to the slow rate regime ($\gamma < \frac{1}{4}$).

**Remark 1 (Usage of Regularity Condition in Assumption 8)** *In the following proof, when it comes to non-parametric models, we split it into two conditions that all the nonparametric models that satisfy (8), where we replace it with (10) if needed since so far we only consider kNN learner in such category, or (9) should work.*

## B.4 Examples of Stability Conditions

**Example 8 (Expected LOO Stability of kNN Model)** *For Example 3 with $k_n = 1$, we can reparametrize the data-driven decision (i.e. $\hat{z}(x)$) by*

$$\hat{z}(x) = \sum_{j=1}^n \mathbf{1}_{x \in R_j} \operatorname*{argmin}_{z \in Z} \ell(z;Y_j),$$

*where $R_j \subseteq \mathcal{X}$ is the region (neighborhod of $X_j$) such that the closest point from $\{X_i\}_{i \in [n]}$ is $X_j$. Then since the decision is bounded $\|z\| \leq C$ from Assumption 3, we have:*

$$\mathbb{E}[(\hat{z}(X) - \hat{z}^{(-i)}(X))^2] \leq C^2(\mathbb{P}(R_j))^2 = O\left(\frac{1}{n^2}\right).$$

*Thus $\alpha_n = \frac{C}{n}$ in the 1-NN Model.*

*More generally, if $k_n = o(n)$, the stability condition there can be shown through a symmetry technique from [24] to obtain $\alpha_n = O\left(\sqrt{k_n}/n\right) = o(1/\sqrt{n})$.*

**Example 9 (Expected LOO Stability of Parametric Models)** *Suppose the objective $\ell(z;Y)$ is strongly convex and Assumptions 2 and 3 hold. Then $\alpha_n = O(1/n)$ in a standard empirical risk minimization approach from [15], which includes all the parametric models we discuss.*

# C Proofs in Section 3

The technical details of Theorem 1 and Theorem 2 have been specified in Section 4:

*Proofs of Theorem 1.* This follows from a combination of the results in Lemma 1 and Lemma 2. □

*Proofs of Theorem 2.* This follows from a combination of the results in Theorem 3 and Theorem 4. □

Before going to the detail proofs of evaluation bias and coverage guarantees for both plug-in and CV estimators, we first mention two lemmas that will be used in the following detailed results:

**Lemma 6 (Performance Gap)** *Suppose Assumptions 1 and 3 hold. For $\hat{z}(\cdot)$ in Definitions 1 and 2, we have $c(\hat{z}) - c(z^*) = O_p(n^{-2\gamma})$. And $\mathbb{E}_{\mathcal{D}_n}[c(\hat{z})] - c(z^*) = C/n^{2\gamma} + o\left(n^{-2\gamma}\right)$ for some $C > 0$.*

To show this result, we take a second-order Taylor expansion for $v(z; x)$ at $z_o^*(x)$ and notice the first-order term can be eliminated through Assumption 3 without or with constraints.

**Lemma 7 (Validity of Variability in Plug-in and Cross-Validation Approaches)** *Denote $\sigma^2 = Var_{\mathbb{P}^*}[\ell(z^*(X); Y)]$. Suppose Assumptions 1 and 3 hold. Then variance estimators $\hat{\sigma}_p^2$ in (1) satisfy $\hat{\sigma}_p^2 \xrightarrow{L_1} \sigma^2$; And $\hat{\sigma}_{kcv}^2$ in (2) satisfy $\hat{\sigma}_{kcv}^2 \xrightarrow{L_1} \sigma^2$.*

This result shows the width of each interval does not vary significantly in terms of $\Theta(n^{-1/2})$ and demonstrates the need to study the bias for each approach to distinguish between them.

*Proof of Lemma 6.* First, we consider the nonparametric models in Assumption 2 where $z^*(x) = z_o^*(x)$.

Case 1: $\mathcal{Z}$ is an unconstrained set (Assumption 3). We take the second-order Taylor expansion at the center $z^*(x)$ for the inner cost objective $v(z; x) = \mathbb{E}_{\mathbb{P}_{Y|x}}[\ell(z; Y)], \forall x$. That is,

$$
\mathbb{E}_{\mathbb{P}^*}[\ell(\hat{z}(X); Y)] - \mathbb{E}_{\mathbb{P}^*}[\ell(z^*(X); Y)]
$$
$$
= \mathbb{E}_{\mathbb{P}_X}[\nabla_z v(z^*(x); x)(\hat{z}(x) - z^*(x))] + \frac{1}{2}\mathbb{E}_{\mathbb{P}_X}[(\hat{z}(x) - z^*(x))^\top \nabla_{zz}^2 v(z; x)(\hat{z}(x) - z^*(x))] \quad (11)
$$
$$
+ o(\mathbb{E}_{\mathbb{P}_X}\|\hat{z}(x) - z^*(x)\|^2).
$$

From Assumption 3, the first-order term above becomes zero since $\nabla_z v(z^*(x); x) = 0$. And for the second-order term above, we have: $\mathbf{0} \preceq H_{xx} \preceq \lambda_U I_{d_x \times d_x} \forall x$ for some $\lambda_U > 0$ from Assumption 3. This implies that:

$$
0 \le \frac{1}{2}\mathbb{E}_{\mathbb{P}_X}[(\hat{z}(X) - z^*(X))^\top \nabla_{zz}^2 v(z; x)(\hat{z}(X) - z^*(X))] \le \frac{\lambda_U d_x}{2}\mathbb{E}_{\mathbb{P}_X}[\|\hat{z}(X) - z^*(X)\|^2].
$$

Therefore, we have $\mathbb{E}_{\mathbb{P}^*}[\ell(\hat{z}(X); Y)] - \mathbb{E}_{\mathbb{P}^*}[\ell(z^*(X); Y)] = \Theta(\|\hat{z} - z^*\|_{2,\mathbb{P}_x}^2)$.

Case 2: $\mathcal{Z}$ is a constrained set (Assumption 6). Recall $\nabla_{zz}\bar{v}(z_o^*(x); x) := \nabla_{zz}v(z_o^*(x); x) + \sum_{j \in J} \alpha_j^*(x; z_o^*(x))\nabla_{zz}g_j(z_o^*(x))$ Similarly, we take the second-order Taylor expansion at the center $z^*(x)$ for the inner cost $v(z; x)$. For each $x \in \mathcal{X}$, we obtain:

$$
v(\hat{z}(x); x) - v(z^*(x); x)
$$
$$
= \nabla_z v(z^*(x); x)(\hat{z}(x) - z^*(x)) + \frac{1}{2}(\hat{z}(x) - z^*(x))^\top \nabla_{zz} v(z^*(x); x)(\hat{z}(x) - z^*(x)) + o(\|\hat{z}(x) - z^*(x)\|^2),
$$
$$
= -\left(\sum_{j \in J}\alpha_j^*(x)\nabla_z g_j(z^*(x))\right)(\hat{z}(x) - z^*(x)) + o(\|\hat{z}(x) - z^*(x)\|^2)
$$
$$
+ \frac{1}{2}(\hat{z}(x) - z^*(x))^\top \left(\nabla_{zz}\bar{v}(z^*(x); x) - \sum_{j \in J}\alpha_j^*(x)\nabla_{zz}g_j(z^*(x))\right)(\hat{z}(x) - z^*(x)),
$$
$$(12)$$

where the second equality follows by the KKT condition from Assumption 6. And if we take the second-order Taylor expansion for $g_j(z)$ at the center $z^*(x)$ for each $j \in J$ and $x \in \mathcal{X}$, we have:

$$
g_j(\hat{z}(x)) - g_j(z^*(x)) = \nabla_z g_j(z^*(x))(\hat{z}(x) - z^*(x))
$$
$$
+ \frac{1}{2}(\hat{z}(x) - z^*(x))^\top \nabla_{zz}g_j(z^*(x))(\hat{z}(x) - z^*(x)) + o(\|\hat{z}(x) - z^*(x)\|^2) \quad (13)
$$

The left-hand side above in (13) converges to 0 since $g_j(\hat{z}(x)) - g_j(z^*(x)) \xrightarrow{P} 0, \forall j \in B$ from the first-order expansion of $g_j(z)$ and $\hat{z}(x) \xrightarrow{P} z^*(x)$ in Definition 2. Summing the terms of (13) over $j \in J$ with each weight

being $\alpha_j^*(x)$ and plugging it back into (12) to cancel $\hat{z}(x) - z^*(x)$ out in the first-order term, we have:

$$v(\hat{z}(x); x) - v(z^*(x); x) = \frac{1}{2}(\hat{z}(x) - z^*(x))^\top \nabla_{zz}\bar{v}(z^*(x); x)(\hat{z}(x) - z^*(x))$$

$$+ \sum_{j \in J} \alpha_j^*(x)(g_j(z^*(x)) - g_j(\hat{z}(x))) + o(\|\hat{z}(x) - z^*(x)\|^2)$$

$$= \frac{1}{2}(\hat{z}(x) - z^*(x))^\top \nabla_{zz}\bar{v}(z^*(x); x)(\hat{z}(x) - z^*(x)) + o(\|\hat{z}(x) - z^*(x)\|^2).$$

$$(14)$$

where the second equality above holds by (13). Then we integrate the left-hand side over $x \in \mathcal{X}$ with the underlying measure $\mathbb{P}_X$ and obtain:

$$c(\hat{z}) - c(z^*) = \frac{1}{2}\mathbb{E}_{\mathbb{P}_X}[(\hat{z}(x) - z^*(x))^\top \nabla_{zz}\bar{v}(z^*(x); x)(\hat{z}(x) - z^*(x))] + o(\|\hat{z} - z^*\|_{2,\mathbb{P}_X}^2).$$

Then we consider parametric models in Assumption 1 if we suppose Assumption 6 holds. Here, the limiting decision $z^*(x)$ is often not equal to $z_o^*(x)$. When we do not have constraints We analyze the first-order term above and have: $\mathbb{E}_{\mathbb{P}_X}[G_x(\hat{z}(x) - z^*(x)] = 0$ under Definitions 1 and 2. asp:add-param-decision

When $\mathcal{Z}$ is unconstrained, in Equation (11), the first-order optimality condition in Assumption 7 gives rise to $\mathbb{E}_{\mathbb{P}_X}[\nabla_z v(z^*(x); x)(\hat{z}(x) - z^*(x))] = 0$. And the analysis under Assumption 1 follows similarly as the case of Assumption 2. The only difference being that we directly analyze $c(\hat{z}) - c(z^*)$ over $x \in \mathcal{X}$. When $\mathcal{Z}$ is constrained, the first-order optimality condition in : $\nabla_z c(z^*) + \sum_{j \in J} \mathbb{E}_{\mathbb{P}^*}[\alpha_j^*(\theta^*; z^*(x))\nabla_z g_j(z^*(x))] = 0$ In both cases, we obtain $\mathbb{E}_{\mathbb{P}^*}[\ell(\hat{z}(X); Y)] - \mathbb{E}_{\mathbb{P}^*}[\ell(z^*(X); Y)] = \Theta(\|\hat{z} - z^*\|_{2,\mathbb{P}_x}^2)$.

Then since $\mathbb{E}_{\mathbb{P}_{Y|x}}[\|\hat{z} - z^*\|_{2,\mathbb{P}_x}^2] = \Theta(n^{-2\gamma_v})$. Then we have:

$$\mathbb{E}_{\mathcal{D}_n}[c(\hat{z})] - c(z^*) = C/n^{2\gamma} + o(n^{-2\gamma}),$$

which finishes the proof. $\square$

*Proof of Lemma 7.* We establish the $L_1$ convergence of the empirical variance term $\hat{\sigma}_p^2$ to the true variance $\text{Var}_{\mathbb{P}^*}[\ell(z^*(X); Y)]$. Applying the $L_1$-LLN to $\ell(z^*(X); y)$ since the cost function is bounded, we have:

$$\frac{1}{n}\sum_{i=1}^{n}(\ell(z^*(X_i); Y_i) - c(z^*))^2 \xrightarrow{L_1} \text{Var}_{\mathbb{P}^*}[\ell(z^*(X); Y)]$$

since $\mathcal{D}_n$ is i.i.d. and $z^*(X)$ is independent with $\mathcal{D}_n$. Therefore, we need to show that the following term converges to 0:

$$\mathbb{E}_{\mathcal{D}_n}\left[\left|\frac{1}{n}\sum_{i=1}^{n}\left[(\ell(\hat{z}(X_i); Y_i) - c_n(\hat{z}))^2 - (\ell(z^*(X_i); Y_i) - c(z^*))^2\right]\right|\right]$$

$$\leq \mathbb{E}_{\mathcal{D}_n}\left[\left|(\ell(\hat{z}(X_i); Y_i) - c_n(\hat{z}))^2 - (\ell(z^*(X_i); Y_i) - c(z^*))^2\right|\right] \quad (15)$$

$$\leq \mathbb{E}_{\mathcal{D}_n}\left[\left|\ell^2(\hat{z}(X_i); Y_i) - \ell^2(z^*(X_i); Y_i)\right|\right] + \mathbb{E}_{\mathcal{D}_n}\left[\left|c_n^2(\hat{z}) - c^2(z^*)\right|\right]$$

$$+ 2\mathbb{E}_{\mathcal{D}_n}[|\ell(\hat{z}(X_i); Y_i))c_n(\hat{z}) - \ell(z^*(X_i); Y_i)c(z^*)|].$$

For the first term on the right-hand side in (15), we have:

$$\mathbb{E}_{\mathcal{D}_n}\left[\left|\ell^2(\hat{z}(X_i); Y_i) - \ell^2(z^*(X_i); Y_i)\right|\right] \leq 2M_0\mathbb{E}_{\mathcal{D}_n}\left[\left|\ell(\hat{z}(X_i); Y_i) - \ell(z^*(X_i); Y_i)\right|\right]$$

$$\leq 2M_0 M_1 \mathbb{E}_{\mathcal{D}_n}[|\hat{z}(X_i) - z^*(X_i)|] \to 0.$$

For the second term on the right-hand side in (15), we have:

$$\mathbb{E}_{\mathcal{D}_n}\left[\left|c_n^2(\hat{z}) - c^2(z^*)\right|\right] \leq \mathbb{E}_{\mathcal{D}_n}\left[\left|c_n^2(\hat{z}) - c_n^2(z^*)\right|\right] + \mathbb{E}_{\mathcal{D}_n}\left[\left|c_n^2(z^*) - c^2(z^*)\right|\right]$$

$$\leq 2M_0 M_1 \mathbb{E}_{\mathcal{D}_n}[|\hat{z}(X_i) - z^*(X_i)|] + 2M_0\mathbb{E}_{\mathcal{D}_n}[|c_n(z^*) - c(z^*)|] \to 0.$$

For the third term on the right-hand side in (15), we have:

$$\mathbb{E}_{\mathcal{D}_n}[|\ell(\hat{z}(X_i); Y_i))c_n(\hat{z}) - \ell(z^*(X_i); Y_i)c(z^*)|]$$

$$\leq \mathbb{E}_{\mathcal{D}_n}[|c_n(\hat{z})(\ell(\hat{z}(X_i); Y_i) - \ell(z^*(X_i); Y_i))|] + \mathbb{E}_{\mathcal{D}_n}[|\ell(z^*(X_i); Y_i)(c_n(\hat{z}) - c(z^*))|]$$

$$\leq M_0 M_1 \mathbb{E}_{\mathcal{D}_n}[|\hat{z}(X_i) - z^*(X_i)|] + M_0\mathbb{E}_{\mathcal{D}_n}[|c_n(\hat{z}) - c_n(z^*)|] + M_0\mathbb{E}_{\mathcal{D}_n}[|c_n(z^*) - c(z^*)|] \to 0.$$

Therefore, we show that the variance estimator of the plug-in estimator converges.

In terms of the convergence of $\hat{\sigma}_{kcv}$, following the same routine as before, we also only need to show:

$$\mathbb{E}_{\mathcal{D}_n}\left[\left|\frac{1}{n}\sum_{k\in[K]}\sum_{i\in N_k}\left((\ell(\hat{z}^{(-N_k)}(X_i);Y_i)-\hat{A}_{kcv})^2-(\ell(z^*(X_i);Y_i)-c(z^*))^2\right)\right|\right]$$

$$\leq \frac{1}{n}\sum_{k\in[K]}\sum_{i\in N_k}\mathbb{E}_{\mathcal{D}_n}\left[\left|(c(\hat{z}^{(-N_k)}(X_i);Y_i)-\hat{A}_{kcv})^2-(c(z^*(X_i);Y_i)-c(z^*))^2\right|\right].$$

Then we can use the same error decomposition as in (15) and show that $L_1$-consistency. $\qquad\square$

# D   Proofs of Evaluation Bias in Section 4.1

## D.1   Evaluation Bias of the Plug-in Estimator

*Proof of Lemma 1.* We first consider the case where $\ell(z;Y)$ is twice differentiable with respect to $z$ for all $Y$. And the proof generalizing to the piecewise twice differentiable function satisfying Assumption 2 is the same as in the parametric setup following Theorem 3 from [36],

The bias for parametric models is provided in Theorem 1 from [36] where $\gamma=1/2$ and the bias is $\Theta(n^{-1/2})$. For nonparametric models that satisfy Definition 2, recall the definition of $z_n(x)$, and we define:

$$T_1 = \frac{1}{n}\sum_{i=1}^n \ell(\hat{z}(X_i);Y_i) - \frac{1}{n}\sum_{i=1}^n \ell(z_n(X_i);Y_i)$$

$$T_2 = \mathbb{E}[\ell(z_n(X);Y) - \ell(\hat{z}(X);Y)].$$

We expand the term inside the expectation as:

$$\mathbb{E}[\frac{1}{n}\sum_{i=1}^n \ell(\hat{z}(X_i);Y_i)] - \mathbb{E}[\ell(\hat{z}(X);Y)] = \mathbb{E}[T_1] + \mathbb{E}[T_2] + \mathbb{E}\left[\frac{1}{n}\sum_{i=1}^n \ell(z_n(X_i);Y_i) - \mathbb{E}[\ell(z_n(X);Y)]\right],$$

$$= \mathbb{E}[T_1] + \mathbb{E}[T_2],$$

Then the second equality follows by $z_n(\cdot)$ is a deterministic mapping and the observation that $\{(X_i,Y_i)\}_{i\in[n]}$ are i.i.d.

(i) Consider the nonparametric model with respect to the expansion scenario in (9). For (9), we have: $\hat{z}(x) - z_n(x) = \frac{1}{n}\sum_{i=1}^n \phi_{n,x}((X_i,Y_i)) + o_p(n^{-2\gamma_v})$. And that higher-order term $o_p(n^{-\gamma_v})$ can be ignored since we only focus on the term with the order of $o(n^{-2\gamma_v})$. More detailedly, ignoring the $o(\|\hat{z}(\cdot)-z_n(\cdot)\|_2^2)=o(n^{-2\gamma_v})$ term, we take second-order Taylor expansions to both terms $T_1$ and $T_2$.

For the term $T_1$, we have:

$$\mathbb{E}[T_1] = \mathbb{E}\left[\frac{1}{n}\sum_{i=1}^n \nabla_z\ell(z_n(X_i);Y_i)^\top(\hat{z}(X_i)-z_n(X_i))\right.$$

$$\left. + \frac{1}{2n}\sum_{i=1}^n(\hat{z}(X_i)-z_n(X_i))^\top\nabla_{zz}^2\ell(z_n(X_i);Y_i)(\hat{z}(X_j)-z_n(X_j))\right]$$

$$= \frac{1}{n}\mathbb{E}[\nabla_z\ell(z_n(X_i);Y_i)^\top\phi_{n,x}((X_i,Y_i))] + \frac{1}{2}\mathbb{E}[(\hat{z}(X_i)-z_n(X_i))^\top\nabla_{zz}^2\ell(z_n(X_i);Y_i)(\hat{z}(X_i)-z_n(X_i))],$$

where the second equality above follows by:

$$\mathbb{E}\left[\frac{1}{n}\sum_{i=1}^n \nabla_z\ell(z_n(X_i);Y_i)^\top(\hat{z}(X_i)-z_n(X_i))\right]$$

$$= \mathbb{E}\left[\frac{1}{n}\sum_{i=1}^n \nabla_z\ell(z_n(X_i);Y_i)^\top(\frac{1}{n}\sum_{i=1}^n \phi_{n,X_i}((X_i,Y_i)))\right] \tag{16}$$

$$= \frac{1}{n^2}\sum_{i=1}^n \mathbb{E}[\nabla_z\ell(z_n(X_i);Y_i)^\top\phi_{n,X_i}((X_i,Y_i))] + \frac{1}{n^2}\sum_{i\neq j}\mathbb{E}[\nabla_z\ell(z_n(X_i);Y_i)^\top\phi_{n,X_i}((X_j,Y_j))]$$

$$= -\frac{\mathbb{E}[\nabla_z\ell(z_n(X);Y)^\top\phi_{n,X}((X,Y))]}{n} + 0 = -\frac{\mathbb{E}[\nabla_z\ell(z_n(X);Y)^\top\phi_{n,X}((X,Y))]}{n},$$

where the third equality above in (16) follows by the independence of the model index under $i$ and $j$ under the conditional expectation. More specifically, $\forall i \neq j$, we have:

$$\mathbb{E}[\nabla_z \ell(z_n(X_i); Y_i)^\top \phi_{n,X_i}((X_j, Y_j))] = \mathbb{E}_{(X_i, Y_i)} \mathbb{E}_{(X_j, Y_j)}[\nabla_z \ell(z_n(X_i); Y_i)^\top \phi_{n,X_i}((X_j, Y_j))]$$

$$= \mathbb{E}_{(X_i, Y_i)}[\nabla_z \ell(z_n(X_i, Y_i))^\top \mathbb{E}_{(X_j, Y_j)}[\phi_{n,X_i}((X_j, Y_j))]] \overset{(a)}{=} 0.$$

The equality $(a)$ above follows by the fact that conditioned on any covariate $x$, $\mathbb{E}_{(X,Y)}[\phi_{n,x}((X,Y))] = 0$. This implies the last equality of (16).

For the term $T_2$, we have:

$$T_2 = \mathbb{E}[\nabla_z \ell(z_n(X); Y)^\top (\hat{z}_n(X) - z(X))] - \frac{1}{2} \mathbb{E}[(\hat{z}(X) - z_n(X))^\top \nabla^2_{zz} \ell(z_n(X); Y)(\hat{z}(X) - z_n(X))]$$

$$= 0 - \frac{1}{2} \mathbb{E}[(\hat{z}(X) - z_n(X))^\top \nabla^2_{zz} \ell(z_n(X); Y)(\hat{z}(X) - z_n(X))],$$

where the first equality of $T_2$ follows by the chain rule of the conditional expectation under the stochastic $\mathcal{D}_n$:

$$\mathbb{E}[\nabla_z \ell(z_n(X); Y)^\top (\hat{z}(X) - z_n(X))] = \mathbb{E}_{\mathbb{P}^*}[\nabla_z \ell(z_n(X); Y)^\top \mathbb{E}_{\hat{z}}[\hat{z}(X) - z_n(X)]] = 0,$$

On the other hand, we show that the second-order term of right-hand side of $\mathbb{E}[T_1]$ and $\mathbb{E}[T_2]$ is $o(n^{-2\gamma})$. We only consider the case of $d_z = 1$. This is because generalizing to the case of $d_z > 1$ only requires to sum over each elementwise component from the matrix $(\hat{z}(x) - z_n(x))(\hat{z}(x) - z_n(x))^\top$.

We first consider bounding the difference as follows:

$$\mathbb{E}_{\mathbb{P}_X, \hat{z}}[\|\hat{z}(X) - z_n(X)\|^2] - \mathbb{E}_{\mathbb{P}_X, \hat{z}}[\|\hat{z}(X_i) - z_n(X_i)\|^2]$$

$$= \mathbb{E}\left[\left\|\frac{1}{n} \sum_{i=1}^n \phi_{n,X}((X_i, Y_i))\right\|^2\right] - \mathbb{E}\left[\left\|\frac{1}{n} \sum_{j=1}^n \phi_{n,X_i}((X_j, Y_j))\right\|^2\right] + o(n^{-2\gamma_v})$$

$$= \frac{1}{n^2} \sum_{i \in [n]} \mathbb{E}_{\mathbb{P}_X, X_i}[\phi^2_{n,x}((X_i, Y_i))] - \frac{1}{n^2}\left(\sum_{j \in [n], j \neq i} \mathbb{E}_{X_i, X_j}[\phi^2_{n,X_i}((X_j, Y_j))] + \mathbb{E}_{X_i}[\phi^2_{n,X_i}((X_i, Y_i))]\right) + o(n^{-2\gamma_v})$$

$$= -\frac{1}{n^2}\left(\mathbb{E}_{X_i}[\phi^2_{n,X_i}((X_i, Y_i))] - \mathbb{E}_{\mathbb{P}_X, X_i}[\phi^2_{n,x}((X_i, Y_i))]\right) + o(n^{-1+1-2\gamma_v}) = o(n^{-2\gamma_v}),$$

(17)

where the second equality above follows by expanding $\left(\frac{1}{n} \sum_{i=1}^n \phi_{n,x}((X_i, Y_i))\right)^2$ and $\left(\frac{1}{n} \sum_{i=1}^n \phi_{n,X_i}((X_i, Y_i))\right)^2$ such that the inner product terms of $X_i$ and $X_j$ cancels out. Then we apply the result from Assumption 2 to obtain the result in (17).

Then we consider the noise difference that incorporates the Hessian matrix, that is putting $\nabla_{zz}\ell(z_n(X_i); Y_i)$ into $\hat{z}(X_i) - z_n(X_i)$ and $\nabla_{zz}[\ell(z_n(X); Y)]$ into $\hat{z}(X) - z_n(X)$. Note that if we take $\nabla_{zz}\ell(z_n(X); Y) = g((X, Y))g((X, Y))^\top$, then we can replace the original influence function $\phi_{n,X}((X, Y))$ with $\phi_{n,X}((X, Y))g((X, Y))$, which does not affect the order of the influence function with respect to $n$ since $g((X, Y))$ is of the constant level. Then it reduces to the similar analysis in (17) as the term $\mathbb{E}_{\mathbb{P}_X, \hat{z}}[\|\hat{z}(X) - z_n(X)\|^2] - \mathbb{E}_{\mathbb{P}_X, \hat{z}}[\|\hat{z}(X_i) - z_n(X_i)\|^2]$.

Therefore, based on the two results above, the numerator part of the only bias term in (16) becomes:

$$- \mathbb{E}_{\mathbb{P}_{(X,Y)}}[\nabla_z \ell(z_n(X); Y)^\top \phi_{n,X}((X, Y))]$$

$$= \mathbb{E}_{\mathbb{P}_{(X,Y)}}[\nabla_z \ell(z_n(X); Y)^\top H^{-1}_{n,X} w_{n,x}(X) \nabla_z \ell(z_n(X); Y)]$$

$$= \mathbb{E}_{\mathbb{P}_X}[w_{n,X}(X) \mathbb{E}_{\mathbb{P}_{Y|x}} \text{Tr}[H^{-1}_{n,x} \cdot \nabla_z \ell(z_n(X); Y) \nabla_z \ell(z_n(X); Y)^\top]]$$

$$= \mathbb{E}_{\mathbb{P}_{(X,Y)}}[\text{Tr}[H^{-1}_{n,X} \cdot \nabla_z \ell(z_n(X); Y) \nabla_z \ell(z_n(X); Y)^\top]] = \Theta(\mathbb{E}_{\mathbb{P}_{(X,Y)}}[w_{n,x}(X)]^{-1}) = \Theta(n^{1-2\gamma_v}).$$

where $H_x = \nabla_{zz} \mathbb{E}_{\mathbb{P}_{Y|x}}[w_{n,x}(X)\ell(z_n(x); Y)]$ and the last equation follows from (9).

(ii) Consider the nonparametric model with respect to the expansion scenario in (10). Consider the expansion of $\mathbb{E}[T_1]$ and $T_2$, following the similar proof calculations to cancel out the term $\mathbb{E}_{\mathbb{P}_X, \hat{z}}[\|\hat{z}(X) - z_n(X)\|^2] - \mathbb{E}_{\mathbb{P}_X, \hat{z}}[\|\hat{z}(X_i) - z_n(X_i)\|^2]$ and plugging in the expansion, we have:

$$\mathbb{E}[T_1 + T_2] = \frac{n}{k_n} \text{Tr}[\mathbb{E}[H^{-1}_{k_n, X_i} \ell(z_n(X_i); Y_i)\ell(z_n(X_i); Y_i)^\top]] = \Theta\left(\frac{n}{k_n}\right) = \Theta(n^{1-2\gamma_v}).$$

Therefore, in both cases, $-\frac{\mathbb{E}[\nabla_z \ell(z_n(X); Y)^\top \phi_{n,X}((X, Y))]}{n} = \Theta(n^{-2\gamma_v}) > 0$, which finishes the proof.

**Proposition 4 (Optimistic Bias for kNN)** *Suppose Assumptions 1, 2 and 3 hold. Consider kNN estimators in Example 3, then there exists one $\mathbb{P}_{(X,Y)}(n)$ for each $n$ such that the bias term $\mathbb{E}_{\mathcal{D}_n}[c_n(\hat{z}) - c(\hat{z})] = \Theta(1/k_n) = \Theta(n^{-2\gamma_v})$.*

In the proof of Proposition 4, we construct $\mathbb{P}_{(X,Y)}(n)$ as a mixture distribution where the marginal support $\mathbb{P}_X$ of each component is separate and the conditional distribution $\mathbb{P}_{Y|x}$ for $x$ within each component is the same. Then to analyze the bias, we need to focus on the bias within each component, which reduces a well-known problem to the plug-in bias for the standard stochastic optimization (e.g., [46, 36]) with a $\Theta(1/n)$ bias with $n$ samples; In our kNN case, we have $k_n$ effective samples used to obtain $\hat{z}(X_i)$ for each $i \in [n]$, therefore constituting an $\Theta(1/k_n)$ bias.

We state the following non-contextual bias lemma before diving into Proposition 4:

**Lemma 8 (Optimistic Bias in Stochastic Optimization from [46, 36])** *Suppose Assumptions 1, 2, 3 holds. Consider the sample average approximation (SAA) solution $\hat{z}(x) = \hat{z}, \forall x \in \mathcal{X}$, where:*

$$\hat{z} \in \operatorname*{argmin}_{z \in \mathcal{Z}} \frac{1}{n} \sum_{i=1}^n \ell(z; Y_i),$$

*Then: $\mathbb{E}[\frac{1}{n} \sum_{i=1}^n c(\hat{z}; Y_i)] = \mathbb{E}[c(\hat{z}; Y)] + \frac{C}{n} + o(\frac{1}{n})$ for some $C < 0$.*

*Proof of Proposition 4.* For each pair choice $n$ and $k_n$, we construct an example of $\mathbb{P}_{(X,Y)}$. W.l.o.g., we assume $k_n := \frac{n}{2k_n} \in \mathbb{Z}$ and construct the following multi-cluster distribution $\mathbb{P}_X = \sum_{i \in [k_n]} \frac{1}{k_n} \mathbf{1}_{X \in R_i} U(R_i)$, where $U(R_i)$ denotes the uniform distribution over the region $R_i$, the area of each $|R_i|$ being the same. We can partition the regions $\{R_i\}_{i \in [k_n]}$ such that $\operatorname{diam}(R_i) < d(R_i, R_j), \forall i$ and $j \neq i$. Therefore, $\mathbb{E}[\sum_{i=1}^n \mathbf{1}_{X_i \in R_j}] = 2k_n, \forall j$. And each conditional distribution $Y|X \sim \sum_{i \in [k_n]} \mathbb{P}_Y^i \mathbf{1}_{X \in R_i}$ for $k_n$ distributions $\{\mathbb{P}_Y^i, i \in [k_n]\}$.

Under the event $\mathcal{E} = \{\sum_{i=1}^n \mathbf{1}_{X_i \in R_j} \geq k_n, \forall j\}$, each in-sample decision $\hat{z}(X_i)$ only selects the data from the same region and incurs the same optimistic bias as the SAA method. Recall $\hat{z}(X_i)$ from Example 3 is equivalent to the SAA method using that $k_n$ samples since it uses the data from the same $Y$ distribution as $X_i$ conditioned on each $X_i$ (i.e. some $\mathbb{P}_Y^i$). That is $\hat{z}(X_i) \in \operatorname{argmin}_{j \in N_{i,k_n}} \ell(z; Y_j)$ for the $k_n$ nearest covariates $X$ with indices $N_{i,k_n}$, which includes $Y_i$ and other $k_n - 1$ points from the same cluster.

Recall the result from the general optimistic bias result in the uncontextual stochastic optimization (Lemma 8). Since $\hat{z}(X_i)$ only involves $k_n$ samples around $X_i$, we immediately have:

$$\mathbb{E}[\ell(\hat{z}(X_i); Y_i)|\mathcal{E}] = \mathbb{E}[\ell(\hat{z}(X); Y)|\mathcal{E}] + \Theta\left(\frac{1}{k_n}\right),$$

Then we show $\mathbb{P}(\mathcal{E}) \approx 1$ when $n$ is large, which follows from a union of $k_n$ Chernoff bounds with the multiplicative form applying to $n$ random variables $\{\mathbf{1}_{X_i \in R_j}\}_{i \in [n]} \forall j \in [k_n]$. That is:

$$\mathbb{P}(\mathcal{E}^c) = \mathbb{P}(\sum_{i=1}^n \mathbf{1}_{X_i \in R_j} < k_n, \exists j) \leq \sum_{j \in [k_n]} \mathbb{P}(\sum_{i=1}^n \mathbf{1}_{X_i \in R_j} < k_n) \leq k_n \exp(-2k_n/8),$$

which converges to 0 exponentially fast with respect to the sample size $n$ since we set $k_n = \Omega(n^\delta)$ for some $\delta > 0$. Therefore, we have:

$$\mathbb{E}[\frac{1}{n} \sum_{i=1}^n \ell(\hat{z}(X_i); Y_i)] = \mathbb{E}[\ell(\hat{z}(X_i); Y_i)]$$

$$= \mathbb{E}[\ell(\hat{z}(X_i); Y_i)|A]\mathbb{P}(A) + \mathbb{E}[\ell(\hat{z}(X_i); Y_i)|A^c]\mathbb{P}(A^c)$$

$$= \mathbb{E}[\ell(\hat{z}(X); Y)|A]\mathbb{P}(A) + o\left(\frac{C}{k_n}\right) + \mathbb{E}[\ell(\hat{z}(X); Y)|A^c]\mathbb{P}(A^c) + \mathbb{P}(A)\frac{C}{k_n}$$

$$= \mathbb{E}[\ell(\hat{z}(X); Y)] + \Theta\left(\frac{1}{k_n}\right),$$

where the third equality follows by the fact that: $|\mathbb{E}[\ell(\hat{z}(X_i); Y_i)|A^c] - \mathbb{E}[\ell(\hat{z}(X); Y)|A^c]|\mathbb{P}(A^c) = o\left(\frac{1}{k_n}\right)$ from the comparison between the exponentially tail and $k_n = \Omega(n^\delta)$. □

## D.2 Evaluation Bias of Cross-Validation Estimator

*Proof of Lemma 2.* We first consider the $K$-fold CV for any fixed $K$. Note that from Lemma 6, Following the performance gap result from Lemma 6, we have:

$$c(\hat{z}) - c(z^*) = \frac{C}{n^{2\gamma}} + o(n^{-2\gamma}), \gamma < \frac{1}{4}.$$

Since $K$-fold CV is an unbiased estimate of model performance trained with $n(1 - 1/K)$ samples, then we have:
$$\mathbb{E}[\hat{A}_{kcv}] = \mathbb{E}_{\mathcal{D}_n}[c(\hat{z}^{(-N_k)})] = c(z^*) + C/(n(1 - 1/K)^{2\gamma}) + o(n(1 - 1/K))^{-2\gamma}$$

Then comparing the two equations above, if we ignore the terms of $o(n^{-2\gamma})$ for any fixed $K$, we have:

$$\mathbb{E}[\hat{A}_{kcv} - c(\hat{z})] = \frac{C}{n^{2\gamma}(1 - \frac{1}{K})^{2\gamma}} - \frac{C}{n^{2\gamma}}$$
$$= \frac{C}{n^{2\gamma}}((1 - 1/K)^{-2\gamma} - 1)$$
$$\geq \frac{C}{n^{2\gamma}}\left(\frac{1}{1 - 2\gamma/K} - 1\right)$$
$$= \frac{C}{n^{2\gamma}} \cdot \frac{2\gamma}{K - 2\gamma}.$$

where the first inequality follows by Bernouli's inequality that $(1 + x)^r \leq 1 + rx$ for $r \in [0, 1]$ and $x \geq -1$. For the other side, we have:

$$\mathbb{E}[\hat{A}_{kcv} - c(\hat{z})] = \frac{C}{n^{2\gamma}}((1 - 1/K)^{-2\gamma} - 1) \leq \frac{C}{n^{2\gamma}}\left((1 + \frac{2}{K})^{2\gamma} - 1\right) \leq \frac{4\gamma C}{Kn^{2\gamma}},$$

where the first inequality follows from $(1 - x)^{-1} \leq 1 + 2x$ when $x \in (0, \frac{1}{2}]$ and the second inequality applies the Bernouli's inequality again. Therefore, the bias of $K$-fold CV is $\Theta(K^{-1}n^{-2\gamma})$.

Then we consider LOOCV, which can be directly calculated through the proof of Lemma 6. Suppose we are in the unconstrained case but we take the second-order Taylor expansion with Maclaurin remainder to both $c(\hat{z}^{(-i)}) - c(z^*)$ and $c(\hat{z}) - c(z^*)$. This obtains:

$$\mathbb{E}[c(\hat{z}^{(-i)}) - c(\hat{z})] = \mathbb{E}[c(\hat{z}^{(-i)})] - c(z^*) - (\mathbb{E}[c(\hat{z})] - c(z^*))$$
$$= \frac{1}{2}\mathbb{E}[(\hat{z}^{(-i)}(x) - z^*(x))^\top H_{\hat{z}^{(-i)}}(\hat{z}^{(-i)}(x) - z^*(x))] - \frac{1}{2}\mathbb{E}[(\hat{z}(x) - z^*(x))^\top H_{\hat{z}}(\hat{z}(x) - z^*(x))],$$
$$= \frac{1}{2}\mathbb{E}[(\hat{z}^{(-i)}(x) - z^*(x))^\top (H_{\hat{z}^{(-i)}} - H_{\hat{z}})(\hat{z}^{(-i)}(x) - z^*(x))]$$
$$+ \Theta(\mathbb{E}[\|\hat{z}^{(-i)}(x) - z^*(x)\|^2] - \mathbb{E}[\|\hat{z}(x) - z^*(x)\|^2]) = \Theta(n^{-1-2\gamma}) + o(1/n).$$

where $H_z = \nabla^2_{zz}c(\lambda(x)z^* + (1 - \lambda(x))z)$ as the Maclaurin remainder with $\lambda(x) \in [0, 1], \forall x$. And the last equality follows by the continuity of $H_z$ and then recall the stability condition such that $\mathbb{E}_{\mathcal{D}_n, \mathbb{P}^*}[\|\hat{z}(x) - \hat{z}^{(-i)}(x)\|] = \Theta(n^{-1})$ and apply to both terms. $\square$

# E Proofs of Variability in Section 4.2

## E.1 Variability of the Plug-in Estimator

### E.1.1 S3 is sufficient for S1

**Proposition 5 (Fast Rate Implies Stability)** *Suppose Assumptions 1, 2, 3 and 4 hold. For $\hat{z}(\cdot)$ in Assumption 2 with $\gamma_v > 1/4$, then $\beta_n = o\left(n^{-1/2}\right)$.*

*Proof of Proposition 5.* Note that $\forall i \in [n]$, we have:

$$\mathbb{E}_{\mathcal{D}_n}[\|\hat{z}(X_i) - \hat{z}^{(-i)}(X_i)\|] = \mathbb{E}_{\mathcal{D}_n}[\|(\mathbb{E}_{\mathcal{D}_n}[\hat{z}(X_i)] - \mathbb{E}_{\mathcal{D}_n}[\hat{z}^{(-i)}(X_i)])$$
$$+ (\hat{z}(X_i) - \mathbb{E}[\hat{z}(X_i)]) - (\hat{z}^{(-i)}(X_i) - \mathbb{E}[\hat{z}^{(-i)}(X_i)])\|]$$
$$\leq (\mathbb{E}_{\mathcal{D}_n}[\hat{z}(X_i)] - \mathbb{E}_{\mathcal{D}_n}[\hat{z}^{(-i)}(X_i)])$$
$$+ \mathbb{E}_{\mathcal{D}_n}[\|(\hat{z}(X_i) - \mathbb{E}[\hat{z}(X_i)]) - (\hat{z}^{(-i)}(X_i) - \mathbb{E}[\hat{z}^{(-i)}(X_i)])\|],$$

where the first part denotes the bias difference and the second part denotes the variance difference. Then we need to show the two differences term are of order $o(n^{-1/2})$. We divide them into two lemmas (Lemma 9 and Lemma 10). Then the result holds. $\square$

**Lemma 9 (Bias Stability)** *When $\alpha_n = o(n^{-1/2})$ and $\gamma_v > \frac{1}{4}$, $\mathbb{E}_{\mathcal{D}_n}[\hat{z}(X_i)] - \mathbb{E}_{\mathcal{D}_n}[\hat{z}^{(-i)}(X_i)] = o(n^{-1/2})$ for $\hat{z}(\cdot)$ in Assumption 2.*

*Proof of Lemma 9.* First, notice that: $\mathbb{E}_{\mathcal{D}_n}[\hat{z}^{(-i)}(X_i)] = \mathbb{E}_{\mathbb{P}_X,\mathcal{D}_n}[\hat{z}^{(-i)}(x)]$ since $X_i$ and $\hat{z}^{(-i)}$ are independent. And we have $\mathbb{E}[\hat{z}(x) - \hat{z}^{(-i)}(x)] = o(n^{-1/2})$ from $\alpha_n = o(n^{-1/2})$. Therefore, we only need to show:

$$\mathbb{E}_{\mathbb{P}_X,\hat{z}}[\hat{z}(X_i) - \hat{z}(x)] = o(n^{-1/2}).$$

In the following two cases, we decompose the influence expansion for the two estimators.

When $\hat{z}(\cdot)$ satisfies (8) (or (10) specially): Note that $\gamma_v > \frac{1}{4}$ is equivalent to saying $k_n = w(n^{1/2})$. From (10), $\forall i \in [n]$, we have:

$$\mathbb{E}[\hat{z}(x)] - \mathbb{E}_{P_X}[z_n(x)] = -\mathbb{E}\left[\frac{H_{k_n,x}^{-1}}{k_n} \sum_{i \in \mathcal{N}_{\mathcal{D}_n,x}(k_n)} \nabla_z \ell(z_n(x); Y_i)\right] + o(k_n^{-1})$$

$$= -\mathbb{E}_{\mathbb{P}_X}\left[\frac{H_{k_n,x}^{-1}}{k_n} \mathbb{E}_{\mathcal{D}_n}\left[\sum_{i \in \mathcal{N}_{\mathcal{D}_n,x}(k_n)} \nabla_z \ell(z_n(x); Y_i)\right]\right] + o(k_n^{-1}) = o(k_n^{-1}),$$

where the second equality follows by the definition of $z_n(x)$ such that $\mathbb{E}_{\mathcal{D}_n}[\sum_{i \in \mathcal{N}_{\mathcal{D}_n,x}(k_n)} \nabla_z \ell(z_n(x); Y_i)] = 0$. In contrast, ignoring the term $o(k_n^{-1})$ in the derivation, we have:

$$\mathbb{E}[\hat{z}(X_i)] - \mathbb{E}_{P_X}[z_n(X_i)] = -\mathbb{E}\left[\frac{H_{k_n,X_i}^{-1}}{k_n} \sum_{j \in \mathcal{N}_{\mathcal{D}_n,X_i}(k_n)} \nabla_z \ell(z_n(X_j); Y_j)\right] + o(k_n^{-1})$$

$$= -\mathbb{E}\left[\frac{H_{k_n,X_i}^{-1}}{k_n} \sum_{j \in \mathcal{N}_{\mathcal{D}_n,X_i}(k_n),j \neq i} \nabla_z \ell(z_n(X_j); Y_j)\right] - \frac{1}{k_n}\mathbb{E}\left[H_{k_n,X_i}^{-1} \nabla_z \ell(z_n(X_i); Y_i)\right]$$

$$= 0 - \frac{1}{k_n}\mathbb{E}\left[H_{k_n,X_i}^{-1} \nabla_z \ell(z_n(X_i); Y_i)\right]$$

$$= -\frac{1}{k_n}\mathbb{E}\left[H_{k_n,X_i}^{-1} \nabla_z \ell(z_n(X_i); Y_i)\right]$$

where the third equality follows by $X_i$ and $\{X_j\}_{j \in [n], j \neq i}$ are independent and using the same argument as the previous ones $\mathbb{E}_{\mathcal{D}_n}[\sum_{i \in \mathcal{N}_{\mathcal{D}_n,x}(k_n)} \nabla_z \ell(z_n(x); Y_i)] = 0$. Therefore, we have:

$$\mathbb{E}[\hat{z}(X_i) - \hat{z}(x)] = -\frac{1}{k_n}\mathbb{E}_{\mathcal{D}_n}\left[H_{k_n,X_i}^{-1} \nabla_z \ell(z_n(X_i); Y_i)\right] + o(k_n) = \Theta(k_n) = o(n^{-1/2}).$$

When $\hat{z}(\cdot)$ satisfies (9) or (7), following similar results as in kNN models, $\mathbb{E}[\hat{z}(x)] - \mathbb{E}[z_n(x)] = o(n^{-2\gamma_v}) = o(n^{-1/2})$. Besides, we have:

$$\mathbb{E}[\hat{z}(X_i)] - \mathbb{E}_{P_X}[z_n(X_i)] = -\frac{1}{n}\mathbb{E}[H_{\mathcal{D}_n,X_i}^{-1} w_n(X_i, X_i) \nabla_z \ell(z_n(X_i); Y_i)]$$

$$= -\frac{1}{n}\mathbb{E}[H_{\mathcal{D}_n,X_i}^{-1} \nabla_z \ell(z_n(X_i); Y_i)] = o(n^{-2\gamma_v}) = o(n^{-1/2}).$$

**Lemma 10 (Variance Stability)** *When* $\alpha_n = o(n^{-1/2})$ *and* $\gamma_v > \frac{1}{4}$, $\mathbb{E}_{\mathcal{D}_n}[|(\hat{z}(X_i) - \mathbb{E}[\hat{z}(X_i)]) - (\hat{z}^{(-i)}(X_i) - \mathbb{E}[\hat{z}^{(-i)}(X_i)])|] = o(n^{-1/2})$ *for* $\hat{z}(\cdot)$ *in Assumption 2.*

*Proof of Lemma 10.* When $\hat{z}(\cdot)$ satisfies (8) (or (10) specifically): Ignoring the $o(k_n^{-1})$ terms across equalities, we have:

$$(\hat{z}(X_i) - \mathbb{E}[\hat{z}(X_i)]) - (\hat{z}^{(-i)}(X_i) - \mathbb{E}[\hat{z}^{(-i)}(X_i)])$$

$$= -\frac{H_{k_n,X_i}^{-1}}{k_n} \sum_{j \in \mathcal{N}_{\mathcal{D}_n,X_i}(k_n)} \nabla_z \ell(z_n(X_j); Y_j) + \frac{H_{k_n-1,X_i}^{-1}}{k_n-1} \sum_{j \in \mathcal{N}_{\mathcal{D}_{n-1},X_i}(k_{n-1})} \nabla_z \ell(z_{n-1}(X_j); Y_j)$$

$$= -\frac{H_{k_n,X_i}^{-1}}{k_n} \nabla_z \ell(z_n(X_i); Y_i) + \left(\frac{H_{k_n-1,X_i}^{-1}}{k_n-1} - \frac{H_{k_n,X_i}^{-1}}{k_n}\right) \sum_{j \in \mathcal{N}_{\mathcal{D}_n,X_i}(k_n),j \neq i} (\nabla_z \ell(z_n(X_j); Y_j) - \nabla_z \ell(z_{n-1}(X_j); Y_j)).$$

We take the expectation to the two terms on the right-hand side. For the first term, we have:

$$\mathbb{E}\left[|\frac{H_{k_n,X_i}^{-1}}{k_n} \nabla_z \ell(z_n(X_i); Y_i)|\right] \leq \frac{\mathbb{E}_{X_i}[H_{k_n,X_i}^{-1} \nabla_z \ell(z_n(X_i); Y_i)]}{k_n} = \Theta(k_n^{-1}).$$

For the second term, we have:

$$\mathbb{E}\left[\left\|\left(\frac{H^{-1}_{k_{n-1},X_i}}{k_{n-1}} - \frac{H^{-1}_{k_n,X_i}}{k_n}\right)\sum_{j\in\mathcal{N}_{\mathcal{D}_n,X_i}(k_n),j\neq i}(\nabla_z\ell(z_n(X_j);Y_j) - \nabla_z\ell(z_{n-1}(X_j);Y_j))\right\|\right]$$

$$= \mathbb{E}_{X_i}\left[\left\|\left(\frac{H^{-1}_{k_{n-1},X_i}}{k_{n-1}} - \frac{H^{-1}_{k_n,X_i}}{k_n}\right)\right\|\right]\mathbb{E}_{\mathcal{D}_n\setminus X_i}\left[\left\|\sum_{j\in\in\mathcal{N}_{\mathcal{D}_n,X_i}(k_n),j\neq i}(\nabla_z\ell(z_n(X_j);Y_j) - \nabla_z\ell(z_{n-1}(X_j);Y_j))\right\|\right]$$

where the first equality follows by $X_i$ and $\{X_j\}_{j\neq i}$ are independent. On one hand, we know $H^{-1}_{k_n-1,X_i} = H_{k_n,X_i} + o(1)$, which implies that: $\mathbb{E}_{X_i}\left[\left\|\left(\frac{H^{-1}_{k_{n-1},X_i}}{k_{n-1}} - \frac{H^{-1}_{k_n,X_i}}{k_n}\right)\right\|\right] = o(k_n^{-1})$. On the other hand:

$$E_{\mathcal{D}_n\setminus X_i}\left[\left\|\sum_{j\in\in\mathcal{N}_{\mathcal{D}_n,X_i}(k_n),j\neq i}(\nabla_z\ell(z_n(X_j);Y_j) - \nabla_z\ell(z_{n-1}(X_j);Y_j))\right\|\right]$$

$$\leq k_n L\mathbb{E}_{\mathcal{D}_n\setminus X_i}[\|z_n(X_i) - z_{n-1}(X)\|] = o(k_n n^{-1/2}).$$

Combining these two arguments, we have: $\mathbb{E}_{\mathcal{D}_n}[\|(\hat{z}(X_i) - \mathbb{E}[\hat{z}(X_i)]) - (\hat{z}^{(-i)}(X_i) - \mathbb{E}[\hat{z}^{(-i)}(X_i)])\|] = \Theta(k_n^{-1}) + o(n^{-1/2}) = o(n^{-1/2})$ for kNN models.

When $\hat{z}(\cdot)$ satisfies (9) or (7), we use the notation in Assumption 2. Then variability difference becomes:

$$\mathbb{E}_{\mathcal{D}_n}[\|(\hat{z}(X_i) - \mathbb{E}[\hat{z}(X_i)]) - (\hat{z}^{(-i)}(X_i) - \mathbb{E}[\hat{z}^{(-i)}(X_i)])\|]$$

$$= \mathbb{E}\left[\left\|\frac{1}{n}\phi_{n,X_i}((X_i,Y_i)) + \sum_{j\neq i}\left(\frac{\phi_{n,X_i}}{n} - \frac{\phi_{n-1,X_i}}{n-1}\right)((X_j,Y_j))\right\|\right]$$

$$\leq \frac{\mathbb{E}[\|\phi_{n,X_i}((X_i,Y_i))\|]}{n} + \frac{1}{n-1}\mathbb{E}\left[\left\|\sum_{j\neq i}(\phi_{n,X_i} - \phi_{n-1,X_i})((X_j,Y_j))\right\|\right]$$

$$+ \frac{1}{n(n-1)}\sum_{j\neq i}\mathbb{E}[\|\phi_{n,X_i}((X_j,Y_j))\|]$$

$$= O(n^{-2\gamma_v}) + O(n^{-\frac{1}{2}}\cdot n^{-\gamma_v}) + O(n^{-1}\cdot n^{1/2-\gamma_v}) = O(n^{-2\gamma_v}).$$

where the second equality follows by the fact that $\mathbb{E}[\|\phi_{n,x}((X,Y))\|] = O(n^{1-2\gamma_v})$ from Assumption 8; the second term $\{(\phi_{n,x} - \phi_{n-1,x}((X_j,Y_j))|\}_{j\in[n],j\neq i}$ are $n - 1$ i.i.d. random variables with each mean 0 and variance $\mathbb{E}[\|(\phi_{n,x} - \phi_{n-1,x})((X_j,Y_j))|^2] < \infty$. Then the central limit theorem and uniform convergence imply: $\frac{1}{n-1}\mathbb{E}\left[\left\|\sum_{j\neq i}(\phi_{n,X_i} - \phi_{n-1,X_i})((X_j,Y_j))\right\|\right] = O(n^{-1/2})$. This finishes our proof for the variability stability for nonparametric models. $\qquad\square$

### E.1.2 S3 is necessary for S2

**Proposition 6 (Optimistic Bias Implies Coverage Invalidity)** *Suppose Assumptions 1, 2 and 3 hold. When* $\mathbb{E}_{\mathcal{D}_n}[B] = \mathbb{E}_{\mathcal{D}_n}[c_n(\hat{z}) - c(\hat{z})] = \Omega(n^{-1/2})$, S2 *does not hold.*

*Proof of Proposition 6.* When the evaluation bias $\mathbb{E}[B_n] = \Omega(n^{-\frac{1}{2}})$, we show that we cannot have: $\sqrt{n}B \xrightarrow{d} N(0,\sigma^2)$.

Recall from the proof of Lemma 1, we have: $B = T_1+T_2+c_n(z_n)-c(z_n)$. We can verify that $\text{Var}_{\mathcal{D}_n}[T_1+T_2] = o(n^{-2\gamma_v})$ through the same analysis as in Lemma 1. Therefore, we have: $n^{2\gamma}(T_1 + T_2) \xrightarrow{d} n^{2\gamma}\mathbb{E}[T_1 + T_2] = \Theta(1) > 0$ by Chebyshev inequality. Denote the right-hand side of the convergence limit as $C > 0$. Besides $\sqrt{n}(c_n(z_n) - c(z_n)) \xrightarrow{d} N(0,\sigma^2)$ from CLT and $z_n(\cdot) \xrightarrow{p} z^*(\cdot)$. Therefore, if $\gamma = 1/4$, we have: $\sqrt{n}B \xrightarrow{d} N(C,\sigma^2)$; if $\gamma < 1/4$, we have: $n^{2\gamma}B \xrightarrow{d} C$. In both cases, $I_p$ cannot provide a valid coverage guarantee for $c(\hat{z})$. $\qquad\square$

### E.1.3 S1 is sufficient for S2

**Proposition 7 (Stability Implies Coverage)** *Suppose Assumptions 1, 3 and 4 hold. Then* S1 *implies* S2.

**Lemma 11 (CLT for Plug-in Estimator)** *Suppose Assumptions 1, 2, 3 and 4 hold. If $\beta_n = o(n^{-1/2})$, then the plug-in estimator $\hat{A}_p$ in (1) satisfies $\sqrt{n}(\hat{A}_p - c(\hat{z})) \xrightarrow{d} N(0, \sigma^2)$.*

*Proof of Proposition 7.* This result directly follows by the combination of Lemma 11 and Lemma 7. $\qquad\square$.

The proof of Lemma 11 follows from verifying the stochastic equincontinuity condition of the plug-in estimator as follows:

**Lemma 12 (General Result for Stochastic Equicontinuity of Plug-in Estimator, Lemma 2 of [18])** *For the estimator $\hat{g} \in \mathcal{G}$ which is function of $v$ that is estimated through $\{V_i\}_{i \in [n]}$ and $\hat{g}^{(-i)}$ is estimated through $\{V_j\}_{j \in [n]} \backslash \{V_i\}$ and there exists some $g_0$ such that $\hat{g} \xrightarrow{L_2} g_0$, denote $A(V; g) = \mathbb{E}_v[a(v; g)]$ and $A_n(V; g) = \frac{1}{n} \sum_{i=1}^n a(V_i; g)$. Suppose we have the following condition:*

$$\mathbb{E}_{\hat{g}, \hat{g}^{(-i)}}[|a(V_i, \hat{g}) - a(V_i, \hat{g}^{(-i)})|] = o(n^{-1/2}), \forall i \in [n];$$

$$\mathbb{E}_{\hat{g}, \hat{g}^{(-i)}, V}[|a(V, \hat{g}) - a(V, \hat{g}^{(-i)})|^2] = o(n^{-1}), \forall i \in [n]$$

*and for $g_1, g_2 \in \mathcal{G}$, $\mathbb{E}[(a(V; g_1) - a(V; g_2))^2] \leq L\mathbb{E}_v[\|g_1 - g_2\|_2^q]$ for some $q < \infty$. Then $\sqrt{n}((A(\hat{g}) - A(g_0)) - (\hat{A}_n(\hat{g}) - \hat{A}_n(g_0))) = o_p(1)$.*

*Proof of Lemma 11.* This result follows by considering $c(\hat{z})$ and $c(z^*)$ as the empirical and population version of the objective, i.e.,terms $A_n(\cdot)$ and $A(\cdot)$ there in Lemma 12. Compared with our notion, we need to verify the following three conditions:

- $\mathbb{E}_{\hat{z}, \hat{z}^{(-i)}} \mathbb{E}[|\ell(\hat{z}(X_i); Y_i) - \ell(\hat{z}^{(-i)}(X_i); Y_i)|] = o(n^{-1/2})$;

- $\mathbb{E}_{\hat{z}, \hat{z}^{(-i)}} \mathbb{E}[|\ell(\hat{z}(X); Y) - \ell(\hat{z}^{(-i)}(X); Y)|^2] = o(n^{-1})$;

- For any two measurable function $z_1(X), z_2(X)$, we have:
$$\mathbb{E}_{\mathbb{P}^*}[(\ell(z_1(X); Y) - \ell(z_2(X); Y))^2] \leq L(\mathbb{E}_{\mathbb{P}_X}[\|z_1(X) - z_2(X)\|])^q,$$
  for some $L > 0$ and $q < \infty$.

The first two results directly follow by the stability condition that $\alpha_n, \beta_n = o\left(n^{-1/2}\right)$ and the conditions in Assumption 2 that $\nabla_z \ell(z; Y)$ is bounded. And the third condition is verified through the first-order expansion that:

$$\begin{aligned}
\mathbb{E}_{\mathbb{P}_{(X,Y)}}[(\ell(z_1(X); Y) - \ell(z_2(X); Y))^2] &= \mathbb{E}_{\mathbb{P}_{X,Y}}[((\nabla_z \ell(\tilde{z}(X); Y))^\top (z_1(X) - z_2(X)))^2] \\
&\leq (\mathbb{E}_{\mathbb{P}_X}[\mathbb{E}_{\mathbb{P}_{Y|X}} \|\nabla_z \ell(\tilde{z}(X); Y)\|^2])(\mathbb{E}_{\mathbb{P}_X}[\|z_1(X) - z_2(X)\|^2]) \\
&\leq M_1^2 (\mathbb{E}_{\mathbb{P}_X}[\|z_1(X) - z_2(X)\|^2]),
\end{aligned}$$

where we take $L \geq \mathbb{E}_{\mathbb{P}_X}[\mathbb{E}_{\mathbb{P}_{Y|x}} \|\nabla_z \ell(\tilde{z}(X); Y)\|^2]$ and $q = 2$ such that these conditions hold. $\qquad\square$

### E.1.4 Other Details

**Corollary 1 (Coverage of Plug-in Estimator's Interval)** *Suppose Assumptions 1, 2, 3,and 4 hold. If $\gamma > 1/4$, $I_p$ provides a valid coverage guarantee for $c(\hat{z}), c(z^*), \mathbb{E}_{\mathcal{D}_n}[c(\hat{z})]$.*

For parametric models under Assumption 1, we have $\gamma = 1/2 > 1/4$, and the expected LOO stability notion is satisfied. Therefore, Corollary 1 always holds for parametric models.

We first state the following result based on the standard Slutsky's theorem:

**Lemma 13 (Bias and CLT)** *For a random sequence $A_n$ satisfying $\sqrt{n}(A_n - A) \xrightarrow{d} N(0, \sigma^2)$, if another random term $B = A + o_p(n^{-1/2})$, then $\sqrt{n}(A_n - B) \xrightarrow{d} N(0, \sigma^2)$.*

*Proof of Corollary 1.* Note that if $\gamma > 1/4$, we have: $\sqrt{n}(c_n(\hat{z}) - c(\hat{z})) \xrightarrow{d} N(0, \hat{\sigma}^2)$ from Lemma 11. Combining it with Lemma 7, it is easy to see that $I_p$ provides a valid coverage guarantee for $c(\hat{z})$.

Furthermore, consider $A_n = c_n(\hat{z})$ and $A = c(\hat{z})$. And if we set $B = \mathbb{E}_{\mathcal{D}_n}[c(\hat{z})]$ and $c(z^*)$ respectively, then we have: $B = A + o_p(n^{-1/2})$ from Lemma 6. Then applying Lemma 13, we stil have:

$$\sqrt{n}(c_n(\hat{z}) - c(\hat{z})) \xrightarrow{d} N(0, \sigma^2)$$

$$\sqrt{n}(c_n(\hat{z}) - \mathbb{E}_{\mathcal{D}_n}[c(\hat{z})]) \xrightarrow{d} N(0, \sigma^2).$$

Therefore, $I_p$ provides a valid coverage guarantee for $c(z^*)$ and $\mathbb{E}_{\mathcal{D}_n}[c(\hat{z})]$. $\qquad\square$

## E.2 Variability of Cross-Validation Estimator

Before the proofs of the equivalence condition of cross-validation estimator (i.e. Theorem 4), we list the CLT for cross-validation.

**Lemma 14 (CLT of Cross-Validation)** *Suppose Assumptions 1, 2, 3 and 4 hold. Then $\hat{A}_{kcv}$ in (2) satisfies:* $\sqrt{n}(\hat{A}_{kcv} - \tilde{A}) \xrightarrow{d} N(0, \sigma^2)$ *with* $\tilde{A} = \sum_{k \in [K]} \mathbb{E}_{\mathbb{P}^*}[\ell(\hat{z}^{(-N_k)}(X); Y)]/K$. *Here the same result holds for* $\hat{A}_{loocv}$ *by setting* $K = n$.

Note that the asymptotic normality of cross-validation does not depend on $\gamma$. However, the center from the CLT in Lemma 14 is different from $c(\hat{z})$ or $c(z^*)$, and the convergence rate $\gamma$ determines whether the difference is small. Therefore, the validity of the interval for $K$-fold CV to cover $c(\hat{z})$ still depends on $\gamma$.

Before moving to prove Lemma 14, we introduce the following stability from the conditions in [10]. We will show that this can be directly implied from our assumptions of the cost function and the expected LOO stability:

**Definition 5 (Loss Stability)** *Continuing from the setup in the main body with $\mathcal{D}_n$, we call the loss stability [39] by:*

$$\alpha_n^{ls} := \frac{1}{n} \sum_{i=1}^{n} \mathbb{E}_{\mathbb{P}_{(X,Y)}, \mathcal{D}_n}[(\bar{\ell}(\hat{z}(X); Y) - \bar{\ell}(\hat{z}^{(-i)}(X); Y))^2],$$

*where* $\bar{\ell}(\hat{z}(X); Y) = \ell(\hat{z}(X); Y) - \mathbb{E}_{\hat{z}}[\ell(\hat{z}(X); Y)]$.

We rewrite Theorem 1 in [10] and replace the uniformly integrable condition with our assumptions (since the bounded cost function condition in Assumption 2 directly implies the uniformly integrable condition there):

**Lemma 15 (CV of CLT from Theorem 1 in [10])** *Suppose Assumptions 1, 2, 3 and 4 hold. If $\alpha_n^{ls} = o(1/n)$. Then we have:*

$$\sqrt{n}(\hat{A}_{kcv} - \tilde{A}) \xrightarrow{d} N(0, \sigma^2).$$

*Proof of Lemma 14.* To show this result, we only need to verify the loss stability is $o(1/n)$ above. This follows by:

$$(\ell(\hat{z}(X); Y) - \mathbb{E}_{\hat{z}}[\ell(\hat{z}(X); Y)])^2 - (\ell(\hat{z}^{(-i)}(X); Y) - \mathbb{E}_{\hat{z}^{(-i)}}[\ell(\hat{z}^{(-i)}(X); Y)])^2$$
$$\leq 2(\ell(\hat{z}(X); Y) - \ell(\hat{z}^{(-i)}(X); Y)))^2 + 2(\mathbb{E}_{\hat{z}}[\ell(\hat{z}(X); Y)] - \mathbb{E}_{\hat{z}^{(-i)}}[\ell(\hat{z}^{(-i)}(X); Y)])^2.$$

For the first term of the right-hand side above, due to the bounded gradient of $\ell$, following the first-order Taylor expansion inside the square notation, we have:

$$(\ell(\hat{z}(X); Y) - \ell(\hat{z}^{(-i)}(X); Y)))^2 \leq M_1^2(\hat{z}(X) - \hat{z}^{(-i)}(X))^2.$$

Then if we take expectation over $X$, the right-hand side reduces to $M_1^2 \alpha_n^2 = o(1/n)$.

For the second term of the right-hand side above, we have:

$$(\mathbb{E}_{\hat{z}}[\ell(\hat{z}(X); Y)] - \mathbb{E}_{\hat{z}^{(-i)}}[\ell(\hat{z}^{(-i)}(X); Y)])^2$$
$$\leq \mathbb{E}_{\hat{z}, \hat{z}^{(-i)}}[(\ell(\hat{z}(X); Y) - \ell(\hat{z}^{(-i)}(X); Y))^2] \leq M_1^2 \mathbb{E}_{\hat{z}, \hat{z}^{(-i)}}[(\hat{z}(X) - \hat{z}(X))^2].$$

Then if we take expectation over $X$, the right-hand side reduces to $M_1^2 \alpha_n^2 = o(1/n)$. Therefore, we can apply Lemma 15 to see that CLT for cross-validation holds. $\square$

*Proof of Theorem 4.* We first consider the coverage validity of $c(\hat{z})$.

(i) LOOCV. Since Lemma 14 holds, in this case, we only need to show that the term:

$$\frac{1}{n} \sum_{i=1}^{n} \mathbb{E}_{\mathbb{P}^*}[\ell(\hat{z}^{(-i)}(X); Y)] - \mathbb{E}_{\mathbb{P}^*}[\ell(\hat{z}(X); Y)] = o_p(n^{-\frac{1}{2}}). \tag{18}$$

Then combining this with the Lemma 13, we obtain the valid coverage of LOOCV intervals for the $c(\hat{z})$. From the first-order Taylor expansion to $\mathbb{E}_{\mathbb{P}_{Y|x}}[\ell(z; Y)]$, we have:

$$\frac{1}{n} \sum_{i=1}^{n} \mathbb{E}_{\mathbb{P}^*}[\ell(\hat{z}^{(-i)}(X); Y)] - \mathbb{E}_{\mathbb{P}^*}[\ell(\hat{z}(X); Y)]$$
$$= \frac{1}{n} \sum_{i=1}^{n} \mathbb{E}_{\mathbb{P}_X}[\nabla_z \mathbb{E}_{\mathbb{P}_{Y|x}} \ell(\hat{z}(X); Y)(\hat{z}^{(-i)}(X) - \hat{z}(X))] + o_p\left(|\hat{z}^{(-i)}(X) - \hat{z}(X)|\right), \tag{19}$$

Since the gradient of the cost function $\nabla_z \ell(z; Y)$ is bounded and following the stability condition:

$$\mathbb{E}_{\mathbb{P}_X}[|\hat{z}^{(-i)}(X) - \hat{z}(X)|] = o\left(n^{-1/2}\right).$$

Therefore, plugging it back above into (19), we immediately see that $\frac{1}{n}\sum_{i=1}^{n}\mathbb{E}_{\mathbb{P}^*}[c(\hat{z}^{(-i)}(X); Y)] - \mathbb{E}_{\mathbb{P}^*}[\ell(\hat{z}(X); Y)] = o_p\left(n^{-1/2}\right)$ there. Then applying Lemma 13 and Lemma 7 would obtain the result of $\lim_{n\to\infty}\mathbb{P}(c(\hat{z}) \in I_{loocv}) = 1 - \alpha$.

(ii) $K$-fold CV. For the part that S4 implies S5: if $\gamma > 1/4$, then we can show $\tilde{A} - c(\hat{z}) = o_p(n^{-1/2})$ too by taking the Taylor expansion for each $c(\hat{z}^{(-N_k)}) - c(z^*), \forall k \in [K]$ and $c(\hat{z}) - c(z^*)$. Then both terms are only $o_p(n^{-1/2})$. Then using Lemma 13 with $\sqrt{n}(\hat{A}_{kcv} - \tilde{A}) \overset{d}{\to} N(0, \sigma^2)$, we have: $\sqrt{n}(\hat{A}_{kcv} - c(\hat{z})) \overset{d}{\to} N(0, \sigma^2)$. Combining this with Lemma 7 obtains S5.

For the part that S5 implies S4, we prove by contradiction. Note that if $\gamma \leq 1/4$, we have: $n^{2\gamma}(\tilde{A} - c(\hat{z})) \overset{p}{\to} C > 0$ from the proof of Lemma 6. Then if $\gamma = 1/4$, we have $\sqrt{n}(\hat{A}_{kcv} - c(\hat{z})) \overset{d}{\to} N(C, \sigma^2)$; if $\gamma < 1/4$, we have: $n^{2\gamma}(\hat{A}_{kcv} - c(\hat{z})) \overset{p}{\to} C$. In both cases, S5 is invalid. Then we finish the proof.

Then we consider the coverage invalidity of $c(z^*)$ and prove it by contradiction. Suppose otherwise $\sqrt{n}(\hat{A}_{kcv} - c(z^*)) \overset{d}{\to} N(0, \sigma^2)$ (the coverage validity holds). If $\gamma < 1/4$, then $n^{2\gamma}(\hat{A}_{kcv} - c(z^*)) \overset{d}{\to} 0$. Therefore, $n^{2\gamma}(\hat{A}_{kcv} - c(z^*)) \overset{p}{\to} 0$. However, recall $\hat{A}_{kcv}$ is the estimate of $n(1 - 1/K)$ samples from Lemma 6, we know, $n^{2\gamma}(\hat{A}_{kcv} - c(z^*)) \overset{p}{\to} C/(1 - 1/K)^{2\gamma}$ and contradict with the coverage validity condition; If $\gamma = 1/4$, we still have $n^{1/2}(\hat{A}_{kcv} - c(z^*)) \overset{p}{\to} C/(1 - 1/K)^{2\gamma}$, which contradicts with $\sqrt{n}(\hat{A}_{kcv} - c(z^*)) \overset{d}{\to} N(0, \sigma^2)$. Therefore, as long as $\gamma \leq 1/4$, we do not have such coverage for $c(z^*)$ in both $K$-fold and LOOCV approaches. $\square$

Furthermore, we may provide the coverage validity of CV intervals for $c(z^*)$ as in the plug-in approach when $\gamma > 1/4$.

**Corollary 2 (Coverage of CV Intervals)** *Suppose Assumptions 1, 2, 3 and 4 hold. When $\gamma > 1/4$, both $I_{kcv}$ and $I_{loocv}$ in (2) provide valid coverage guarantees for $c(\hat{z}), \mathbb{E}[c(\hat{z})]$ and $c(z^*)$.*

*Proof of Corollary 2.* Since we are in the situation where Lemma 14 holds. Then when $\hat{z}(\cdot) - z^*(\cdot) = o_p(n^{-\frac{1}{4}})$, we only need to show that $\tilde{A} - \mathbb{E}_{\mathbb{P}^*}[\ell(z^*(X); Y)] = o_p(n^{-\frac{1}{2}})$, which is given by the result already presented such that $\mathbb{E}_{\mathbb{P}^*}[c(\hat{z}^{(-N_k)}(X); Y) - \mathbb{E}_{\mathbb{P}^*}[\ell(z^*(X); Y)] = O(\|\hat{z}^{(-N_k)}(X) - z^*(X)\|^2) = o_p(n^{-\frac{1}{2}})$ since $n - |N_k| = n(1 - 1/K) = \Theta(n)$. Then combining this with Lemma 13, that interval produced in Lemma 14 provides valid coverage guarantees for $\mathbb{E}_{\mathbb{P}^*}[\ell(z^*(X); Y)]$. The argument that the interval provides valid guarantee for $\mathbb{E}_{\mathbb{P}^*}[\ell(\hat{z}(X); Y)]$ holds similarly. $\square$

# F  Detailed Experimental Results in Section 5

The experiments were run on a normal PC laptop with Processor 8 Core(s), Apple M1 with 16GB RAM. It took around 80 hours to run all the experiments including regression and portfolio study. All the optimization problems, if cannot solved directly using `scikit-learn`, are implemented through the standard solver `cvxopt`.

We consider the regression and CVaR portfolio optimization problem. The latter two objectives are two classical constrained contextual piecewise linear optimization problems. Each case we run $m = 500$ problem instances. For the standard error reported in Figure 1 and the following tables, we calculate it as:

$$\sigma_x = \frac{\sigma}{\sqrt{m}},$$

where $\sigma$ is the standard deviation of each reported result (i.e. interval width, bias size). We report 1-sigma standard error since the standard error of both interval width and bias scales are small.

The corresponding table with a full set of sample sizes $n$ is shown as follows in Table 3.

## F.1  Regression Study

**Setups.** We construct the synthetic dataset through the `scikit-learn` using `make_regression` function. More specifically, we set 10 features and standard deviation being 1, and others being the default setup. We set random seed from 0 - 500 to generate 500 independent instances. And we approximate $c(\hat{z})$ through additional 10000 independent and identicailly distributed test samples.

Table 3: Evaluation performance of different methods, where boldfaced values mean **valid coverage** for $c(\hat{z})$ (i.e., within [0.85, 0.95]) and boldfaced values in parantheses mean valid coverage for $c(z^*)$. IW and biases for kNN and Forest in the regression problem are presented in unit $\times 10^3$.

| method | $n$ | Plug-in | | | 2-CV | | | LOOCV | | |
|---|---|---|---|---|---|---|---|---|---|---|
| - | - | cov90 | IW | bias | cov90 | IW | bias | cov90 | IW | bias |
| **Regression Problem** ($d_x = 10, d_y = 1$) | | | | | | | | | | |
| Ridge | 600 | 0.76 (0.78) | 0.24 | 0.04 | 0.08 (0.01) | 0.33 | -0.34 | 0.82 (0.59) | 0.25 | 0.00 |
| | 1200 | 0.77 **(0.95)** | 0.16 | 0.02 | 0.55 (0.31) | 0.18 | -0.08 | 0.78 (0.90) | 0.17 | 0.00 |
| | 2400 | **0.85** (0.97) | 0.11 | 0.01 | 0.79 (0.79) | 0.12 | -0.02 | **0.86 (0.95)** | 0.12 | -0.00 |
| | 4800 | **0.88 (0.93)** | 0.08 | 0.00 | **0.89 (0.92)** | 0.08 | -0.01 | **0.89 (0.92)** | 0.08 | 0.00 |
| kNN $n^{2/3}$ | 600 | 0.81 (0.00) | 3.78 | 0.31 | 0.66 (0.00) | 4.19 | -1.53 | 0.82 (0.00) | 3.84 | 0.05 |
| | 1200 | 0.80 (0.00) | 2.48 | 0.12 | 0.43 (0.00) | 2.72 | -1.43 | 0.80 (0.00) | 2.50 | -0.03 |
| | 2400 | 0.84 (0.00) | 1.63 | 0.08 | 0.30 (0.00) | 1.77 | -1.21 | **0.85** (0.00) | 1.64 | -0.01 |
| | 4800 | **0.87** (0.00) | 1.11 | 0.02 | 0.12 (0.00) | 1.20 | -1.13 | **0.86** (0.00) | 1.11 | -0.03 |
| Forest | 600 | 0.74 (0.00) | 4.13 | 0.82 | 0.57 (0.00) | 4.73 | -1.92 | 0.77 (0.00) | 4.32 | 0.04 |
| | 1200 | 0.77 (0.00) | 2.71 | 0.40 | 0.41 (0.00) | 3.06 | -1.83 | 0.69 (0.00) | 2.78 | 0.02 |
| | 2400 | 0.77 (0.00) | 1.77 | 0.26 | 0.29 (0.00) | 1.97 | -1.54 | 0.72 (0.00) | 1.80 | 0.01 |
| | 4800 | **0.86** (0.00) | 1.19 | 0.15 | 0.10 (0.00) | 1.33 | -1.29 | **0.85** (0.00) | 1.20 | -0.03 |
| **CVaR-Portfolio Optimization** ($d_x = 5, d_y = 5$) | | | | | | | | | | |
| SAA | 600 | 0.68 (0.62) | 0.05 | -0.00 | 0.61 (0.65) | 0.05 | -0.00 | 0.71 (0.78) | 0.05 | 0.00 |
| | 1200 | 0.82 **(0.88)** | 0.04 | 0.00 | 0.82 **(0.87)** | 0.04 | 0.00 | **0.89 (0.88)** | 0.04 | -0.01 |
| | 2400 | **0.90 (0.89)** | 0.02 | -0.00 | **0.91 (0.89)** | 0.02 | -0.00 | **0.92 (0.89)** | 0.02 | -0.01 |
| kNN $n^{1/4}$ | 600 | 0.00 (0.00) | 0.31 | 2.41 | 0.83 (0.00) | 0.81 | -0.22 | **0.91** (0.00) | 0.76 | -0.05 |
| | 1200 | 0.00 (0.00) | 0.23 | 2.01 | 0.65 (0.00) | 0.53 | -0.21 | **0.92** (0.00) | 0.50 | -0.03 |
| | 2400 | 0.00 (0.00) | 0.17 | 1.72 | 0.42 (0.00) | 0.35 | -0.17 | **0.92** (0.00) | 0.33 | -0.00 |
| | 4800 | 0.00 (0.00) | 0.12 | 1.43 | 0.15 (0.00) | 0.23 | -0.17 | **0.88** (0.00) | 0.22 | -0.00 |

**Models.** We consider the following optimization models by calling the standard `scikit-learn` package: (1) Ridge Regression Models, implemented through `linear_model.Ridge(alpha = 1)`; (2) kNN, implemented through `KNeighborsRegressor` with nearest neighbor number being $\lceil 2n^{2/3} \rceil$; (3) Random Forest, implemented through `RandomForestRegressor` with 50 subtrees and sample ratio being $n^{-0.6}$.

**Additional Results.** Table 3 reports all results in the regression (and the portfolio optimization) case, which is a superset of Table 2. In Table 4, we present the standard error of the interval width and bias for each method used in Table 3. The bias size of plug-in and 2-CV are significant if we take the size of the standard deviation into account. Here, the plug-in approach still has the smallest standard error of the bias and interval width.

Table 4: Interval Widths and Biases for each Evaluation Procedure for the Regression Problem (Mean and Standard Error).

| method | $n$ | Plug-in | | 2-CV | | LOOCV | |
|---|---|---|---|---|---|---|---|
| - | - | IW | bias | IW | bias | IW | bias |
| Ridge | 600 | 0.24±0.00 | 0.04±0.00 | 0.33±0.00 | -0.34±0.01 | 0.25±0.00 | 0.00±0.00 |
| | 1200 | 0.16±0.00 | 0.02±0.00 | 0.18±0.00 | -0.08±0.00 | 0.17±0.00 | 0.00±0.00 |
| | 2400 | 0.11±0.00 | 0.01±0.00 | 0.12±0.00 | -0.02±0.00 | 0.12±0.00 | -0.00±0.00 |
| | 4800 | 0.08±0.00 | 0.00±0.00 | 0.08±0.00 | -0.01±0.00 | 0.08±0.00 | 0.00±0.00 |
| kNN | 600 | 3778.22±47.28 | 314.75±63.46 | 4187.77±52.33 | -1532.71±68.55 | 3840.78±48.04 | -54.78±63.46 |
| | 1200 | 2477.47±30.83 | 115.55±44.65 | 2718.83±33.69 | -1428.88±49.23 | 2503.31±31.16 | -34.28±44.70 |
| | 2400 | 1628.36±19.84 | 80.66±27.45 | 1771.38±21.58 | -1206.15±32.33 | 1638.95±19.97 | -25.37±27.49 |
| | 4800 | 1108.71±13.45 | 23.85±18.95 | 1199.71±14.53 | -1130.71±24.16 | 1113.23±13.50 | -27.48±18.99 |
| Forest | 600 | 4132.75±56.14 | 824.40±72.64 | 4733.39±61.46 | -1917.23±91.12 | 4315.90±57.17 | 41.13±81.11 |
| | 1200 | 2711.23±37.61 | 404.59±49.88 | 3062.29±41.06 | -1832.96±70.97 | 2782.08±37.87 | 24.61±62.26 |
| | 2400 | 1768.86±24.05 | 259.31±32.17 | 1966.98±26.35 | -1543.64±55.67 | 1801.58±24.52 | 13.72±42.25 |
| | 4800 | 1185.55±16.51 | 145.36±19.48 | 1328.21±17.97 | -1287.63±42.28 | 1201.60±16.70 | -10.49±31.94 |

We change the fold number to be 5, 10, 20 to allow variabilities in the fold number. We run all procedures again and report results in Table 5. Although $K$-fold CV with a large number of $K$ has better coverage when $n$ is small compared with that of 2-CV in Table 3, as $n$ becomes larger, the coverage becomes smaller while the bias

becomes more significant compared with the interval length. This indicates the bias decrease order is slower than that of the interval and validates our theoretical results.

Table 5: Performance Evaluation of 5, 10, 20-CV Intervals compared with plug-in estimator in the Regression Problem (Mean and Standard Error), where boldfaced values mean **valid coverage** for $c(\hat{z})$ (i.e., within [0.85, 0.95]), IW and bias for each procedure are presented in unit $\times 10^3$ and each entry is averaged over 200 experiment repetitions.

| method | $n$ | Plug-in | | 5-CV | | 10-CV | | 20-CV | |
|---|---|---|---|---|---|---|---|---|---|
| - | - | cov90 | IW (bias) | cov90 | IW (bias) | cov90 | IW (bias) | cov90 | IW (bias) |
| | | | | **Regression Problem** ($d_x = 10, d_y = 1$) | | | | | |
| Forest | 1200 | 0.77 | 2.71 (0.40) | 0.62 | 2.83 (-0.42) | 0.65 | 2.81 (-0.48) | 0.69 | 2.74 (-0.30) |
| | 2400 | 0.77 | 1.77 (0.26) | 0.66 | 1.83 (-0.37) | 0.60 | 1.76 (-0.29) | 0.64 | 1.76 (-0.23) |
| | 4800 | **0.86** | 1.19 (0.15) | 0.47 | 1.24 (-0.32) | 0.56 | 1.26 (-0.38) | 0.63 | 1.24 (-0.18) |
| | 9600 | **0.85** | 0.73 (0.11) | 0.42 | 0.76 (-0.28) | 0.52 | 0.73 (-0.22) | 0.51 | 0.74 (-0.17) |
| | 19200 | **0.86** | 0.47 (0.07) | 0.34 | 0.49 (-0.23) | 0.42 | 0.48 (-0.13) | 0.45 | 0.48 (-0.11) |
| kNN $n^{2/3}$ | 1200 | 0.81 | 2.48 (0.12) | 0.76 | 2.57 (-0.46) | 0.77 | 2.51 (-0.46) | 0.76 | 2.48 (-0.21) |
| | 2400 | 0.80 | 1.63 (0.08) | 0.78 | 1.68 (-0.38) | 0.72 | 1.62 (-0.29) | 0.74 | 1.71 (-0.24) |
| | 4800 | 0.84 | 1.11 (0.03) | 0.66 | 1.14 (-0.37) | 0.70 | 1.15 (-0.21) | 0.68 | 1.14 (-0.18) |
| | 9600 | **0.88** | 0.75 (0.04) | 0.61 | 0.77 (-0.28) | 0.69 | 0.70 (-0.15) | 0.70 | 0.69 (-0.12) |
| | 19200 | **0.85** | 0.46 (0.04) | 0.49 | 0.47 (-0.23) | 0.66 | 0.46 (-0.09) | 0.68 | 0.46 (-0.04) |
| | | | | **CVaR-Portfolio Optimization** ($d_x = 5, d_y = 5$) | | | | | |
| kNN $n^{1/4}$ | 2400 | 0.00 | 0.172 (1.716) | 0.76 | 0.337 (0.079) | 0.82 | 0.328 (0.035) | **0.85** | 0.329 (0.028) |
| | 4800 | 0.00 | 0.126 (1.423) | 0.74 | 0.221 (0.038) | 0.79 | 0.218 (0.034) | 0.80 | 0.217 (0.025) |
| | 9600 | 0.00 | 0.094 (1.108) | 0.66 | 0.147 (0.029) | 0.72 | 0.146 (0.028) | 0.76 | 0.144 (0.021) |

## F.2 CVaR-Portfolio Optimization

**Setups.** We set $\eta = 0.2$, $d_x = d_y = d_z = 5$ and $\mathbb{P}_X = N(\mathbf{0}, \mathbf{\Sigma})$ with $(\mathbf{\Sigma})_{ij} = 0.8^{|i-j|}, \forall i, j$. And the conditional distribution $(Y)_i | x = 0.3 \times (Bx)_i + 2|\sin \|x\|_2| + \epsilon, \forall i \in [d_z]$ for each $\epsilon \sim N(0, 4)$.

**Models.** We consider the following optimization learners for the subsequent performance assessment: (1) Sample Average Approximation (SAA): We ignore features when making decisions but consider the same decision space $\{z \in \mathcal{Z}\}$. That is, for any covariate $x$, we output the same decision: $\hat{z} \in \text{argmin}_{z \in \mathcal{Z}} \frac{1}{n} \sum_{i \in [n]} [\ell(z; Y)]$; And the convergence rate $\gamma = \frac{1}{2}$; (2) kNN: We use the model from Example 3 with $k_n = \min\{3n^{\frac{1}{4}}, n-1\}$. In this case, the convergence rate $\gamma_v = \frac{1}{8}$, which violates the validity condition of both plug-in and $K$-fold CV intervals.

## F.3 Regression Study in the Additional Real-World Dataset

We include one real-world dataset `puma32H`[2] with 33 features and 1,000,000 samples as a regression task. We report the coverage probability of plug-in, 2-CV, and 5-CV for a kNN model with $k_n = n^{2/3}$, where each entry denotes the coverage probability estimated over 100 experimental repetitions for that procedure in Table 6. Here each experimental repetition is conducted differently by shuffling the entire dataset. For a given sample size $n$, the procedure is to select the first $n$ rows as the training sample for each model and approximate the true model performance $c(\hat{z})$ by averaging over the remaining samples in that dataset.

Table 6: Coverage Results of the method kNN in the dataset `puma32H`, where boldfaced values mean **valid coverage** for $c(\hat{z})$ (i.e., within [0.85, 0.95]).

| $n$ | 10000 | 20000 | 40000 |
|---|---|---|---|
| Plug-in | **0.94** | **0.93** | 0.90 |
| 2-CV | **0.88** | 0.81 | 0.76 |
| 5-CV | **0.91** | **0.86** | 0.81 |

In Table 6, the plug-in approach provides valid coverages in this case while 2-CV and 5-CV do not, especially when $n$ is larger (20000 and 40000). These results continue to validate our asymptotic theory.

---

[2]The dataset is available at `https://www.openml.org/d/1210`.

# G Further Discussion and Comparison

## G.1 Additional Discussion in Model Selection

A difference between the model selection and model evaluation task we focus on in this paper is that in the former, we focus on the performance rank between different models instead of the absolute performance value. Intuitively, the accuracy of the performance rank depends on not only the evaluations of the compared models, but also the inter-dependence between these evaluation estimates. This adds complexity to understanding the errors made by each selection approach. There are some discussions for specific problems, such as linear regression, classification, and density estimation (see, e.g., Section 6 in [5]). However, the theoretical understanding of model selection remains wide open in general problems.

That being said, there are some model selection problems where plug-in easily leads to a naive selection.

- Selecting hyperparameters: Consider the best regularized parameter $\alpha$ in the ridge regression. Plug-in always chooses $\alpha = 0$ since it has the smallest training loss;

- Selecting the best model class: Consider the regression problem $\hat{f} \in \operatorname{argmin}_{f \in \mathcal{F}_i} \sum_{i=1}^{n} (f(X_i) - Y_i)^2$ for the nested classes $\mathcal{F}_1 \subset \mathcal{F}_2 \subset \mathcal{F}_3$, and we want to select the best $\mathcal{F}_i$ among the three classes. Plug-in always selects the largest class, $\mathcal{F}_3$, since it has the smallest training loss.

In these problems, CV can select the regularization parameter or model class different from the naive choice. It hints that CV could be better than plug-in for such a problem, but this is theoretically not well-understood in general cases.

## G.2 Additional Discussion in the High-Dimensional Setting

For the high-dimensional setting, i.e., feature dimension $p$ and sample size $n$ both go to infinity such that $p/n$ converges to a nonzero constant, besides specific problems like (generalized) linear regression, the problem is wide open to our best knowledge. For linear models, we have the following result towards the bias of our considered three procedures:

- For plug-in, when $p > n$, the predictor interpolates the training data so that the training loss becomes zero (unless we add regularization); when $p/n \to c \in (0, 1)$, plug-in still suffers a large bias from overfitting and cannot be used directly to construct the point estimator and confidence interval to evaluate model performance. Some bias correction procedures ([27]) are proposed to construct consistent point estimators in these high-dimensional scenarios.

- For $K$-fold CV, the point estimate also suffers from a non-vanishing bias [58] and may not be a good choice for evaluating model performance.

- For LOOCV, recent literature [58, 48] shows that its bias goes to zero.

Despite the known results above for the bias, no theoretically valid coverage guarantees exist for high-dimensional linear models in the literature, since almost all existing CV literature is based on some stability conditions and is only valid under low-dimensional asymptotic regimes ([7, 9, 37] and ours).

For our three procedures, we validate claims for the point estimates above and also investigate the coverage performance of interval estimates, using the same simulation data setup for ridge regression with $\alpha = 1$ as in our regression problems. We show the simulation results in Table 7, where both plug-in and 5-CV suffer from large bias and have zero coverage. The bias of LOOCV is small but the LOOCV interval suffers from poor coverage. These results indicate that constructing theoretically valid intervals remains open and challenging for high-dimensional problems, and is important to devise improved CV or bias correction approaches.

Table 7: Evaluation performance of **high-dimensional** ridge regression (100 repetitions) in the regression problem

| $(n, p)$ | Plug-in | | 5-CV | | LOOCV | |
|---|---|---|---|---|---|---|
| | cov90 | bias | cov90 | bias | cov90 | bias |
| (300, 200) | 0.00 | 21.63 | 0.00 | -48.72 | 0.59 | -0.38 |
| (900, 600) | 0.00 | 4.62 | 0.00 | -8.45 | 0.51 | -0.03 |
| (1800, 1200) | 0.00 | 2.81 | 0.00 | -5.75 | 0.58 | -0.01 |

## G.3 Comparison with Nested Cross Validation

We elaborate on the discussion of the nested cross-validation and show it is not suitable under our slow rate regime. Recall that the nested cross-validation (NCV) in [9] is designed for improving the covariance estimate in high-dimensional regimes. Therefore, NCV does not help theoretically and empirically:

**Theoretical Invalid Coverages.** NCV does not offer valid coverages for nonparametric models theoretically: The interval length of NCV is controlled by $\sqrt{MSE}$. Due to the restriction $\sqrt{MSE} \in [SE, \sqrt{K}SE]$ in Section 4.3.2 in [9] (SE $= \Theta(n^{-1/2})$ is the standard error in their eq. (2)), this interval length is $\Theta(1/n^{1/2})$ for a fixed $K$. In nonparametric models with $\gamma < 1/4$, this length is overly small compared with the bias of NCV, which is $Cn^{-2\gamma}$ for a constant $C > 0$ smaller than that of (2) in our main body. This is despite that the CV point estimator in [9] used bias correction in their eq. (9), that correction is for parametric models but not nonparametric models. Then the bias in [9] dominates the interval length and leads to an invalid coverage asymptotically. Furthermore, [9] does not provide rigorous theoretical guarantees on the valid coverage of their interval (9), even in their own setting. They only correct the variance estimate for parametric models.

**Empirical Poor Performance.** NCV does not perform well empirically in our setting: This is expected from the above explanation. We implement NCV with 5 and 10 folds for randomforest in the regression problem with the same setup in Appendix F.1 and report results in Table 8. Here, plug-in, 5-CV, 10-CV correspond to the intervals (2) in our paper. For NCV, we use Algorithm 1 in [9] (following their practical MSE restriction in Section 4.3.2) and bias estimation in Appendix C of [9] to construct the interval for NCV (5-NCV and 10-NCV).

While the bias of NCV is controllable and the interval gives nearly valid coverage when n is small, the larger bias order starts to exert effect when $n$ is large, leading to a significant drop in the coverage of NCV which becomes invalid.

**Computational Inefficiency.** Note that [9] does not focus on computation expense as we do in our paper in comparing LOOCV with plug-in. As mentioned in [9], NCV is computationally very demanding, requiring over 1000 random splits to stabilize the variance estimate and needs to refit in total $1000K$ times.

Table 8: Evaluation performance of NCV versus CV for the forest learner in the regression problem, where boldfaced values mean **valid coverage** for $c(\hat{z})$ (i.e., within [0.85, 0.95]), IW and bias for each procedure are presented in unit $\times 10^2$ and each entry is averaged over 200 experiment repetitions.

| $n$ | Plug-in | | | 5-CV | | | 5-NCV | | | 10-CV | | | 10-NCV | | |
|---|---|---|---|---|---|---|---|---|---|---|---|---|---|---|---|
| | cov90 | IW | bias | cov90 | IW | bias | cov90 | IW | bias | cov90 | IW | bias | cov90 | IW | bias |
| 10000 | **0.88** | 7.27 | 0.35 | 0.41 | 7.53 | -4.36 | 0.81 | 15.06 | -1.13 | 0.48 | 7.41 | -2.65 | 0.80 | 13.86 | -1.07 |
| 30000 | **0.86** | 3.73 | 0.22 | 0.38 | 3.82 | -2.48 | 0.74 | 7.65 | -0.56 | 0.29 | 3.80 | -2.21 | 0.65 | 7.59 | -1.15 |
| 100000 | **0.90** | 1.71 | 0.11 | 0.19 | 1.75 | -2.25 | 0.53 | 3.51 | -0.48 | 0.27 | 1.73 | -1.26 | 0.56 | 3.45 | -0.61 |
| 300000 | **0.88** | 0.92 | 0.08 | 0.13 | 0.93 | -1.76 | 0.47 | 1.87 | -0.28 | 0.26 | 0.92 | -0.56 | 0.4 | 1.85 | -0.31 |

