# OpenReview forum: "Is Cross-validation the Gold Standard to Estimate Out-of-sample Model Performance?"
_NeurIPS.cc/2024/Conference — NeurIPS 2024 poster_

### Official Review · Reviewer_tGmJ · 2024-07-09

**Soundness:** 4
**Presentation:** 3
**Contribution:** 4
**Rating:** 8
**Confidence:** 3

**Summary:**

This is a primarily theoretical paper describing the bias and confidence intervals formed from different methods of assessing the true out-of-sample error of various models. In particular, this paper compares cross-validation (CV) to the "plug-in" estimator (i.e., the training loss). The authors assess the methods based on two criteria: how quickly the converge towards the true out-of-sample error as one gathers more data $n$, and whether or not the confidence intervals (formed as the usual $1-\alpha$ intervals of a CLT approximation to the folds of cross-validation / loss on each training point) achieve the proper coverage of $1-\alpha$.

The authors make two main very practical conclusions, regardless of model type (parametric or nonparametric):

1. That there is essentially no point to using $K$-fold CV for some fixed $K$, as it has equal bias to the plug-in estimate and achieves coverage for strictly fewer models than does the plug-in.
2. The utility of leave-one-out CV (LOOCV) is questionable, since although it has somewhat better astymptotic performance than plug-in, the gains may not justify its usually enormous computational cost.

**Strengths:**

*Contribution:* CV is an extremely widely used methodology that had, until somewhat recently, very slim theoretical underpinnings. I think this paper is a great help in ameliorating that problem, and it even makes some surprising and well-backed-up arguments about the use of CV in practice. In particular, I don't think many attendees of NeurIPS would believe the statement "using the training loss is pretty much just as good as using $K$ fold CV"! But, with this paper (hopefully) at the conference, people may start to ask themselves interesting questions about the use of $K$ fold.

In terms of a more specific contribution, the authors note that previous theoretical work has been light in many cases where the convergence rate of the model is asymptotically less than $1/n^{1/4}$, whereas their paper details this case (as well as the $>1/n^{1/4}$ case) clearly. I'm not deeply familiar enough with the theoretical CV literature to validate this claim about previous work, but I think this is an important contribution in and of itself if it's correct.

*Clarity:* In general, the paper is pretty well written. It's a fairly technical paper, but the authors do a good job of making it straightforward to read. I even found the first chunk of the proofs pretty easy to read through (although I got tripped up eventually, see weaknesses). Overall, I found all concepts completely clearly and precisely defined, which I think is not true of many theoretical papers submitted to NeurIPS.

**Weaknesses:**

I have a few miscellaneous comments. I don't think any of them are particularly important with the exception of the first two.

**Claimed strength of results**

The introduction says "[we] identify *necessary* conditions, in contrast to merely *sufficient* conditions in the literature, under which the plug-in and CV variants exhibit low biases and valid coverage." I think a paper truly doing this would be incredibly exciting. However, I do not think this paper does this. In particular, there are eight enumerated assumptions that are used throughout the paper (Assumptions 1-4 in particular), and I did not see any analysis that these assumptions are necessary. The paper *does* completely break down how the bias and coverage depend on the convergence rate of the model at hand. And I think that's great. But I think it means these statements in the introduction and abstract aren't quite correct and should be clarified.

**Utility of CV versus plug-in (i.e., training loss)**

The paper kind of implies in its intro / abstract that it is going to show $K$-fold and LOOCV are actually not as good as the plug in estimator. I think Theorems 1 and 2 (the main theoretical results of the paper) show a more nuanced picture than this, though:

1. $K$-fold.  Theorem 1 shows that $K$-fold (for $K = O(1)$)  has bias that is equal in magnitude to plug-in, and Theorem 2 shows it has equivalent asymptotic coverage. The conclusion from this is very clearly spelled out: "plug-in is always no worse than $K$-fold CV." This might make it *sound* like $K$-fold is a waste of time to a practitioner, but I would argue it's. In particular, plug-in is biased to be *below* the true out-of-sample error, whereas $K$-fold is biased to be *above* the true error. Think of the perspective of a risk-adverse user or a regulator deciding whether to allow a model in a high-stakes situation (self-driving cars, medical procedures, etc.). I really can't see such users ever seeing plug-in as equivalent to $K$-fold; in fact, I would argue that they would hugely prefer to use $K$-fold precisely because its evaluation of the model will be conservative.

2. I thought some of the discussion in the paper was critical of LOOCV's statistical performance, however, LOOCV seems to outperform plug-in in both Theorem 1 and 2. I appreciated the point that one might see the statistical gains as minor in exchange for the computational expense. But I think it could be clearer that the takeaway is "maybe don't bother with LOOCV, especially if you have a parametric model that's nearing it's 'asymptotic regime'".

I'm calling this out here because I think this is a really interesting paper that might get highlighted at the conference. Many non-theoretically oriented people attend NeurIPS, and I really wouldn't want them to get the wrong idea based on reading the introduction / abstract. A doctor making the medical device that will diagnose any of our future tumors might be in attendance -- do you *really* want them to use plug-in over $K$-fold to assess the quality of their algorithms?

**Small issues in theorems**

1. I think it should be made clear when $K$ is a constant, and when it can vary with $n$. I *think* Theorem2 wants it to be a constant whereas Theorem 4 allows it to scale with $n$. But this should be clear.
2. In Theorem 2, why do we care about the coverage of $c(z^*)$? Is this just a fun fact? Some interpretation would be good.
3. Theorem 4 seems to give a much more specific description of LOOCV's bias than does Theorem 1 -- why not state Theorem 4's result in Theorem 1?
4. I was a little confused by Theorem 3/4 being labeled as "Theorems"; they seem like Lemmas to me given that they are building blocks used for Theorems 1/2.
5. In Theorem 2, when $\gamma_v \leq 1/4$, the asymptotic coverage of plug-in is given as $\leq 1-\alpha$. The text then states that equality holds (i.e. coverage is $= 1-\alpha$) when $\gamma_v > 1/4$. But the point is that we're covering $\gamma_v \leq 1/4$, so I think the $\leq$ should be a $<$.
6. I often thought that restating the theorem would be helpful before the proofs in the Appendices; I had to open the paper in multiple windows to flip through everything.
7. There are a lot of random quantities floating around the paper, and it wasn't always clear what expectations were being taken over. It would be good to subscript expectations with what they're with respect to wherever possible.
8. Line 130: I think the upper bound on the Hessian here is from Assumption 2, not Assumption 3.
9. I got a little lost in the proof of Theorem 3 when it suddenly jumped into discussing Assumption 8. First, Assumption 8 isn't an assumption of the Theorem, and Assumption 8 seems to be about nonparametric models, whereas Theorem 3 is generic. I think it could be more clear what's going on here.

**High dimensions**

I think a comparison in high dimensions is a major missing piece to the story here -- at least for parametric models. In particular, the use of cross-validation in the fixed dimension $n \to \infty$ case is a lot less relevant, as in this case it's well known (e.g. from empirical risk minimization theory) that the plug-in estimate will perform well. In high dimensions, say where $n/d \to$ some constant, the plug-in often does not perform well, whereas cross-validation can. I don't think this is a significant direction for change in the paper, but I do think a little discussion (maybe in the future work section) would be good.

**Questions:**

I'm curious how the results hold up in high dimensions, but I don't think this is a critical direction for the paper.

**Limitations:**

I think the authors could use a little more discussion of the limitations of their results (see the first section under "Weaknesses"). I don't think there are any societal impacts of this paper.

---

> ### Author Rebuttal · Authors · 2024-08-07
>
> We sincerely thank you for recognizing our contributions both on the theoretical and practical fronts, and also for the detailed and very helpful suggestions. We address your comments point-by-point as follows.
>
> **Claimed strength of results**: We would follow your suggestion and change that claim in the introduction to "This analysis helps us provide a complete breakdown of how bias and coverage depend on the convergence rate of the model at hand. This in turn fills in the gap in understanding which methods outperform which others, in regimes that have appeared challenging for previous works." We will also make similar changes in the abstract.
>
> Actually, our original thought was similar to your comment in that we regarded the complete breakdown as a provision of both sufficient and necessary conditions, with the list of assumptions being used to prescribe some regularity in the setting. Nonetheless, we see your very valid point that the use of the term ``necessary conditions" would cause confusion, and so we would change that as you suggested.
>
>
> **Utility of CV versus plug-in**:
> We agree with the reviewer on these additional helpful insights, and would add them into our discussion in Section 3. Specifically,
> - In the "Comparing plug-in and K-fold CV" part, we would add "In terms of the magnitude of bias and interval coverage, plug-in is always no worse than K-fold CV. However, in terms of the direction of bias, plug-in gives an optimistic evaluation of the true model performance, while K-fold CV gives a pessimistic evaluation. In high-stake scenarios where a conservative evaluation is preferred, e.g., evaluating the treatment effect of a new drug, K-fold CV can be more desirable than plug-in. "
> - In the "Comparing plug-in and LOOCV" part, we would add "In these cases where LOOCV and plug-in give similar evaluation intervals statistically, plug-in should be preferred as it is computationally much more efficient."
>
>
>
> **Small issues in theorems**:
> First, we would incorporate your suggestions in Points 4 (change to lemmas), 6, 7, 8.
> - For point 1, indeed K can scale with n in the result of Theorem 4, and we would clarify the dependence of K there.
> - For point 2, the coverage of $c(z^*)$ is of interest to some problems in stochastic optimization, namely in estimating the optimality gap to decide whether one should stop an optimization algorithm. We have mentioned some of this literature [10, 42, 45, 53, 54] in Appendix A. We would move this related discussion to the exposition after Theorem 2.
> - For point 3, indeed Theorem 4 implies Theorem 1. Nonetheless, our Theorem 1 highlights the difference among the three types of estimators with respect to the sample size $n$ and model convergence rate $\gamma_v, \gamma$, which we think is helpful for the reader. We are happy to change and merge Theorems 1 and 4 if the reviewer thinks this would be better.
> - For point 5, the general condition in Line 155 is under $\gamma \leq 1/4$. However, the condition in line 156 - 157 is given by $\gamma_v$. Note that $\gamma = \min(\gamma_b, \gamma_v)$, so that equality (i.e., the valid coverage guarantee of the plug-in estimator) holds when $\gamma \leq 1/4$ and $\gamma_v > 1/4$.
> - For point 9, we apologize for the confusion in the proof of Theorem 3. First, in terms of Assumption 8, the models we study in this paper are covered in Definitions 1 and 2, while Assumption 8 governs the regularity of the (nonparametric) models under Definition 2. Next, for parametric models, existing theoretical results ([36, 46]) have shown that the bias is of order $\Theta(1/n)$ (since $\gamma = 1/2$ as we mention in Proposition 1) under our assumptions. We would incorporate the missing discussion on the relation with Assumption 8 and in terms of the parametric model in the proof in our revised version.
>
>
> **High-dimensional Investigation**:
> The reviewer certainly raises a good question about high-dimensional investigation. We provide some discussions in the Global Response. Essentially, both plug-in and $K$-fold CV may not perform well in this setting, and while there are works on addressing very specific problems (e.g., linear regression), to our best knowledge it remains an open problem to understand the comparisons among different evaluation methods for general high-dimensional problems.
>
> Lastly, regarding whether empirical risk minimization theory can give insights on the performance of plug-in (even in low dimension), our take is that, as this theory is mostly based on non-asymptotic bounds instead of asymptotically exact results (as we mentioned in Section 6), it could be difficult to know whether CV outperforms plug-in or vice versa. Besides, the performance of nonparametric estimators is not directly revealed from the standard empirical risk minimization theory which mostly focuses on parametric models (though with exceptions). Because of these, CV is still more commonly applied than plug-in, even though as our paper suggests plug-in could be superior when taking into account both statistical and computational benefits.

---

> > ### Comment · Reviewer_tGmJ · 2024-08-13
> >
> > Summary: after reading the reviews and other responses, I think the paper will be even stronger. I would vote for it being highlighted at the conference (e.g. an oral presentation); I think it has a lot of thought-provoking theoretical **and** empirical implications.
> >
> > I think there was a lot of interesting discussion in the above reviews and responses! I especially thought JBLc’s comments on model selection —and the authors’ responses — were interesting, as this is a really straightforward failure case of plug-in that I hadn’t thought of while reading the paper. I think calling this out in the paper adds to the mystery a bit: plug-in is just as biased as K-fold for any *fixed* model, and yet, somehow, is horrible for model selection. This seems like an interesting direction for future work. I also appreciated the discussion of the high-dimensional work as an interesting direction for more future work.
> >
> > On merging Theorem 1&4 vs keeping them separate: I would just take the above comment as an $n=1$ sized observation that I found it confusing when reading the paper. I don't think it's a huge deal.

---

> > > ### Author Response · Authors · 2024-08-14
> > >
> > > We greatly thank the reviewer again for your constructive feedback and thoughtful reading for reviews. We also sincerely thank you for giving us a high evaluation! Specifically, we would emphasize the model selection and high-dimensional issues as our future work in our revised version.

---

### Official Review · Reviewer_8MSs · 2024-07-11

**Soundness:** 3
**Presentation:** 3
**Contribution:** 3
**Rating:** 5
**Confidence:** 2

**Summary:**

The paper establishes asymptotic results for bias and coverage of three different kinds of validation schemes, namely k-fold cross-validation, plug-in validation, and leave-one-out cross-validation. The setup is general and includes both parametric and non-parametric models. There are experimental results that cover parts of the finite sample context.

**Strengths:**

- The research area is important and interesting.
- The theoretical results are presented clearly and I have not found any errors in the derivations (but have not looked closely at the proofs).
- The assumptions are clearly stated and not overly restrictive.

**Weaknesses:**

- The paper is very technical in nature and contains an appendix of 20 pages, most of which are also technical. (I count to seven theorems, 14 lemmas, and seven propositions.) It is not reasonable to expect reviewers to be able to cover this much material in the limited time frame of the review process in NeurIPS. I think the paper would be more appropriate for a journal submission.
- The code in the supplement is not documented at all and does not contain any configuration or setup for reproducing the results. The project should come with a `pyproject.toml` file, `requirements.txt`, or similar configuration. The zip file also contains numerous files that shouldn't be there, such as cache files. As a result, it is hard to check the validity of the numerical experiments. The experiments also seem to depend on commercial software (MOSEK), which is not ideal.
- The empirical results are questionable to me. You say that the typical number of folds in cross-validation is 5, yet the majority of your experiments use 2 folds, which I have never seen used in practice. I agree that the most common values are 5 and 10, and your experiments should reflect this, plus values above and below to show how the effect of $K$. Perhaps 5, 10, and 20. You even say you haven't "cherry-picked" your results, but this is exactly what this looks like to me, especially when you only show 5 fold results in a few of the cases.
- The empirical results all rely on data with 10 features, making $n/p \geq 60$ everywhere. This is not really representative of the contexts in which these models are used. You should consider $p > n$ or at least lower ratios than this. [8] show that the coverage of the intervals depend on this ratio.
- The results are based on one particular type of interval estimate for $k$-fold cross validation, but it is not clear that this is a good choice. (See **Questions**.)

**Questions:**

### Questions

- It seems to me that your results hinges on the interval estimates defined via (1) and (2) but, as you yourself say, these are not the only possible estimates. And in fact in [8] it is shown that these estimates in fact have as poor coverage as the naive estimates, which I suspect must influence your results as well. Why did you not use the more accurate estimates presented in [8] and why is there not more discussion of the results in that paper in general?

### Suggestions

- Perhaps the title could be more informative. If I understand the paper correctly, you are stating that it is indeed _not_ the gold standard to evaluate model performance. So just say that in the title instead.
- L29: It would help to be more descriptive about what the "plug-in" approach is. Many techniques "re-use training data for model evaluation". Cross-validation also fits this description, for instance.
- L29: "Reuses" should be "reuse".
- L31: "Model rates" could mean several things. Be more descriptive.
- L33: Left quotation mark around "center" is not formatted correctly.
- L258: Title escapes into margin.
- L352: "as" should be "as they are".
- Figure 1: Double y-axes are generally hard to read. I suggest you just divide the plot into two instead.
- L953: You already have one example 3. Maybe you forgot to use the example environment here?
- L973: Same as the previous comment.
- Appendix: Equations escape into the right margin in several places.

**Limitations:**

- I am missing a discussion on optimistic vs. pesimistic bias
- The listed limitations should include the limitation of looking at one specific way of constructing the interval estimates.
- The results have no societal impacts as far as I can tell.

---

> ### Author Rebuttal · Authors · 2024-08-07
>
> We thank the reviewer for recognizing the significance of our results and the helpful suggestions. We address the reviewers' main concerns as follows.
>
> **Our choice of interval estimate, and not use the interval suggested in [8]**: The reviewer raises the valid point that our interval is one particular choice and questions why we don't use the interval in [8]. First, we point out that the interval in [8], namely eq. (9) in [8], which uses nested CV (NCV) in both the point and variance estimates in the interval, is designed for *(high-dimensional) parametric models*. In contrast, our paper focuses on the distinct setting of *nonparametric models*, especially in the most challenging *slow rate* regime. That is, NCV is not designed for, and would not be competitive to our setting. In fact, NCV does not help theoretically and empirically:
> - NCV does not offer valid coverages for nonparametric models theoretically: The interval length of NCV is controlled by $\sqrt{MSE}$. Due to the restriction $\sqrt{MSE} \in [SE, \sqrt{K} SE]$ in Sec 4.3.2 in [8] (SE = $\Theta(n^{-1/2})$ is the standard error in their eq. (2)), this interval length is $\Theta(1/n^{1/2})$ for a fixed $K$. In nonparametric models with $\gamma < 1/4$, this length is overly small compared with the bias of NCV being $C n^{-2\gamma}$ (for a constant $C > 0$ smaller than that of eq. (2) in our paper). This is despite that the CV point estimator in [8] used bias correction in their eq. (9), that correction is for parametric models but not nonparametric models. Then the bias in [8] dominates the interval length and leads to an invalid coverage asymptotically.
> - NCV does not perform well empirically in our setting: This is expected from the above explanation. We implement NCV with 5 and 10 folds for randomforest in the regression problem (same setting as Figure 1 in our paper), shown in Table 2 of the attached pdf. While the bias of NCV is controllable and the interval gives nearly valid coverage when n is small, the larger bias order starts to exert effect when $n$ is large, leading to a significant drop in the coverage of NCV which becomes invalid.
> - NCV is computationally very demanding, requiring over 1000 random splits to stabilize the variance estimate (as mentioned in [8]) and needs to refit in total 1000*K times. Note that [8] does not focus on computation expense as we do in our paper in comparing LOOCV with plug-in.
> - [8] does not provide rigorous theoretical guarantees on the valid coverage of their interval (9), even in their own setting. They only correct the variance estimate for parametric models.
>
> We will include the above comparison discussion with NCV in [8] in our future work part, and include the additional numerical results in Appendix in our revised version. In our comparison discussion, we would properly indicate that [8] focuses on a different setting and hence is disadvantageous for ours.
>
> Finally, back to our interval estimate (2), we stress that our choice is natural -- It is widely used with valid statistical properties and outperformance over earlier works as shown in [9]. [8] also considers this natural choice and shows that it performs well under large samples in their experiment (n>400 in Figure 10 in [8]). Our paper and [8] essentially point out two different problems by using (2): [8] corrects the variance estimate in high dimension but parametric models while we show that this interval does not yield valid coverage for nonparametric models with slow rates.
>
> **Code Reproducibility**: We apologize for not documenting the code well previously. We now provide clean documentation on the experimental configuration for reproducibility and also change the default solver to CVXOPT (open source package). Here is the anonymous link to our revised code: [Link](https://anonymous.4open.science/r/CV_GoldStandard-8E35).
>
> **Empirical results using different fold numbers**: We have shown 5-CV results for the regression problem in Table 5, and newsvendor problem in Table 7 in the paper. To alleviate further the reviewer's concern, we now also provide additional results of 5, 10, 20-CV for regression and portfolio tasks for more sample sizes in Table 1 of the attached pdf. We will include them in our revised version and use 5-CV throughout the main body. As shown in these results, larger K still suffers from coverage problems when n is large, even though it can help when $n$ is small. These continue to support the validity of our theoretical results.
>
> **Results on different dimensional problems**: As other reviewers mention, our main contribution and novelty are for the asymptotic regime with $n\to\infty$ and fixed $p$. This corresponds to most classical CV papers with statistical guarantees [5,6,9]. The models we choose, i.e., linear models, kNN, randomforest are common under this setup. Correspondingly, our empirical results are used to validate theoretical results in this regime. Nonetheless, we provide discussions on the high-dimensional case (including p > n) in the Global Response, but we politely point out that this is not our main focus and is worth a separate substantial work on its own (as Reviewer tGmJ also hints).
>
> **Other suggestions**: We use the question form in the title because the answer is not clear-cut - LOOCV is indeed the best statistically but computationally expensive. We will incorporate other great suggestions from the reviewer in our revised paper.
>
> **Optimistic versus Pessimistic Bias**: Optimistic bias refers to underestimating the expected cost. Plug-in suffers such bias since it is evaluated on the same training data. In contrast, pessimistic bias refers to overestimating the expected cost. CV suffers such bias since CV is unbiased for the evaluation using fewer training samples than the whole dataset, and appears more erroneous than it should be compared to the true evaluation using the whole dataset. We will incorporate this discussion after stating the current Theorems 1 and 2.

---

> > ### Comment · Reviewer_8MSs · 2024-08-10
> > **Rebuttal**
> >
> > Thanks! The additional experiments and comments regarding the choice of interval estimate definitely alleviate many of my concerns, and I will raise my score as a result. Apologies for my ignorance regarding your choice of regime. I still, however, contend that the paper is not a good fit for this format and would fit better for a journal submission.

---

> > > ### Author Response · Authors · 2024-08-12
> > >
> > > We greatly thank you for your constructive feedback and thoughtful reply to our response. We also sincerely thank you for increasing the score for our paper! We understand your consideration of the amount and technical nature of the materials, and we are doing our best to make our comprehensive results digestible and the key messages clear in the main body. Most importantly, we believe our results and messages would gain interests and be very useful to the general ML community -- and we are glad to see that you and other reviewers recognize this as well. Lastly, we also politely note that, compared to other published NeurIPS papers on related subjects in the learning theory field, our 20-page appendix perhaps is not that long according to our best understanding.

---

### Official Review · Reviewer_PDH8 · 2024-07-12

**Soundness:** 3
**Presentation:** 3
**Contribution:** 3
**Rating:** 7
**Confidence:** 3

**Summary:**

This paper considers model evaluation using plug-in method, CV and leave-one-out CV. The main contribution of this paper lies in the asymptotic bias and coverage performance analysis of these methods. The result is that in most cases, it turns out that the plug-in method is no worse than CV or leave-one-out CV.

**Strengths:**

The paper is clearly written and the message is clean. This paper provides new insights regarding CV and leave-one-out CV on model evaluation. I think this work contributes to a recent line of work on understanding the role of CV in model evaluation and model selection.

**Weaknesses:**

It has been recently revealed in [8] that the CV might be estimating the risk of the algorithm, i.e., E(c(\hat{z})), which probably explains why the performance of CV for estimating c(\hat{z}) is not so good. Even in terms of model evaluation, the recent results in [6,9] show that it might be better to treat CV as estimating the averaged performance of a couple of models, so CV might have a large bias for estimating c(\hat{z}).

The same notation of z in l88  and l101 refers to the prediction function and values in the prediction domain, respectively, which is a little bit confusing.

**Questions:**

1.In the overparameterized-regime, we would expect the predictor to interpolate, and the plug-in estimator would be zero, would we expect CV to perform better? In another way of speaking, does the result in this paper assume the dimension is fixed and the sample size goes to infinity?
2.Could the authors give an example where \alpha_n = o(n^{-1/2}) and \gamma \leq 1/4?
3.What happens for algorithm evaluation, i.e., estimating E(c(\hat{z})) and constructing a confidence interval for that? The algorithm evaluation task is an equally important problem.
4.Could we simply use sample splitting to perform model evaluation? For example, if n is large, then we can use half data for training and half data for evaluation, and I think it would give a pretty accurate evaluation for the model trained with n/2 samples.

**Limitations:**

This paper reveals that CV may not be a good choice for model evaluation. So a natural question is to ask is the plug-in method the most reasonable one we can use? Maybe the authors could provide some guidance for practitioners.

---

> ### Author Rebuttal · Authors · 2024-08-07
>
> We sincerely thank the reviewer for recognizing our contributions and clarity, and also for raising the list of very helpful questions.
>
> Regarding "Weaknesses":
>
> **Difference with literature investigating roles of CV**: First, the reviewer mentions that [8] says CV estimates the risk of the algorithm which explains the poor performance of CV for estimating $c(\hat z)$. However, [8] is mostly interested in parametric models, where the difference between $c(\hat{z})$ and $E[c(\hat{z})]$ (or $c(z^*)$) is generally of order $O_p(1/n)$ and negligible compared with the variability, as we mentioned in Section 1 and 3. Therefore, CV still yields valid coverage guarantees for these parametric models.  On the other hand, it is unknown when such bias is significant and leads to invalid coverages in *general nonparametric models* which is our main focus and novelty.
>
> The reviewer also indicates correctly that recent literature [6,9] say that CV accurately estimates the averaged performance and provides valid coverage guarantees for this quantity. However, like above, it is unclear when the difference between such an averaged performance and true model performance affects the coverage guarantee for general nonparametric models.
>
>
> **Notations**: We will change the notation $z$ to $\tilde z$ when referring to the prediction domain.
>
> Regarding "Questions":
>
> **Overparametrized-regime**: We assume the dimension is fixed and the sample size goes to infinity in this paper. We agree with you that the plug-in estimator would be zero under overparametrized-regime. However, whether CV would perform better in that regime is still an open question (e.g., K-fold CV does not perform well in our additional experiments; please refer to more details on the high-dimensional case in our Global Response).
>
> **Examples under given conditions**: $\alpha_n = o(n^{-1/2})$ is satisfied for general random forest and kNN learners from the algorithmic stability literature [14,15,40] discussed in our paper. Moreover, we provide the exact formula of $\gamma_v, \gamma_b$ ($\gamma = \min(\gamma_v, \gamma_b)$) for each learner in Examples 3 and 4 in Section 3. Specifically, in regression problems, $\gamma \leq 1/4$ is satisfied for general nonparametric models (including kNN learners in Example 3, Forest learner in Example 4, kernel estimators in Example 7) as long as $d_x \geq 2$ from Lemma 3 of Appendix B.
>
> **Algorithmic evaluation task**: We agree that algorithmic evaluation is also an important and related task. Our paper focuses on model evaluation but our analysis can shed light on constructing intervals for algorithmic evaluation, especially when the difference of the two targeted intervals is relatively small. In Corollaries 1 and 2 in Appendices E.1.4 and E.2 of our paper, when $\gamma > 1/4$, both plug-in and CV intervals (1) and (2) (regardless of $K$-fold or LOOCV) provide valid coverages for $E_{D_n}[c(\hat z)]$; Otherwise when $\gamma \leq 1/4$, for LOOCV, since $\sqrt{n}(\hat A_{loocv} - c(\hat z))\overset{p}{\to} N(0, \sigma^2)$ and $c(\hat z) = E_{D_n}[c(\hat z)] + O_p(\sqrt{Var_{D_n}[c(\hat z)]})$, the LOOCV interval (2) is valid for covering $E[c(\hat z)]$ when $Var_{D_n}[c(\hat z)] = o(1/n)$; For plug-in, the condition for the coverage validity for $E[c(\hat z)]$ is equivalent to $Var_{D_n}[c(\hat z)] = o(1/n)$ and $\gamma_v > 1/4$. However, it appears open whether there is a more explicit model rate representation for $Var_{D_n}[c(\hat z)] = o(1/n)$.
>
> **Sample Splitting**: We agree with you that such sample splitting gives an accurate evaluation for the model trained with $n/2$ samples. However, we are interested in the model trained with a full set of samples $\hat z(x) = A(D_n;x)$. The expected costs of these two models differ by $\Theta(n^{-2\gamma})$ due to the performance gap when using $n$ versus $n/2$ samples, which is similar to the evaluation bias using 2-fold CV. When $\gamma < 1/4$, the difference is $\omega(n^{-1/2})$, which is large and exceeds the interval length (of order $\Theta(n^{-1/2})$). That is, the interval used to evaluate the model with $n/2$ samples cannot provide valid coverage guarantees for $c(\hat z)$.
>
> **Guidance to practitioners**:
>
> First, in terms of the magnitude of bias, LOOCV is always smaller than plug-in, while plug-in is no larger than K-fold CV. Despite this bias ordering, the adoption of a method over another should also take into account the variability and computational demand, specifically:
> - For parametric models, and nonparametric models with a fast rate ($\gamma > 1/4$, which includes sieves estimators in our reference [18] when the true function $f(x)$ is 2$d_x$-th continuously differentiable in our Examples in Line 198-202), biases in all three considered procedures, plug-in, LOOCV and K-fold CV, are negligible compared to the variability captured in interval coverage. Correspondingly, all three intervals provide valid statistical coverages. Among them, plug-in is the most computationally efficient and should be preferred.
> - For nonparametric models with a slow rate but small variability ($\gamma_v > 1/4, \gamma \leq 1/4$), which include kNN with $k_n = \omega(\sqrt n)$ in Example 3 and the forest learner in Example 4 in our paper, the biases in plug-in and LOOCV are negligible but K-fold is not. Correspondingly, both plug-in and LOOCV provide valid coverages but $K$-fold CV does not. Since plug-in is computationally much lighter than LOOCV, it is again preferred.
> - For nonparametric models with slow rate ($\gamma_v \leq 1/4$), which include kNN with $k_n = \Theta(\sqrt n)$ in Example 3, only LOOCV has a negligible bias and provides valid coverages, and hence should be adopted.
>
> The above being said, we caution that, in terms of the direction of bias, plug-in is optimistic while K-fold CV is pessimistic, and so the latter could be preferred if a conservative evaluation is needed to address high-stake scenarios (as Reviewer tGmJ suggests).

---

> > ### Comment · Reviewer_PDH8 · 2024-08-09
> >
> > Thank you for the detailed response and I will keep my positive score.

---

> > > ### Author Response · Authors · 2024-08-12
> > >
> > > We greatly thank the reviewer again for your constructive feedback and for giving us a high score.

---

### Official Review · Reviewer_JBLc · 2024-07-12

**Soundness:** 4
**Presentation:** 4
**Contribution:** 3
**Rating:** 8
**Confidence:** 3

**Summary:**

The paper argues that Cross-Validation (CV), commonly used to evaluate machine learning models, may not be as statistically beneficial, especially in challenging nonparametric regimes. The paper shows that plug-in is always no worse than K-fold CV for models with any convergence rate. While leave-one-out CV can have a smaller bias than plug-in, this benefit is minimal compared to the variability of the evaluation. The theories are validated by numerical experiments.

**Strengths:**

1. The paper addresses a well-motivated and significant problem to machine learning community: the statistical benefits of cross-validation in model evaluation.
2. The paper is well-written and well-structured, offering comprehensive background information and connecting its conclusions to existing analyses while highlighting its novelty. The framework presented is versatile, applicable to both parametric and nonparametric regimes.
3. The encouraging conclusion that the plug-in method performs no worse than CV is supported by robust theoretical analysis and convincing numerical experiments.

**Weaknesses:**

1. While the paper acknowledges that it is unclear whether the analysis and conclusions can be extended to model selection, I am eager to see more discussion on this topic, possibly including some simple simulation explorations.
2. Although the theories are validated on several synthetic datasets, it would be beneficial to provide experimental results on real-world datasets as well.

**Questions:**

1. In eq (2), every datapoint $(x_{i}, y_{i})$ is used more than once ($k-1$ times) in k-fold CV, the empirical variance should be scaled of $\frac{1}{n(k-1)}$?
2. While the numerical experiments are well-designed, providing results from experiments on real-world data would be even more convincing.

**Limitations:**

The authors clearly state the limitations of the framework, leaving me eager to see the follow-up work.

---

> ### Author Rebuttal · Authors · 2024-08-07
>
> We sincerely thank you for your recognition of the importance and clarity of our paper, and also for your very helpful comments. To respond to your main suggestions, we have run additional experiments on model selection and real-world data set. These experimental results will be incorporated into the Appendix. There we will also provide additional discussion on model selection that we leave for future work, which we reveal below as well.
>
> # Model Selection
>
> A difference between model selection and the model evaluation task we focus on in this paper is that in the former, we focus on the performance rank between different models instead of the absolute performance value. Intuitively, the accuracy of the performance rank depends on not only the evaluations of the compared models, but also the inter-dependence between these evaluation estimates. This adds complexity to understanding the errors made by each selection approach. There are some discussion for specific problems, such as linear regression, classification and density estimation (see, e.g., Section 6 of our reference [3]). However, the theoretical understanding of model selection remains wide open in general problems.
>
> That being said, there are some model selection problems where plug-in easily leads to a naive selection:
> - Selecting hyperparameters: Consider the best regularized parameter $\alpha$ in the ridge regression. Plug-in always chooses $\alpha = 0$ since it has the smallest training loss.
> - Selecting the best model class: Consider the regression problem $\hat f \in argmin_{f \in F_i} \sum_i^n (f(X_i) - Y_i)^2$ for the nested classes $F_1 \subset F_2 \subset F_3$, and we want to select the best $F_i$ among the three classes. Plug-in always selects the largest class, $F_3$, since it has the smallest training loss.
>
> In these problems, CV can select the regularization parameter or model class different from the naive choice. It hints that CV could be better than plug-in for such problem, but this is theoretically not well-understood. Per the reviewer's suggestions, we include some simulation experiments on model selection using plug-in versus CV as follows, where in each case we report the result averaged over 100 experimental repetitions:
>
> ## Case 1: Select the hyperparameter in ridge regression
> Suppose we have a "misspecified" linear data generating process: $Y = \sum_i^{50}(X^{(i)} + sin(X^{(i)})) + \epsilon$ with $X^{(i)}$ being the $i$-th component of $X$ and $\epsilon \sim U(0, 1)$. We want to find the best hyperparameter $\alpha$ from the regularized linear models $\beta_{\alpha} \in argmin_{\beta} (Y_i - \beta^{\top}X_i)^2 + \alpha ||\beta ||^2$ that minimizes the expected cost $E[(Y - \beta_{\alpha}^{\top}X)^2]$. We show the model selection results in the following table, where the first column of each procedure represents the best $\alpha$ that the procedure finds and the second column represents the true cost under such $\alpha$:
>
> |n | Plug-in | | 5-CV| | LOOCV | |
> |--|--|--|--|--|--|--|
> |80|0.0|105.4|10.1|74.1|12.1|70.6|
> |100|0.0|57.2|34.3 |49.9|24.2 | 50.8|
> |200|0.0|36.8|44.4|36.9| 18.2 | 36.5|
>
> When $n$ is small, both 5-CV and LOOCV find a better decision than the plug-in. but the accurate evaluation of LOOCV does not necessarily yield a better selection than 5-CV (e.g. $n = 100$). Plug-in always chooses $\alpha = 0$ and its relative performance depends on the power of regularizations, i.e., when $n = 200$, adding $\alpha$ does not improve much and thus the performance of plug-in to both 5-CV and LOOCV. However, when $n = 80, 100$, plug-in suffers from worse performance than 5-CV and LOOCV.
>
> ## Case 2: Select the best model class
> Given each $D_n$, we want to select the best among ridge, kNN, and random forest models. Continue the same data process as Case 1, but with 10 features. In each experimental repetition, we select $\lambda \sim U(0, 5)$ randomly and generate data from $Y|X: \sum_i^{10} X^{(i)} + \lambda sin(X^{(i)}) + \epsilon$. We show the results of model selection procedures, where the first column of each procedure represents the probability of finding the correct best model, and the second column represents the true cost under the model selected by each procedure:
> |n | Plug-in | | 5-CV| | LOOCV | |
> |--|--|--|--|--|--|--|
> |100|0.52|46.93|0.88|42.07|0.90 |41.64|
> |200|0.57|42.74|0.88|38.19|0.89|37.86|
> |400|0.71|39.94|0.96|37.26|0.95|37.26|
> Here, both 5-CV and LOOCV help find a better decision than plug-in and there is not much difference between 5-CV and LOOCV.
>
> # Real-world Experiments
> We include one real-world dataset [puma32H](https://www.openml.org/d/1210) with 33 features and 1000000 samples as a regression task. The codes and instructions for reproducing are in the updated codebase link in the Global Response. We report the coverage probability of plug-in, 2-CV, and 5-CV for a kNN model with $k_n = n^{2/3}$, where each entry denotes the coverage probability estimated over 100 experimental repetitions for that procedure. Each experimental repetition is conducted differently by shuffling the entire dataset. We then select the first n rows as the training sample for each model and approximate the true model performance by averaging over the remaining samples in the dataset.
>
> |n | 10000| 20000 | 40000|
> |--|--|--|--|
> |Plug-in | **0.94**| **0.93** |**0.90** |
> |2-CV|**0.88**|0.81 |0.76 |
> |5-CV| **0.91**| **0.86**|0.81|
>
> Plug-in provides valid coverages in this case while 2-CV, 5-CV do not, especially when $n$ is larger (20000 and 40000). These continue to validate our asymptotic theory.
>
> **Scaling of the Variance**:
> In equation (2), each datapoint $(x_i, y_i)$ is only used **once for evaluation** since the index of each datapoint is only counted once in one of $\{N_k, k \in [K]\}$, the collection of $K$ partitions of $[n]$.  Note that this formula is the same as the standard all-pairs variance estimator in Theorem 5 in the reference [9].

---

> > ### Comment · Reviewer_JBLc · 2024-08-08
> >
> > Thank you for the detailed discussions and all the experiments. The experiment results look good to me, so I have raised the score.

---

> > > ### Author Response · Authors · 2024-08-12
> > >
> > > We greatly thank the reviewer for your constructive feedback earlier, and for reading our reply closely and increasing your score!

---

### Author Rebuttal · Authors · 2024-08-07

We sincerely thank all the reviewers for their helpful suggestions. In this Global Response, we provide additional discussion on several aspects that address the reviewers' comments: practical guidance from our results, differences with existing literature regarding roles of CV, high-dimensional problems, and additional experimental results.

# Guidance to Practitioners
First, in terms of the magnitude of bias, LOOCV is always smaller than plug-in, while plug-in is no larger than K-fold CV. Despite this bias ordering, the adoption of a method over another should also take into account the variability and computational demand, specifically:
- For parametric models, and nonparametric models with a fast rate ($\gamma > 1/4$, including sieves estimators in our reference [18] when the true function $f(x)$ is 2$d_x$-th continuously differentiable in our Examples in Line 198-202), biases in all three considered procedures, plug-in, LOOCV and K-fold CV, are negligible compared to the variability captured in interval coverage. Correspondingly, all three intervals provide valid statistical coverages. Among them, plug-in is the most computationally efficient and should be preferred.
- For nonparametric models with a slow rate but small variability ($\gamma_v > 1/4, \gamma \leq 1/4$), which include kNN with $k_n = \omega(\sqrt n)$ in Example 3 and the forest learner in Example 4 in our paper, the biases in plug-in and LOOCV are negligible but K-fold is not. Correspondingly, both plug-in and LOOCV provide valid coverages but $K$-fold CV does not. Since plug-in is computationally much lighter than LOOCV, it is again preferred.
- For nonparametric models with slow rate ($\gamma_v \leq 1/4$), which include kNN with $k_n = \Theta(\sqrt n)$ in Example 3, only LOOCV has a negligible bias and provides valid coverages, and hence should be adopted.

The above being said, we caution that, in terms of the direction of bias, plug-in is optimistic while K-fold CV is pessimistic, and so the latter could be preferred if a conservative evaluation is needed to address high-stake scenarios.


# Comparison with Existing Literature Investigating Roles of CV
Our paper considers a distinct setting from [8] in that we consider general nonparametric models instead of (high-dimensional) parametric models, and provide corresponding theoretical guarantees on bias and coverage on competing procedures. To this end, it has been unknown when the biases of these procedures are significant and lead to invalid coverages in *general nonparametric models* which is our main focus and novelty. Additional discussions are provided in response to Reviewer PDH8.

To address the variance estimate problem for high-dimensional parametric models,  [8] proposes nested CV (NCV) in both the point and variance estimates in their interval. However, their remedy is not designed for, and also is not empirically competitive, on the nonparametric settings that we consider. Additional discussions are provided in response to Reviewer 8MSs.

# Additional Experiments
In our attached pdf, we include additional numerical results using different $K$ in CV in Table 1, which continues to provide the insights that K-fold CV suffers from poor coverage when n is large; We also include numerical comparisons between naive CV (in equation (2) of our paper) and nested CV (in our reference [8]) in Table 2; and some high-dimensional simulations in Table 3.

Here is the anonymous link to our revised code: [Link](https://anonymous.4open.science/r/CV_GoldStandard-8E35)

# High Dimensional Discussion
For the high-dimensional setup, i.e., feature dimension $p$ and sample size $n$ both go to infinity such that $p/n$ converges to a nonzero constant, besides specific problems like (generalized) linear regression, the problem is wide open to our best knowledge. For linear models, it is known regarding the bias of our considered three procedures that:
- For plug-in, when $p > n$, the predictor interpolates the training data so that the training loss becomes zero (unless we add regularization); when $p/n \to c \in (0, 1)$, plug-in still suffers a large bias from overfitting and cannot be used directly to construct the point estimator and confidence interval to evaluate model performance. Some bias correction procedures ([26]) are proposed to construct consistent point estimators in these high-dimensional scenarios.
- For $K$-fold CV, the point estimate also suffers from a non-vanishing bias [W2018] and may not be a good choice for evaluating model performance.
- For LOOCV, recent literature [W2018,P2021] shows that its bias goes to zero.


Despite the known results above for the bias, no theoretically valid coverage guarantees exist for high-dimensional linear models in the literature, since almost all existing CV literature is based on some stability conditions and is only valid under low-dimensional asymptotic regimes ([6,8,37] and ours).

For our three procedures, we validate claims for the point estimates above and also investigate the coverage performance of interval estimates (open question), using the same simulation data setup for ridge regression with $\alpha = 1$ as in our regression problems. We show the simulation results in Table 3 of the attached pdf, where both plug-in and 5-CV suffer from large bias and have zero coverage. The bias of LOOCV is small but the LOOCV interval suffers from poor coverage. These results indicate that constructing theoretically valid intervals remains open and challenging for high-dimensional problems, and is important to devise improved CV or bias correction approaches.

[W2018] Wang et al. Approximate LOO for fast parameter tuning in high dimensions. ICML 2018.

[P2021] Patil et al. Uniform consistency of cross-validation estimators for high-dimensional ridge regression. AISTATS 2021.

---

### Decision · Program_Chairs · 2024-09-25

**Decision:**

Accept (poster)

**Comment:**

The reviews are very strong and agreement is high. That said, there are some
fairly significant points of context, some of which are highlighted by the
reviewers that are necessary for the authors to describe carefully in their
revision.

1. By K-fold, the authors mean K is O(1) or even "fixed K". Fixed K is the common
   use case in ML, but not generally the case for theoretical results. For
   example, [3] discusses specifically that K should grow with n for reasonable
   risk estimation performance. (See also the Weaknesses highlighted by tGmj).
2. The authors emphasis is on "estimation" not on "model selection". This is
   important. And the distinction is made clear in the experiments used in the
   Author Rebuttal to JBLc: using plug-in for model selection can be
   catastrophically poor. Additionally, the knn example here is obvious: plug-in
   will always select 1 nearest neighbor with 0 plug-in error. The authors draw out this
   distinction slightly in the Discussion, but it deserves more prominent emphasis.
   And the discussion of "interval" vs. "point estimation" for the prediction risk
   is key to Example 3 (see also, e.g. Table 2 Portfolio Optimization, where plug-in
   in this context has 0 coverage and massive bias).
3. The typical ML use-cases for K-fold CV are (1) tuning models and (2)
   comparing different models (usually to demonstrate that "my novel model in
   this NeurIPS paper is better than all the competitors"), not so much "risk estimation".
   The introduction and
   title seems to suggest that we can just use the plug-in! This is wrong, and
   not what the paper's theory says (nor, what the authors really claim).
   But this distinction is not made clear by the title or the introduction.
4. The necessary conditions (lines 61--62: "allows us to identify necessary
   conditions, in contrast to merely sufficient conditions in the literature,
   under which plug-in and CV variants exhibit low biases and valid coverages")
   are not weak and the range of problems covered is specific (low dimensions,
   bounded predictions or cost, LOO stability, specific range of gamma). This is
   also not described in the introduction.

The message of this paper needs to be carefully calibrated.
As the reviewers suggest, this paper may well question long-held beliefs of the ML
community in a provocative way. But it is written to be
provocative, possibly claiming more than it delivers in a way that could
result in some danger. Without more nuanced discussion of its deliverables, and
additional context, casual readers may use this paper as license
to use the plug-in estimator to compare models, keying on the statement (line
164 "plug-in is always no worse than K-fold CV"). I would urge the authors to
replace the statements of "evaluate model performance" with "estimate
out-of-sample prediction risk". This is more precise and precludes some of the
potential misunderstandings that could arise.